# STOCHASTIC EXTRAGRADIENT WITH FLIP-FLOP SHUF- FLING & ANCHORING: PROVABLE IMPROVEMENTS

## ABSTRACT

In minimax optimization, the extragradient (EG) method has been extensively studied because it outperforms the gradient descent-ascent (GDA) method in both strongly-convex-strongly-concave (SC-SC) and convex-concave (C-C) problems. However, stochastic EG (SEG) has seen limited success in C-C problems, especially in unconstrained cases: known results suffer limitations such as uniformly bounded gradient variance, bounded domain, increasing batch size, monotonicity of components, and/or absence of convergence rates. Motivated by the recent progress in analysis of shuffling-based stochastic optimization methods, we investigate the convergence of *shuffling-based SEG* in unconstrained finite-sum minimax problems, in search of improved convergence guarantees for SEG under minimal algorithm modifications. Our analysis reveals that both random reshuffling and the recently proposed flip-flop shuffling (Rajput et al., 2022) alone can still suffer divergence in C-C problems. However, with an additional simple trick called anchoring, we develop the *SEG with flip-flop anchoring* (SEG-FFA) method which successfully converges in C-C problems, overcoming all the limitations above. We also show upper and lower bounds in the SC-SC setting, demonstrating that SEG-FFA has a provably faster convergence rate compared to other shuffling-based methods.

## 1 INTRODUCTION

Minimax problems with a finite-sum structure, which are optimization problems of the form

$$\min_{\boldsymbol{x}} \max_{\boldsymbol{y}} \ f(\boldsymbol{x}, \boldsymbol{y}) := \frac{1}{n} \sum_{i=1}^{n} f_i(\boldsymbol{x}, \boldsymbol{y}), \tag{1}$$

can be found in many interesting applications in machine learning, such as generative adversarial networks (GANs) (Goodfellow et al., 2014), adversarial training (Mądry et al., 2018), multi-agent reinforcement learning (Wai et al., 2018), fair classification (Mohri et al., 2019), and so on. Deterministic methods for minimax problems, such as *gradient descent-ascent* (GDA) (Arrow & Hurwicz, 1956) and *extragradient* (EG) (Korpelevich, 1976), have been extensively studied in the literature. It is though known that, unlike *gradient descent* (GD) for minimization problems, GDA may diverge even when $f$ is convex on $\boldsymbol{x}$ and concave on $\boldsymbol{y}$. On the other hand, EG finds an optimum under this convex-concave setting (Gorbunov et al., 2022b), and moreover, attains a convergence rate faster than GDA (Azizian et al., 2020) when $f$ is strongly convex on $\boldsymbol{x}$ and strongly concave on $\boldsymbol{y}$.

In contrast, attempts to construct stochastic variants of these algorithms have not been so fruitful. When $f$ is convex-concave, *stochastic gradient descent-ascent* (SGDA) clearly can diverge, as the deterministic GDA does not. To make matters worse, *stochastic extragradient* (SEG) methods have also had limited success on convex-concave problems. As we summarize in Section 2 in more detail, existing versions of SEG and their analyses have limitations that hinder its application to general unconstrained convex-concave problems, such as uniformly bounded gradient variance, bounded domain, increasing batch size, monotonicity of components, and/or absence of convergence rates.

In the context of finite-sum optimization, most existing theoretical studies on stochastic methods (including the aforementioned results) have long been based on the *with-replacement* sampling scheme. In with-replacement sampling, at each iteration $t$ an index $i(t)$ is independently and uniformly sampled among $\{1, \ldots, n\}$. Such a sampling scheme is relatively easy to theoretically analyze, because

the sampled $f_{i(t)}$ is an unbiased estimator of the full objective function $f$. In practice, however, inspired by the empirical observations of faster convergence (Bottou, 2009; Recht & Ré, 2013), the *without-replacement* sampling schemes have been the de facto standard. Among them, the most popular one is the *random reshuffling* (RR) scheme, where in every *epoch* consisting of $n$ iterations, the indices are chosen exactly once in a randomly shuffled order.

This gap between theory and practice is being closed by the recent breakthroughs in stochastic gradient descent (SGD), namely that SGD with RR leads to a provably faster convergence compared to with-replacement SGD when the number of epochs is large enough (Nagaraj et al., 2019; Ahn et al., 2020; Mishchenko et al., 2020a; Nguyen et al., 2021; Yun et al., 2021; 2022). This has motivated further studies on finding other shuffling-based sampling schemes that can improve upon RR, resulting in the discoveries such as the *flip-flop* scheme (Rajput et al., 2022) and GraB (Lu et al., 2022; Cha et al., 2023). The flip-flop scheme is a particularly simple yet interesting modification of RR with improved rates in quadratic problems: a random permutation is used twice in a single epoch (i.e., two passes over $n$ components in an epoch), but the order is reversed in the second pass.

The aforementioned progress in minimization also triggered the study of stochastic minimax methods with shuffling. Similar to minimization problems, SGDA with RR indeed converges faster than the with-replacement SGDA, under assumptions such as strongly-convex-strongly-concave objectives (Das et al., 2022), or $f$ satisfying the Polyak-Łojasiewicz condition (Cho & Yun, 2023). Despite the superiority of EG over GDA, *SEG with shuffling* has not yet been investigated in the literature, and the promising achievements in SGDA motivates us to study the following question:

> ***Can shuffling schemes provide improved convergence guarantees for SEG, in unconstrained convex-concave and strongly-convex-strongly-concave settings?***

More specifically, we are interested in developing shuffling-based variants of SEG with minimal modifications to the algorithm in unconstrained finite-sum minimax problems, and showing that (a) in convex-concave settings, the new method reaches an optimum with a guarantee on the rate of convergence, overcoming the aforementioned limitations of existing results; (b) in strongly-convex-strongly-concave settings, the method converges faster than existing SGDA/SEG variants.

## 1.1 Our Contributions

In this paper, we propose the stochastic extragradient with flip-flop anchoring (SEG-FFA) method, which is SEG amended with the techniques of *flip-flop* sampling scheme and *anchoring*.[1] With such minimal modifications to SEG, we show that SEG-FFA achieves provably improved convergence guarantees. More precisely, our contributions can be listed as follows (see Table 1 for a summary). For clarity, we use SEG-US to refer to with-replacement SEG (US for uniform sampling).

- We first study SEG with RR (SEG-RR) and SEG with flip-flop (SEG-FF), and find out that shuffling alone does *not* fix the divergence issue of SEG-US[2] in convex-concave functions. In particular, we demonstrate that SEG-RR and even SEG-FF still fail to converge in the convex-concave setting, by constructing an explicit counterexample (Theorem 4.2).

- We next investigate the underlying cause for the nonconvergence of SEG-RR and SEG-FF, and a way to remedy this issue. In particular, we identify that either they fail to match the update equation of the reference method EG beyond *first-order* Taylor expansion terms, or attempting to match both the *first-* and *second-order* Taylor expansion terms results in divergence (Proposition 5.2). By adopting a simple technique called *anchoring* on top of flip-flop shuffling, we devise our algorithm SEG-FFA, whose epoch-wise update deterministically matches EG up to second-order Taylor expansion terms (Proposition 5.3).

---

[1] The anchoring in this paper simply averages the initial and terminal points of the epoch. The readers should not be confused with the anchoring in (Cai et al., 2022; Yoon & Ryu, 2021; Lee & Kim, 2021), which are based specifically on the *Halpern iteration* (Halpern, 1967).

[2] This paper focuses on the *same-sample* version of SEG, which uses a single sample for both extrapolation and update steps in the update of SEG (see Beznosikov et al. (2023) for the details). So, to be precise, we are stating here that the *same-sample* SEG-US suffers divergence, which does not contradict Hsieh et al. (2020) where the authors show that the *independent-sample* (i.e., independent samples for extrapolation and update steps) version of SEG converges to optima for convex-concave settings, albeit without a convergence rate.

Table 1: Summary of upper/lower bounds. Pseudocode of algorithms can be found in Appendix A. We only display terms that become dominant for sufficiently large $T$ and $K$. To compare the with-replacement versions (-US) against shuffling-based versions, one can substitute $T = nK$. The optimality measure used for SC-SC problems is $\mathbb{E}[\|\hat{z} - z^*\|^2]$ for the last iterate $\hat{z}$. For C-C problems, we consider $\min_{t=0,\ldots,T} \mathbb{E}[\|Fz_t\|^2]$ for with-replacement methods and $\min_{k=0,\ldots,K} \mathbb{E}[\|Fz_0^k\|^2]$ for shuffling-based methods.

| Method | Strongly-Convex-Strongly-Concave | | Convex-Concave | |
|--------|---------------|---------------|---------------|---------------|
| | Upper Bound | Lower Bound | Upper Bound | Lower Bound |
| SGDA-US | $\mathcal{O}(\frac{1}{T})$ (Loizou et al., 2021) | $\Omega(\frac{1}{T})$ (Cho & Yun, 2023) | N/A | $\Omega(1)$ (as GDA) |
| SEG-US | $\mathcal{O}(\frac{1}{T})$ (Gorbunov et al., 2022a) | $\Omega(\frac{1}{T})$ (Beznosikov et al., 2020) | N/A | $\Omega(1)$ (Thm. 4.2) [†] |
| SGDA-RR | $\tilde{\mathcal{O}}(\frac{1}{nK^2})$ (Das et al., 2022) | $\Omega(\frac{1}{nK^3})$ (Thm. 5.6) | N/A | $\Omega(1)$ (as GDA) |
| SEG-RR | $\mathcal{O}(\frac{1}{nK^{2-3\varepsilon}})$ (Thm. 4.1) | $\Omega(\frac{1}{nK^3})$ (Thm. 5.6) | N/A | $\Omega(1)$ (Thm. 4.2) |
| SEG-FF | $\mathcal{O}(\frac{1}{nK^{2-3\varepsilon}})$ (Thm. 4.1) | – | N/A | $\Omega(1)$ (Thm. 4.2) |
| SEG-FFA | $\mathcal{O}(\frac{1}{nK^{4-5\varepsilon}})$ (Thm. 5.5) | – | $\tilde{\mathcal{O}}(\frac{1}{K^{1/3}})$ (Thm. 5.4) | – |

[†] Hsieh et al. (2020) also show $\Omega(1)$ bounds, but for independent-sample SEG with stepsize $\alpha_t = \beta_t$.

- We prove that SEG-FFA enjoys improved convergence guarantees, as anticipated by our second-order matching principle. Most importantly, we show that SEG-FFA achieves a convergence rate of $\tilde{\mathcal{O}}(1/K^{1/3})$ when $f$ is convex-concave. This result is in stark contrast to other baseline algorithms that diverge under this setting (see the last column of Table 1), and it overcomes limitations of existing SEG results on unconstrained setups.

- Moreover, we show that when $f$ is strongly-convex-strongly-concave, SEG-FFA achieves a convergence rate of $\mathcal{O}(1/nK^{4-5\varepsilon})$ for any fixed $0 < \varepsilon < 2/3$, where $K$ denotes the number of epochs (Theorem 5.5). Additionally, by proving lower bounds $\Omega(1/nK^3)$ for the convergence rates of SGDA-RR and SEG-RR under the same setting (Theorem 5.6), we show that SEG-FFA has a *provable advantage* over these baseline algorithms.

## 2 RELATED WORKS

**Extragradient and EG+** Extragradient method (Korpelevich, 1976) is a widely used minimax optimization method, well-known for resolving the nonconvergence issue of GDA on convex-concave problems. It has been observed by Mokhtari et al. (2020) that this advantage of EG over GDA comes from the fact that the Taylor expansion of update equations of EG and the *proximal point* (PP) method (Martinet, 1970) match each other up to the second-order terms, i.e., $\mathcal{O}(\eta^3)$ approximation error when $\eta$ is the stepsize. In contrast, GDA matches PP only up to first-order terms, being an approximation with error $\mathcal{O}(\eta^2)$.

In this paper, we also consider EG+ (Diakonikolas et al., 2021), which is a generalization of EG. The update rule of EG+ is defined, for stepsizes $\{\eta_{1,k}\}_{k \geq 0}$ and $\{\eta_{2,k}\}_{k \geq 0}$, as

$$\begin{cases} u^k \leftarrow x^k - \eta_{1,k} \nabla_x f(x^k, y^k) \\ v^k \leftarrow y^k + \eta_{1,k} \nabla_y f(x^k, y^k) \end{cases}, \qquad \begin{cases} x^{k+1} \leftarrow x^k - \eta_{2,k} \nabla_x f(u^k, v^k) \\ y^{k+1} \leftarrow y^k + \eta_{2,k} \nabla_y f(u^k, v^k) \end{cases}. \qquad (2)$$

If $f$ is convex-concave, Diakonikolas et al. (2021) show that EG+ reaches an optimum when $\eta_{1,k} \geq \eta_{2,k}$. In particular, when $\eta_{1,k} = \eta_{2,k}$, we recover the standard EG by Korpelevich (1976).

**Shuffling and flip-flop** As previously described, shuffling-based stochastic methods can outperform the methods based on with-replacement sampling, both in theory and in practice. One key property of shuffling-based methods is that, while the individual estimators become biased as they become dependent to other estimators within the same epoch, the overall stochastic error across the epoch decreases dramatically compared to using $n$ independent unbiased estimators. For instance, in SGD with RR (Ahn et al., 2020) and in SGDA with RR (Das et al., 2022), the overall progress made within each epoch exactly matches their deterministic counterparts up to the first-order, leaving an error as small as $\mathcal{O}(\eta^2)$, where $\eta$ is the stepsize. Rajput et al. (2022) observe that, when each

component functions are convex quadratics, then using flip-flop on SGD can reduce the error further to $\mathcal{O}(\eta^3)$, resulting in an even faster convergence.

**Stochastic Variants of Extragradient**   While EG improves upon GDA, unfortunately, SEG has not been able to show a clear advantage over SGDA. To the best of our knowledge, for general unconstrained convex-concave problems, the existing stochastic variants of EG and their analyses face several limitations: (i) many assume that the components have uniformly bounded gradient variance (Diakonikolas et al., 2021; Cai et al., 2022; Pethick et al., 2023), which becomes particularly restrictive for unconstrained setups (see discussion after Assumption 4)[3]; (ii) some (implicity) require that the domain is bounded (Juditsky et al., 2011; Mishchenko et al., 2020b); (iii) some require increasing the batch size for convergence (Diakonikolas et al., 2021; Cai et al., 2022); (iv) sometimes each stochastic component is assumed to be monotone (Mishchenko et al., 2020b); (v) convergence is proved without the above four restrictions in (Hsieh et al., 2020), albeit for independent-sample SEG, but the result lacks a convergence rate. Variance reduction techniques have also been considered (Carmon et al., 2019; Alacaoglu & Malitsky, 2022), but in this case the access to the full objective function $f$ (or its gradient) is assumed. We note that our proposed SEG-FFA overcomes the aforementioned limitations, and reaches an optimum with an explicit rate in unconstrained convex-concave problems, under relatively mild conditions. The readers may also refer to Beznosikov et al. (2023) for a comprehensive overview on this topic.

## 3   NOTATIONS AND PROBLEM SETTINGS

Let $[n] \subset \mathbb{Z}$ denote the set $\{1, \ldots, n\}$. The set of all permutations on $[n]$ will be denoted by $\mathcal{S}_n$.

Recall that we are considering the finite-sum minimax problem as in (1). We denote the *saddle gradient* operators by

$$\boldsymbol{F}(\,\cdot\,) \coloneqq \begin{bmatrix} \nabla_{\boldsymbol{x}} f(\,\cdot\,) \\ -\nabla_{\boldsymbol{y}} f(\,\cdot\,) \end{bmatrix}, \qquad \boldsymbol{F}_i(\,\cdot\,) \coloneqq \begin{bmatrix} \nabla_{\boldsymbol{x}} f_i(\,\cdot\,) \\ -\nabla_{\boldsymbol{y}} f_i(\,\cdot\,) \end{bmatrix}, \quad i = 1, \ldots, n.$$

The derivatives of the operators will be denoted with a prefix $D$. For example, the derivative of $\boldsymbol{F}$ is denoted by $D\boldsymbol{F}$. Often a single vector will be used to denote the minimization and the maximization variable at once. For instance, for $\boldsymbol{z} \in \mathbb{R}^{d_1+d_2}$ which is a concatenation of $\boldsymbol{x} \in \mathbb{R}^{d_1}$ and $\boldsymbol{y} \in \mathbb{R}^{d_2}$, we simply write $\boldsymbol{F}\boldsymbol{z}$ to denote $\boldsymbol{F}(\boldsymbol{x}, \boldsymbol{y})$.

It can be shown that, if $f$ is $\mu$-strongly convex on $\boldsymbol{x}$ and $\mu$-strongly concave on $\boldsymbol{y}$ for some $\mu \geq 0$, then its saddle gradient $\boldsymbol{F}$ is $\mu$-strongly monotone, in the following sense (see e.g., Grimmer et al. (2023) for a proof of this standard fact).

**Assumption 1** (Strong monotonicity). *For $\mu > 0$, we say that an operator $\boldsymbol{F}$ is $\mu$-strongly monotone if for any $\boldsymbol{z}, \boldsymbol{w} \in \mathbb{R}^{d_1+d_2}$, it holds that*

$$\langle \boldsymbol{F}\boldsymbol{z} - \boldsymbol{F}\boldsymbol{w}, \boldsymbol{z} - \boldsymbol{w} \rangle \geq \mu \|\boldsymbol{z} - \boldsymbol{w}\|^2. \tag{3}$$

**Assumption 1′** (Monotonicity). *If (3) holds for $\mu = 0$, then we say that $\boldsymbol{F}$ is monotone.*

Thus, from now on, we will use the term *strongly monotone* (respectively, *monotone*) problems rather than strongly-convex-strongly-concave (respectively, convex-concave) problems. Notice that we only assume that the full saddle gradient $\boldsymbol{F}$ is (strongly) monotone, not for each $\boldsymbol{F}_i$'s.

Other three underlying assumptions we make on the problem (1) can be listed as follows.

**Assumption 2** (Existence of a Solution). *The optimal solution of the problem (1), which we denote by $\boldsymbol{z}^* = (\boldsymbol{x}^*, \boldsymbol{y}^*)$, exists in $\mathbb{R}^{d_1+d_2}$.*

We make this existence assumption for convex-concave problems in order to exclude pathological problems such as $f(x, y) = x + y$.

**Assumption 3** (Operator Smoothness). *Each $f_i$ is $L$-smooth, and each $\boldsymbol{F}_i$ is $M$-smooth. In other words, we have for any $\boldsymbol{z}, \boldsymbol{w} \in \mathbb{R}^{d_1+d_2}$,*

$$\text{(i) } \|\boldsymbol{F}_i \boldsymbol{z} - \boldsymbol{F}_i \boldsymbol{w}\| \leq L \|\boldsymbol{z} - \boldsymbol{w}\|, \qquad \text{(ii) } \|D\boldsymbol{F}_i \boldsymbol{z} - D\boldsymbol{F}_i \boldsymbol{w}\| \leq M \|\boldsymbol{z} - \boldsymbol{w}\|. \tag{4}$$

---

[3]Gorbunov et al. (2022a) do not assume any condition on the variance, but their results require a strong regularity condition that is close to assuming strong monotonicity for the majority of component functions.

It is worth mentioning that the gradient operator $\boldsymbol{F}_i$ arising from a quadratic function $f_i$ is $M$-smooth with $M = 0$. Notice also that, by the finite-sum structure $\boldsymbol{F} = \frac{1}{n}\sum_{i=1}^{n}\boldsymbol{F}_i$, it is then clear that Assumption 3 implies $f$ being $L$-smooth and $\boldsymbol{F}$ being $M$-smooth. The $L$-smoothness assumption on the objective functions is standard in the optimization literature, while the $M$-smoothness assumption on the saddle gradients may look less standard. This smoothness assumption on the saddle gradient (or the Lipschitz Hessian condition) for analyzing SEG-FFA stems from the analysis of the flip-flop sampling scheme (Rajput et al., 2022). In particular, this is needed for bounding the high-order error terms between the (deterministic) EG and SEG-FFA in Section 5.1. Indeed, existing analysis of flip-flop sampling (Rajput et al., 2022) is limited to quadratic functions that trivially have the 0-Lipschitz Hessian, so our analysis is a step forward. Admittedly, the Lipschitz Hessian condition may still look rather strong, but we would like to also point out that this assumption does not lead to unfair comparisons against other algorithms; we show our divergence and lower bound results (Theorems 4.2 and 5.6) for the baseline algorithms using *quadratic* functions (i.e., $M = 0$). We leave studying how one can remove this assumption as an interesting future work.

**Assumption 4** (Component Variance). *There exist constants $\rho \geq 0$ and $\sigma \geq 0$ such that*

$$\frac{1}{n}\sum_{i=1}^{n}\|\boldsymbol{F}_i\boldsymbol{z} - \boldsymbol{F}\boldsymbol{z}\|^2 \leq (\rho\|\boldsymbol{F}\boldsymbol{z}\| + \sigma)^2 \qquad \forall \boldsymbol{z}. \tag{5}$$

In the existing works studying stochastic optimization methods for minimax problems, it is almost always the case where Assumption 4 with $\rho = 0$ is imposed. This *uniform* bound on the variance simplifies the convergence analyses, but it is also fairly restrictive especially in the unconstrained settings. Already for bilinear finite-sum minimax problems, one can easily check that setting $\rho = 0$ prohibits any "noise" in the bilinear terms in component functions. For machine learning applications also, it has been reported that this assumption often fails to hold (Beznosikov et al., 2023). Allowing the variance to grow with the gradient $\boldsymbol{F}\boldsymbol{z}$ makes the assumption much more realistic.

## 4 Shuffling Alone Is Not Enough in Monotone Cases

Under the settings we have discussed, let us study the consequences of applying shuffling-based sampling schemes to SEG. First we describe the precise methods of our consideration, namely the SEG-RR and SEG-FF.

For $k \geq 0$, in the beginning of an epoch, a random permutation $\tau_k$ is sampled from a uniform distribution over $\mathcal{S}_n$. Then, for $n$ iterations, we use each of the component functions once, in the order determined by $\tau_k$. That is, for $i = 0, 1, \ldots, n-1$ we do

$$\begin{aligned}
\boldsymbol{w}_i^k &\leftarrow \boldsymbol{z}_i^k - \alpha_k \boldsymbol{F}_{\tau_k(i+1)}\boldsymbol{z}_i^k, \\
\boldsymbol{z}_{i+1}^k &\leftarrow \boldsymbol{z}_i^k - \beta_k \boldsymbol{F}_{\tau_k(i+1)}\boldsymbol{w}_i^k,
\end{aligned} \tag{6}$$

for some stepsizes $\alpha_k$ and $\beta_k$. In case of using SEG-RR, the epoch is completed here, and we set $\boldsymbol{z}_0^{k+1} \leftarrow \boldsymbol{z}_n^k$ as the initial point for the next epoch.

In case of SEG-FF, we perform $n$ more iterations in the epoch, as proposed in Rajput et al. (2022). In these additional iterations, the component functions are each used once more, but in reverse order. That is, for $i = n, n+1, \ldots, 2n-1$, we do

$$\begin{aligned}
\boldsymbol{w}_i^k &\leftarrow \boldsymbol{z}_i^k - \alpha_k \boldsymbol{F}_{\tau_k(2n-i)}\boldsymbol{z}_i^k, \\
\boldsymbol{z}_{i+1}^k &\leftarrow \boldsymbol{z}_i^k - \beta_k \boldsymbol{F}_{\tau_k(2n-i)}\boldsymbol{w}_i^k.
\end{aligned} \tag{7}$$

Then we set $\boldsymbol{z}_0^{k+1} \leftarrow \boldsymbol{z}_{2n}^k$ as the initial point for the next epoch. The full pseudocode of these methods can be found in Appendix A.

When $\boldsymbol{F}$ is strongly monotone, it is possible to show that both SEG-RR and SEG-FF indeed provide speed-up over SEG-US, by proving a $\mathcal{O}(1/nK^{2-3\varepsilon})$ rate of convergence for a small $\varepsilon > 0$.

**Theorem 4.1.** *Suppose that $\boldsymbol{F}$ is $\mu$-strongly monotone with $\mu > 0$, Assumptions 3, 4 hold, and we are running either SEG-RR or SEG-FF for $K$ epochs. For any given $\varepsilon \in (0, 1/2)$, let the stepsize be chosen to be a constant $\alpha_k = \beta_k = \frac{\omega}{nK^{1-\varepsilon}}$ for a sufficiently small constant $\omega$. Then we achieve the bound*

$$\mathbb{E}\left[\|\boldsymbol{z}_0^K - \boldsymbol{z}^*\|^2\right] \leq \exp\left(-\frac{\mu\omega K^{\varepsilon}}{2}\right)\|\boldsymbol{z}_0^0 - \boldsymbol{z}^*\|^2 + \mathcal{O}\left(\frac{1}{nK^{2-3\varepsilon}}\right). \tag{8}$$

*Proof.* See Section 6 for the proof sketch, and the appendices referred therein for the full proof. □

The well-known convergence rate of SEG-US under strong monotonicity of $\boldsymbol{F}$ is $\Theta(1/T)$, where $T$ is the total number of iterations (Gorbunov et al., 2022a; Beznosikov et al., 2020). Translating this rate to our shuffling-based setting, where there are $\Theta(n)$ iterations per epoch, this rate amounts to $\Theta(1/nK)$. Therefore, Theorem 4.1 shows that simply by controlling the order of how component functions are sampled we can have an improvement in the speed of convergence when $K$ is large enough. Considering that SGDA-RR and PP with RR achieves the rate of $\tilde{\mathcal{O}}(1/nK^2)$ in the strongly monotone setting (Das et al., 2022), the performance of SEG-RR and SEG-FF is on par with the existing known methods.

However, it turns out that the benefit of shuffling does not extend further beyond the "easy" strongly monotone setting. In fact, when $\boldsymbol{F}$ is merely monotone, then in the worst case, SEG-RR and SEG-FF suffers from nonconvergence, just as in the case of SEG-US.

**Theorem 4.2.** *For $n = 2$, there exists a minimax problem with $f(x,y) = \frac{1}{2}\sum_{i=1}^{2}f_i(x,y)$ having a monotone $\boldsymbol{F}$, consisting of $L$-smooth quadratic $f_i$'s satisfying Assumption 4 with $(\rho,\sigma) = (1,0)$, such that SEG-US, SEG-RR and SEG-FF diverge in expectation for any positive stepsizes.*

*Proof.* We provide the explicit counterexample and the proof of divergence in Appendix G.1. In particular, for SEG-US we show $\mathbb{E}[\|\boldsymbol{F}\boldsymbol{z}_{t+1}\|^2] > \mathbb{E}[\|\boldsymbol{F}\boldsymbol{z}_t\|^2]$ for all iterations $t \geq 0$, and for SEG-RR and SEG-FF, we show $\mathbb{E}[\|\boldsymbol{F}\boldsymbol{z}_0^{k+1}\|^2] > \mathbb{E}[\|\boldsymbol{F}\boldsymbol{z}_0^k\|^2]$ for all epochs $k \geq 0$. These results indicate that $\min_{t=0,\ldots,T}\mathbb{E}[\|\boldsymbol{F}\boldsymbol{z}_t\|^2] = \Omega(1)$ for SEG-US and $\min_{k=0,\ldots,K}\mathbb{E}[\|\boldsymbol{F}\boldsymbol{z}_0^k\|^2] = \Omega(1)$ for SEG-RR and SEG-FF, as summarized in Table 1. □

## 5 SEG-FFA: SEG WITH FLIP-FLOP ANCHORING

In this section, we investigate the underlying cause for nonconvergence of SEG-RR and SEG-FF from the perspective of how accurately they match the convergent EG or PP methods in terms of the Taylor expansions of updates. Our analysis is in line with the existing analysis showing the superiority of EG over GDA by comparing how close they are to the PP updates (Mokhtari et al., 2020) and also the analysis of shuffling-based SGD as an approximation of GD updates (Ahn et al., 2020; Rajput et al., 2022).

As an outcome of our analysis, we will propose adding a simple *anchoring* step at the end of each epoch of SEG-FF. After the $2n$ iterations described by (6) and (7), for a predetermined constant $\theta_k \geq 0$ we perform an additional averaging of the initial iterate $\boldsymbol{z}_0^k$ and the last iterate $\boldsymbol{z}_{2n}^k$:

$$\boldsymbol{z}_0^{k+1} \leftarrow \frac{\boldsymbol{z}_{2n}^k + \theta_k \boldsymbol{z}_0^k}{1 + \theta_k}, \tag{9}$$

when setting the initial point for the next epoch. As both flip-flop and anchoring are used, we call this method *Stochastic ExtraGradient with Flip-Flop Anchoring* (SEG-FFA). The particular choice of parameters we employ is $\alpha_k = \beta_k/2$ and $\theta_k = 1$. One may check the pseudocode of SEG-FFA, presented as Algorithm 4 in Appendix A, to see the precise description of the method.

### 5.1 DESIGNING SEG-FFA VIA SECOND-ORDER MATCHING

Let us provide a sketch of the motivations behind introducing the anchoring step (9) and our choice of the parameters $\alpha_k = \beta_k/2$ and $\theta_k = 1$. Proof details are deferred to Section 6 and Appendix C.

As observed by Mokhtari et al. (2020), the key feature of EG behind its superior convergence properties compared to GDA is its update rule closely resembling PP, while the "error" of GDA as an approximation of PP is so large that it hinders convergence. The difference between the updates of EG and PP, in the Taylor expansion, is as small as $\mathcal{O}(\eta^3)$ per iteration, where $\eta$ is the stepsize. On the other hand, GDA and PP show a difference of $\mathcal{O}(\eta^2)$, and this greater "error" explains why GDA diverges while EG and PP converge. Of course, EG and PP are not the only two algorithms that converge in the monotone setting; let us recall the update rule of EG+ method (Diakonikolas et al., 2021), and Taylor-expand it as the following:

$$\boldsymbol{z}^+ := \boldsymbol{z} - \eta_2 \boldsymbol{F}(\boldsymbol{z} - \eta_1 \boldsymbol{F}\boldsymbol{z}) = \boldsymbol{z} - \eta_2 \boldsymbol{F}\boldsymbol{z} + \eta_1\eta_2 D\boldsymbol{F}(\boldsymbol{z})\boldsymbol{F}\boldsymbol{z} + O(\eta_1^2\eta_2). \tag{10}$$

EG+ is known to converge for unconstrained monotone problems if $\eta_1 \geq \eta_2$. When $\eta_1 = \eta_2$, it recovers EG and matches PP up to second-order terms.

Based on these observations, we now state our key principle for designing a convergent version of SEG: *second-order matching*. We would like to choose proper stepsizes, sampling scheme, and anchoring scheme so that our without-replacement SEG can *deterministically* match the update equation of a convergent algorithm (EG/PP or EG+) up to the $O(\eta^2)$ terms (i.e., *second-order* terms). We show that (a) this *second-order matching* can be achieved with *flip-flop anchoring*, but not solely by permutation-based sampling such as RR and flip-flop (without anchoring), and (b) second-order matching indeed grants convergence for monotone problems. In particular, we demonstrate that

1. SEG-RR suffers a poor approximation error of $\mathcal{O}(\eta^2)$ as an approximation of EG/EG+.
2. SEG-FF can match EG+ up to second-order terms, but it results in a choice of stepsizes ($\eta_2 = 2\eta_1$) that make EG+ diverge (Proposition 5.2).
3. SEG-FFA matches EG up to second-order terms to get an error of $\mathcal{O}(\eta^3)$ (Proposition 5.3), achieving convergence in monotone problems (Theorem 5.4) unlike SEG-RR or SEG-FF.

To this end, consider a general form of SEG that incorporates any arbitrary sampling scheme and anchoring. More precisely, in a certain "epoch," the components are chosen in the order of $\boldsymbol{T}_0, \boldsymbol{T}_1, \cdots, \boldsymbol{T}_{N-1}$, where $\boldsymbol{T}_i \in \{\boldsymbol{F}_1, \dots, \boldsymbol{F}_n\}$ for each $i$, and $N$ is some multiple of $n$ (e.g., $N = n$ for SEG-RR, $N = 2n$ for SEG-FF/SEG-FFA). Then, given $\alpha$, $\beta$, and $\theta$, we perform SEG updates

$$\boldsymbol{w}_i \leftarrow \boldsymbol{z}_i - \alpha \boldsymbol{T}_i \boldsymbol{z}_i, \qquad\qquad \boldsymbol{z}_{i+1} \leftarrow \boldsymbol{z}_i - \beta \boldsymbol{T}_i \boldsymbol{w}_i, \qquad\qquad (11)$$

for $i = 0, 1, \dots, N-1$, and then the anchoring step

$$\boldsymbol{z}^\sharp \leftarrow \frac{\boldsymbol{z}_N + \theta \boldsymbol{z}_0}{1 + \theta}, \qquad\qquad (12)$$

so that $\boldsymbol{z}^\sharp$ is used as the initial point of the next epoch. For this method, we show the following.

**Proposition 5.1.** *Suppose that Assumption 3 holds. For some $\epsilon_N = o\left((\alpha + \beta)^2\right)$, it holds that*

$$\boldsymbol{z}^\sharp = \boldsymbol{z}_0 - \frac{\beta}{1+\theta} \sum_{j=0}^{N-1} \boldsymbol{T}_j \boldsymbol{z}_0 + \frac{\alpha\beta}{1+\theta} \sum_{j=0}^{N-1} D\boldsymbol{T}_j(\boldsymbol{z}_0) \boldsymbol{T}_j \boldsymbol{z}_0 + \frac{\beta^2}{1+\theta} \sum_{0 \leq i < j \leq N-1} D\boldsymbol{T}_j(\boldsymbol{z}_0) \boldsymbol{T}_i \boldsymbol{z}_0 + \frac{\epsilon_N}{1+\theta}$$

$$(13)$$

See Appendix C.1 for the proof. Comparing (10) with (13), for them to match up to the second-order,

$$\frac{\eta_2}{n} \sum_{j=1}^{n} \boldsymbol{F}_i \boldsymbol{z}_0 = \frac{\beta}{1+\theta} \sum_{j=0}^{N-1} \boldsymbol{T}_j \boldsymbol{z}_0, \qquad\qquad (14)$$

$$\frac{\eta_1 \eta_2}{n^2} \sum_{j=1}^{n} D\boldsymbol{F}_j(\boldsymbol{z}_0) \boldsymbol{F}_j \boldsymbol{z}_0 + \frac{\eta_1 \eta_2}{n^2} \sum_{i \neq j} D\boldsymbol{F}_j(\boldsymbol{z}_0) \boldsymbol{F}_i \boldsymbol{z}_0$$

$$= \frac{\alpha\beta}{1+\theta} \sum_{j=0}^{N-1} D\boldsymbol{T}_j(\boldsymbol{z}_0) \boldsymbol{T}_j \boldsymbol{z}_0 + \frac{\beta^2}{1+\theta} \sum_{0 \leq i < j \leq N-1} D\boldsymbol{T}_j(\boldsymbol{z}_0) \boldsymbol{T}_i \boldsymbol{z}_0, \qquad (15)$$

must hold simultaneously. Clearly, without-replacement sampling will make (14) hold. However, it is easy to check that random reshuffling falls short of making (15) hold. This is because, if RR is used, then $\boldsymbol{T}_0, \boldsymbol{T}_1, \dots, \boldsymbol{T}_{n-1}$ is nothing but a reordering of $\boldsymbol{F}_1, \dots, \boldsymbol{F}_n$ into $\boldsymbol{F}_{\tau(1)}, \dots, \boldsymbol{F}_{\tau(n)}$, so the RHS of (15) can only contain terms $D\boldsymbol{F}_{\tau(j)}(\boldsymbol{z}_0) \boldsymbol{F}_{\tau(i)} \boldsymbol{z}_0$ with $i \leq j$.

This observation motivates the use of flip-flop sampling, because choosing $\boldsymbol{T}_i = \boldsymbol{T}_{2n-1-i}$ lets the RHS of (15) "cover" all required terms $D\boldsymbol{F}_j(\boldsymbol{z}_0) \boldsymbol{F}_i \boldsymbol{z}_0$. Indeed, flip-flop does resolve *this* issue, but there is still one complication remaining for SEG-FF, as described below.

**Proposition 5.2.** *Suppose we use flip-flop sampling. Without anchoring, or equivalently when $\theta = 0$, in order to make (14) and (15) hold, we have to choose $\beta = {}^{\eta_1}/n$ and $\alpha = {}^{\beta}/2$. However, such a choice leads to $\eta_2 = 2\eta_1$, which is the set of parameters that fails to make EG+ converge.*

The details can be found in Appendix C.2. This result shows that anchoring is necessary to get a stochastic method that achieves second-order matching to a *convergent* method. Now, say we actually use anchoring. Introducing the parameter $\theta$ increases the degree of freedom, and it opens up multiple possible stepsizes and parameters that make (14) and (15) hold. We show that $\alpha_k = {}^{\beta_k}/2$ and $\theta_k = 1$ are the simplest choices that, in fact, lead to the second-order matching to EG (instead of the more general EG+). A more precise statement is in Theorem 6.1.

**Proposition 5.3.** *Suppose that $\boldsymbol{F}$ is $\mu$-strongly monotone with $\mu \geq 0$, and Assumptions 3 and 4. Then, for $\beta_k = \eta$, $\alpha_k = \beta_k/2$, and $\theta_k = 1$,* SEG-FFA *becomes an approximation of EG with error at most $\mathcal{O}(\eta^3)$. In other words, we achieve*

$$\left\| \boldsymbol{z}_0 - \eta n \boldsymbol{F}(\boldsymbol{z}_0 - \eta n \boldsymbol{F} \boldsymbol{z}_0) - \boldsymbol{z}^\sharp \right\| = \mathcal{O}(\eta^3).$$

## 5.2 Convergence Analysis of SEG-FFA

As a result of the second-order matching, we obtain SEG-FFA, a stochastic method that has an error of $\mathcal{O}(\eta^3)$ as an approximation of EG. Achieving this order of magnitude for the approximation error turns out to be the key to the exact convergence to an optimum under the monotone setting, as demonstrated in the following result.

**Theorem 5.4.** *Suppose that $\boldsymbol{F}$ is monotone, Assumptions 2, 3, 4 hold, and we use* SEG-FFA. *By choosing the stepsizes sufficiently small and decaying at the rate of $\eta_k = \mathcal{O}(1/k^{1/3} \log k)$, the iterates generated by* SEG-FFA *achieves the bound of*

$$\min_{k=0,1,\ldots,K} \mathbb{E} \left\| \boldsymbol{F} \boldsymbol{z}_0^k \right\|^2 = \mathcal{O} \left( \frac{(\log K)^2}{K^{1/3}} \right). \tag{16}$$

*Proof.* For the full statement of the theorem and its proof, see Appendix F. $\qquad\square$

The reduced error also shows a gain in the rate of convergence under the strongly monotone setting. This aligns with the intuition that error hinders convergence, hence having a smaller error will be beneficial in the convergence of a method.

**Theorem 5.5.** *Suppose that $\boldsymbol{F}$ is $\mu$-strongly monotone with $\mu > 0$, Assumptions 3, 4 hold, and we are running* SEG-FFA *for $K$ epochs. For any given $\varepsilon \in (0, 2/3)$, let the stepsize be a constant equal to $\alpha_k = \beta_k = \frac{\omega}{nK^{1-\varepsilon}}$ for a sufficiently small constant $\omega$. Then we achieve the bound*

$$\mathbb{E} \left[ \left\| \boldsymbol{z}_0^K - \boldsymbol{z}^* \right\|^2 \right] \leq \exp \left( -\frac{\mu \omega K^\varepsilon}{2} \right) \left\| \boldsymbol{z}_0^0 - \boldsymbol{z}^* \right\|^2 + \mathcal{O} \left( \frac{1}{nK^{4-5\varepsilon}} \right). \tag{17}$$

*Proof.* See Section 6 for the proof sketch, and the appendices referred therein for the full proof. $\quad\square$

Notice the exponent of $4 - 5\varepsilon$ in the convergence rate, which is twice as large as the exponent 2 of SGDA-RR and $2 - 3\varepsilon$ of SEG-RR. In fact, this gain in the rate of convergence turns out to be fundamental. As we show in the following theorem, the theoretical lower bounds of convergence for SGDA-RR and SEG-RR with constant stepsize are both $\Omega(1/nK^3)$. This exhibits that there is a *provable gap* between those methods and SEG-FFA, which attains $\mathcal{O}(1/nK^{4-5\varepsilon})$.

**Theorem 5.6.** *Suppose $n \geq 2$. For both* SGDA-RR *with constant stepsize $\alpha_k = \alpha > 0$ and* SEG-RR *with constant stepsize $\alpha_k = \alpha > 0$, $\beta_k = \beta > 0$, there exists a $\mu$-strongly monotone minimax problem $f(\boldsymbol{z}) = \frac{1}{n} \sum_{i=1}^n f_i(\boldsymbol{z})$ with $\mu > 0$ such that regardless of stepsizes, we have*

$$\mathbb{E} \left[ \left\| \boldsymbol{z}_0^K - \boldsymbol{z}^* \right\|^2 \right] = \begin{cases} \Omega \left( \frac{\sigma^2}{L\mu nK} \right) & \text{if } K \leq L/\mu, \\ \Omega \left( \frac{L\sigma^2}{\mu^3 nK^3} \right) & \text{if } K > L/\mu. \end{cases}$$

*Proof.* The full statement and proof are presented in Appendix G.2. $\qquad\square$

In Appendix H, we present experiments to numerically validate our convergence and divergence analyses. We conducted experiments spanning monotone and strongly monotone cases. The experiments also include a comparison against existing convergent independent-sample SEG method by Hsieh et al. (2020), and an ablation study testing the effect of anchoring applied to uniform sampling and random reshuffling versions of SEG.

## 6 Proof Sketches

Let us briefly sketch the proofs of the convergence results we have presented in the previous sections. To this end, the statements shown in this section may be slightly informal. The precise statements and the details are deferred to the appendices.

## 6.1 WITHIN-EPOCH ERROR ANALYSIS FOR UPPER BOUNDS

Let us focus on the upper bounds of convergence. For the moment, we are only interested in the sequence $\{z_0^k\}_{k \geq 0}$, so for convenience let us drop the subscripts and write $z^k$ to denote $z_0^k$.

The cumulative updates made within an epoch can decomposed into a sum of an exact EG update and a within-epoch error term, which we denote by $r^k$, as

$$z^{k+1} = z^k - \eta_k n F(z^k - \eta_k n F z^k) + r^k. \tag{18}$$

The quality of the method will depend on how small the "noise" term $r^k$ is, as the noise will in general hinder the convergence. It turns out that, regardless of the method that is in use, the noise term can be bounded using a unified format, as follows. The proof can be found in Appendix D.2.

**Theorem 6.1.** *Suppose that $F$ is $\mu$-strongly monotone with $\mu \geq 0$, and Assumptions 3, 4 hold. For each of SEG-RR, SEG-FF, and SEG-FFA, there exists a choice of stepsizes that makes the following hold: For an exponent $a$ that depends on the method, there exist constants $C_1$, $D_1$, $V_1$, $C_2$, $D_2$, and $V_2$, all independent of $\eta_k$ and $n$, such that the error term $r^k$ satisfies a deterministic bound*

$$\left\| r^k \right\| \leq \eta_k^a n^a C_1 \left\| F z^k \right\| + \eta_k^a n^a D_1 \left\| F z^k \right\|^2 + \eta_k^a n^a V_1 \tag{19}$$

*and a bound that holds on expectation*

$$\mathbb{E}\left[ \left\| r \right\|^2 \middle| z^k \right] \leq \eta_k^{2a} n^{2a} C_2 \left\| F z^k \right\|^2 + \eta_k^{2a} n^{2a} D_2 \left\| F z^k \right\|^4 + \eta_k^{2a} n^{2a-1} V_2. \tag{20}$$

*Furthermore, the exponent is $a = 2$ for SEG-RR and SEG-FF, and $a = 3$ for SEG-FFA.*

In other words, SEG-FFA has an error that is order of magnitude smaller then other methods. Thus, it is now intuitively clear that SEG-FFA should have an advantage in the convergence.

## 6.2 META-ANALYSIS FOR CONVERGENCE BOUNDS

When $F$ is $\mu$-strongly monotone with $\mu > 0$, all SEG-RR, SEG-FF, and SEG-FFA do not diverge. In this case, it is possible to establish the following unified analysis of the methods.

**Theorem 6.2.** *Suppose that $F$ is $\mu$-strongly monotone with $\mu > 0$, Assumptions 3, 4 hold, and an optimization method whose within-epoch error satisfies (19) and (20) is run for $K$ epochs. Then, for any given $\varepsilon \in (0, 1 - 1/a)$, by choosing a constant stepsize $\eta_k = \frac{\omega}{nK^{1-\varepsilon}}$ for a sufficiently small constant $\omega$, we achieve the bound*

$$\mathbb{E}\left[ \left\| z^K - z^* \right\|^2 \right] \leq \exp\left( -\frac{\mu \omega K^\varepsilon}{2} \right) \left\| z^0 - z^* \right\|^2 + \mathcal{O}\left( \frac{1}{nK^{2a-2-(2a-1)\varepsilon}} \right). \tag{21}$$

For the precise statement of this result and its proof, see Appendix E. As the polynomial decay will dominate the exponential decay for large enough $K$, the bound we get is essentially $\mathcal{O}(1/K^{2a-2-(2a-1)\varepsilon})$. For SEG-RR and SEG-FF where $a = 2$, we get the upper bound $\mathcal{O}(1/K^{2-3\varepsilon})$, and for SEG-FFA where $a = 3$ we get the upper bound $\mathcal{O}(1/K^{4-5\varepsilon})$.

When $F$ is non-strongly monotone, as shown in Theorem 4.2, SEG-RR and SEG-FF diverge in the worst case. On the other hand, thanks to the second-order matching, the reduced error $\mathcal{O}(\eta^3)$ of SEG-FFA provides us convergence. See Appendix F for the precise statement on the convergence rate and the affiliated details.

## 7 CONCLUSION

We proposed SEG-FFA, a new stochastic variant of EG that uses flip-flop sampling and anchoring. While being a minimal modification from the vanilla SEG, SEG-FFA attains the "second-order matching" property to the deterministic EG, leading to a two-fold improved convergence. On one hand, SEG-FFA reaches an optimum in the monotone setting, unlike many baseline methods such as SEG-US, SEG-RR, and SEG-FF that diverge. Moreover, in the strongly monotone setting, SEG-FFA shows a faster convergence with a provable gap from the other methods.

An interesting future direction would be to extend our work to nonmonotone problems, further investigating the potentials of the second-order matching technique. Also, we note from our experiments that, while our work successfully shows the convergence of SEG-FFA, the convergence rate we derived may be suboptimal. Closing this gap between the observed convergence and the theoretical rate would be an appealing future work.

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

CONTENTS

## A   PSEUDOCODE OF THE ALGORITHMS

We present the pseudocode of the algorithms we consider in this paper in Algorithms 2, 3 and 4, with the pseudocode of the with-replacement stochastic methods in Algorithm 1.

---

**Algorithm 1:** SEG-US / SGDA-US

1 **Input**      : The number of components $n$; stepsize sequences $\{\alpha_t\}_{t\geq 0}$ and $\{\beta_t\}_{t\geq 0}$
2 **Initialize :** $z_0 \in \mathbb{R}^{d_1+d_2}$

3 **for** $t = 0, 1, \ldots$ **do**
4      sample $i(t)$ uniformly from $\{1, \ldots, n\}$
5      **if** *SGDA-US* **then**
6          $z_{t+1} \leftarrow z_t - \alpha_t F_{i(t)} z_t$
7      **end**
8      **else if** *SEG-US* **then**
9          $w_t \leftarrow z_t - \alpha_t F_{i(t)} z_t$
10          $z_{t+1} \leftarrow z_t - \beta_t F_{i(t)} w_t$
11      **end**
12 **end**

---

**Algorithm 2:** SEG-RR / SGDA-RR

1 **Input**      : The number of components $n$; stepsize sequences $\{\alpha_k\}_{k\geq 0}$ and $\{\beta_k\}_{k\geq 0}$
2 **Initialize :** $z_0^0 \in \mathbb{R}^{d_1+d_2}$

3 **for** $k = 0, 1, \ldots$ **do**
4      sample $\tau_k$ uniformly from $\mathcal{S}_n$
5      **for** $i = 0$ **to** $n - 1$ **do**
6          **if** *SGDA-RR* **then**
7              $z_{i+1}^k \leftarrow z_i^k - \alpha_k F_{\tau_k(i+1)} z_i^k$
8          **end**
9          **else if** *SEG-RR* **then**
10              $w_i^k \leftarrow z_i^k - \alpha_k F_{\tau_k(i+1)} z_i^k$
11              $z_{i+1}^k \leftarrow z_i^k - \beta_k F_{\tau_k(i+1)} w_i^k$
12          **end**
13      **end**
14      $z_0^{k+1} \leftarrow z_n^k$
15 **end**

---

**Algorithm 3:** SEG-FF

1 **Input**      : The number of components $n$; stepsize sequences $\{\alpha_k\}_{k\geq 0}$ and $\{\beta_k\}_{k\geq 0}$
2 **Initialize :** $z_0^0 \in \mathbb{R}^{d_1+d_2}$

3 **for** $k = 0, 1, \ldots$ **do**
4      sample $\tau_k$ uniformly from $\mathcal{S}_n$
5      **for** $i = 0$ **to** $n - 1$ **do**
6          $w_i^k \leftarrow z_i^k - \alpha_k F_{\tau_k(i+1)} z_i^k$
7          $z_{i+1}^k \leftarrow z_i^k - \beta_k F_{\tau_k(i+1)} w_i^k$
8      **end**
9      **for** $i = n$ **to** $2n - 1$ **do**
10          $w_i^k \leftarrow z_i^k - \alpha_k F_{\tau_k(2n-i)} z_i^k$
11          $z_{i+1}^k \leftarrow z_i^k - \beta_k F_{\tau_k(2n-i)} w_i^k$
12      **end**
13      $z_0^{k+1} \leftarrow z_{2n}^k$
14 **end**

---

---

**Algorithm 4: SEG-FFA**

---

1 **Input** : The number of components $n$; stepsize sequences $\{\eta_k\}_{k \geq 0}$

2 **Initialize :** $\boldsymbol{z}_0^0 \in \mathbb{R}^{d_1+d_2}$

3 **for** $k = 0, 1, \dots$ **do**

4      sample $\tau_k$ uniformly from $\mathcal{S}_n$

5      **for** $i = 0$ to $n - 1$ **do**

6          $\boldsymbol{w}_i^k \leftarrow \boldsymbol{z}_i^k - \frac{\eta_k}{2} \boldsymbol{F}_{\tau_k(i+1)} \boldsymbol{z}_i^k$

7          $\boldsymbol{z}_{i+1}^k \leftarrow \boldsymbol{z}_i^k - \eta_k \boldsymbol{F}_{\tau_k(i+1)} \boldsymbol{w}_i^k$

8      **end**

9      **for** $i = n$ to $2n - 1$ **do**

10         $\boldsymbol{w}_i^k \leftarrow \boldsymbol{z}_i^k - \frac{\eta_k}{2} \boldsymbol{F}_{\tau_k(2n-i)} \boldsymbol{z}_i^k$

11         $\boldsymbol{z}_{i+1}^k \leftarrow \boldsymbol{z}_i^k - \eta_k \boldsymbol{F}_{\tau_k(2n-i)} \boldsymbol{w}_i^k$

12      **end**

13      $\boldsymbol{z}_0^{k+1} \leftarrow \frac{\boldsymbol{z}_0^k + \boldsymbol{z}_{2n}^k}{2}$

14 **end**

---

## B    USEFUL LEMMATA

**Lemma B.1** (Polarization identity)**.** *For any two vectors $\boldsymbol{a}$ and $\boldsymbol{b}$, it holds that*

$$2 \langle \boldsymbol{a}, \boldsymbol{b} \rangle = \|\boldsymbol{a}\|^2 + \|\boldsymbol{b}\|^2 - \|\boldsymbol{a} - \boldsymbol{b}\|^2$$
$$= \|\boldsymbol{a} + \boldsymbol{b}\|^2 - \|\boldsymbol{a}\|^2 - \|\boldsymbol{b}\|^2 .$$

**Lemma B.2** (Weighted AM-GM inequality)**.** *For any $\gamma > 0$ and two vectors $\boldsymbol{a}$ and $\boldsymbol{b}$ in $\mathbb{R}^d$,*

$$2 |\langle \boldsymbol{a}, \boldsymbol{b} \rangle| \leq \gamma \|\boldsymbol{a}\|^2 + \frac{1}{\gamma} \|\boldsymbol{b}\|^2 .$$

*Proof.* Notice that

$$2 |\langle \boldsymbol{a}, \boldsymbol{b} \rangle| \leq 2 \left( |a_1 b_1| + \dots + |a_n b_n| \right)$$
$$\leq \left( \gamma a_1^2 + \frac{b_1^2}{\gamma} \right) + \dots + \left( \gamma a_d^2 + \frac{b_d^2}{\gamma} \right) = \gamma \|\boldsymbol{a}\|^2 + \frac{1}{\gamma} \|\boldsymbol{b}\|^2 . \qquad \square$$

**Lemma B.3** (Young's inequality)**.** *For any $\gamma > 0$ and two vectors $\boldsymbol{a}$ and $\boldsymbol{b}$,*

$$\|\boldsymbol{a} + \boldsymbol{b}\|^2 \leq (1 + \gamma) \|\boldsymbol{a}\|^2 + \left( 1 + \frac{1}{\gamma} \right) \|\boldsymbol{b}\|^2 .$$

*In particular, as a special case where $\gamma = 1$, it holds that*

$$\|\boldsymbol{a} + \boldsymbol{b}\|^2 \leq 2 \|\boldsymbol{a}\|^2 + 2 \|\boldsymbol{b}\|^2 . \tag{22}$$

**Lemma B.4.** *For any two vectors $\boldsymbol{a}$ and $\boldsymbol{b}$, it holds that*

$$\|\boldsymbol{a} - \boldsymbol{b}\|^2 \geq \frac{1}{2} \|\boldsymbol{a}\|^2 - \|\boldsymbol{b}\|^2 .$$

*Proof.* From (22) it follows that

$$\|\boldsymbol{a}\|^2 = \|(\boldsymbol{a} - \boldsymbol{b}) + \boldsymbol{b}\|^2 \leq 2 \|\boldsymbol{a} - \boldsymbol{b}\|^2 + 2 \|\boldsymbol{b}\|^2 .$$

Simply rearranging the terms gives us the result. $\qquad \square$

The following inequality is also frequently referred to as Young's inequality, so we will also do so.

**Lemma B.5** (Generalized Young's inequality). *For any positive scalars $p_1, \ldots, p_n$ such that $p_1 + \cdots + p_n = 1$ and vectors $\boldsymbol{a}_1, \ldots, \boldsymbol{a}_n$, it holds that*

$$\|p_1 \boldsymbol{a}_1 + \cdots + p_n \boldsymbol{a}_n\|^2 \le p_1 \|\boldsymbol{a}_1\|^2 + \cdots + p_n \|\boldsymbol{a}_n\|^2.$$

*In particular, setting $p_1 = \cdots = p_n = \frac{1}{n}$ and multiplying both sides by $n^2$ yields*

$$\|\boldsymbol{a}_1 + \cdots + \boldsymbol{a}_n\|^2 \le n \left( \|\boldsymbol{a}_1\|^2 + \cdots + \|\boldsymbol{a}_n\|^2 \right).$$

**Lemma B.6.** *Suppose that $\boldsymbol{F}$ is $M$-smooth. Then for any $\boldsymbol{z}$ and $\boldsymbol{w}$ it holds that*

$$\|\boldsymbol{F}\boldsymbol{w} - \boldsymbol{F}\boldsymbol{z} - D\boldsymbol{F}(\boldsymbol{z})(\boldsymbol{w} - \boldsymbol{z})\| \le \frac{M}{2} \|\boldsymbol{w} - \boldsymbol{z}\|^2.$$

*Proof.* The proof closely follows the arguments used for Lemma 1.2.4 in Nesterov (2018), by replacing the gradients therein by saddle gradients. The fundamental theorem of calculus with the $M$-smoothness of $\boldsymbol{F}$ gives us

$$\begin{aligned}
\|\boldsymbol{F}\boldsymbol{w} - \boldsymbol{F}\boldsymbol{z} - D\boldsymbol{F}(\boldsymbol{z})(\boldsymbol{w} - \boldsymbol{z})\| = \left\| \int_0^1 D\boldsymbol{F}(\boldsymbol{z} + t(\boldsymbol{w} - \boldsymbol{z})) \, \mathrm{d}t \, (\boldsymbol{w} - \boldsymbol{z}) - D\boldsymbol{F}(\boldsymbol{z})(\boldsymbol{w} - \boldsymbol{z}) \right\| \\
\le \int_0^1 \|D\boldsymbol{F}(\boldsymbol{z} + t(\boldsymbol{w} - \boldsymbol{z})) - D\boldsymbol{F}(\boldsymbol{z})\| \, \mathrm{d}t \, \|\boldsymbol{w} - \boldsymbol{z}\| \\
\le \int_0^1 Mt \|\boldsymbol{w} - \boldsymbol{z}\| \, \mathrm{d}t \, \|\boldsymbol{w} - \boldsymbol{z}\| \\
= \frac{M}{2} \|\boldsymbol{w} - \boldsymbol{z}\|^2. \qquad \square
\end{aligned}$$

**Lemma B.7.** *Let $\boldsymbol{F}$ be a $\mu$-strongly monotone operator. Let $\boldsymbol{z}^*$ be a point such that $\boldsymbol{F}\boldsymbol{z}^* = \boldsymbol{0}$, and let $\eta > 0$. Then, for any point $\boldsymbol{z}$ in the domain of $\boldsymbol{F}$ and $\boldsymbol{w} := \boldsymbol{z} - \eta \boldsymbol{F}\boldsymbol{z}$, it holds that*

$$\langle \boldsymbol{F}\boldsymbol{w}, \boldsymbol{w} - \boldsymbol{z}^* \rangle \ge \frac{\mu}{2} \|\boldsymbol{z} - \boldsymbol{z}^*\|^2 - \eta^2 \mu \|\boldsymbol{F}\boldsymbol{z}\|^2.$$

*Proof.* By the $\mu$-strong monotonicity of $\boldsymbol{F}$ and Lemma B.4 it holds that

$$\begin{aligned}
\langle \boldsymbol{F}\boldsymbol{w}, \boldsymbol{w} - \boldsymbol{z}^* \rangle &\ge \mu \|\boldsymbol{w} - \boldsymbol{z}^*\|^2 \\
&= \mu \|\boldsymbol{z} - \eta \boldsymbol{F}\boldsymbol{z} - \boldsymbol{z}^*\|^2 \\
&\ge \frac{\mu}{2} \|\boldsymbol{z} - \boldsymbol{z}^*\|^2 - \mu \|\eta \boldsymbol{F}\boldsymbol{z}\|^2
\end{aligned}$$

so we are done. $\qquad \square$

**Lemma B.8** (Nonexpansiveness of the EG operator). *Let $\boldsymbol{F}$ be a monotone $L$-Lipschitz operator. Let $\boldsymbol{z}^*$ be a point such that $\boldsymbol{F}\boldsymbol{z}^* = \boldsymbol{0}$, and $\boldsymbol{z}$ be any point in the domain of $\boldsymbol{F}$. Then, for any $\eta > 0$ it holds that*

$$\|\boldsymbol{z} - \eta \boldsymbol{F}(\boldsymbol{z} - \eta \boldsymbol{F}\boldsymbol{z}) - \boldsymbol{z}^*\|^2 \le \|\boldsymbol{z} - \boldsymbol{z}^*\|^2 - \eta^2(1 - \eta^2 L^2) \|\boldsymbol{F}\boldsymbol{z}\|^2.$$

*Proof.* This lemma is exactly from Gorbunov et al. (2022b); see Section D.7 therein. $\qquad \square$

The following lemma generalizes Lemma 3.2 in Gorbunov et al. (2022b) shown for monotone $\boldsymbol{F}$ to $\mu$-strongly monotone $\boldsymbol{F}$.

**Lemma B.9.** *Let $\boldsymbol{F}$ be a $\mu$-strongly monotone $L$-Lipschitz operator, and let $\boldsymbol{z}$ be any point in the domain of $\boldsymbol{F}$. Then for any $0 < \eta < \frac{1}{L\sqrt{2}}$, it holds that*

$$\|\boldsymbol{F}(\boldsymbol{z} - \eta \boldsymbol{F}(\boldsymbol{z} - \eta \boldsymbol{F}\boldsymbol{z}))\|^2 \le \left(1 - \frac{2\eta\mu}{5}\right) \|\boldsymbol{F}\boldsymbol{z}\|^2.$$

*Proof.* For convenience, let us define $\boldsymbol{w} := \boldsymbol{z} - \eta \boldsymbol{F}\boldsymbol{z}$ and $\boldsymbol{z}^+ := \boldsymbol{z} - \eta \boldsymbol{F}(\boldsymbol{z} - \eta \boldsymbol{F}\boldsymbol{z}) = \boldsymbol{z} - \eta \boldsymbol{F}\boldsymbol{w}$. Because $\boldsymbol{F}$ is $\mu$-strongly monotone, we have

$$\mu \left\| \boldsymbol{z}^+ - \boldsymbol{z} \right\|^2 \leq \left\langle \boldsymbol{F}\boldsymbol{z}^+ - \boldsymbol{F}\boldsymbol{z}, \boldsymbol{z}^+ - \boldsymbol{z} \right\rangle \tag{23}$$
$$= \eta \left\langle \boldsymbol{F}\boldsymbol{z} - \boldsymbol{F}\boldsymbol{z}^+, \boldsymbol{F}\boldsymbol{w} \right\rangle.$$

Also from the $\mu$-strong monotonicity of $\boldsymbol{F}$ we get

$$\mu \left\| \boldsymbol{w} - \boldsymbol{z}^+ \right\|^2 \leq \left\langle \boldsymbol{F}\boldsymbol{w} - \boldsymbol{F}\boldsymbol{z}^+, \boldsymbol{w} - \boldsymbol{z}^+ \right\rangle \tag{24}$$
$$= \eta \left\langle \boldsymbol{F}\boldsymbol{w} - \boldsymbol{F}\boldsymbol{z}^+, \boldsymbol{F}\boldsymbol{w} - \boldsymbol{F}\boldsymbol{z} \right\rangle.$$

Meanwhile, from the $L$-Lipschitzness of $\boldsymbol{F}$ we have

$$\left\| \boldsymbol{F}\boldsymbol{w} - \boldsymbol{F}\boldsymbol{z}^+ \right\|^2 \leq \eta^2 L^2 \left\| \boldsymbol{F}\boldsymbol{w} - \boldsymbol{F}\boldsymbol{z} \right\|^2. \tag{25}$$

Summing up the inequalities (23), (24), (25) with weights $2/\eta$, $1/2\eta$, and $3/2$ respectively, we obtain

$$\frac{\mu}{\eta} \left( 2 \left\| \boldsymbol{z}^+ - \boldsymbol{z} \right\|^2 + \frac{1}{2} \left\| \boldsymbol{w} - \boldsymbol{z}^+ \right\|^2 \right) + \frac{3}{2} \left\| \boldsymbol{F}\boldsymbol{w} - \boldsymbol{F}\boldsymbol{z}^+ \right\|^2$$
$$\leq 2 \left\langle \boldsymbol{F}\boldsymbol{z} - \boldsymbol{F}\boldsymbol{z}^+, \boldsymbol{F}\boldsymbol{w} \right\rangle + \frac{1}{2} \left\langle \boldsymbol{F}\boldsymbol{w} - \boldsymbol{F}\boldsymbol{z}^+, \boldsymbol{F}\boldsymbol{w} - \boldsymbol{F}\boldsymbol{z} \right\rangle + \frac{3\eta^2 L^2}{2} \left\| \boldsymbol{F}\boldsymbol{w} - \boldsymbol{F}\boldsymbol{z} \right\|^2.$$

From this inequality, we can exactly follow the arguments used in the proof of Lemma 3.2 in Gorbunov et al. (2022b) to derive that

$$\frac{\mu}{\eta} \left( 2 \left\| \boldsymbol{z}^+ - \boldsymbol{z} \right\|^2 + \frac{1}{2} \left\| \boldsymbol{w} - \boldsymbol{z}^+ \right\|^2 \right) + \left\| \boldsymbol{F}\boldsymbol{z}^+ \right\|^2 \leq \left\| \boldsymbol{F}\boldsymbol{z} \right\|^2. \tag{26}$$

On the other hand, Young's inequality (Lemma B.3) tells us that

$$\eta^2 \left\| \boldsymbol{F}\boldsymbol{z} \right\|^2 = \left\| \boldsymbol{w} - \boldsymbol{z} \right\|^2 \leq \left( 1 + \frac{1}{4} \right) \left\| \boldsymbol{w} - \boldsymbol{z}^+ \right\|^2 + (1 + 4) \left\| \boldsymbol{z}^+ - \boldsymbol{z} \right\|^2.$$

This, combined with (26), implies that

$$\frac{2\eta\mu}{5} \left\| \boldsymbol{F}\boldsymbol{z} \right\|^2 + \left\| \boldsymbol{F}\boldsymbol{z}^+ \right\|^2 \leq \left\| \boldsymbol{F}\boldsymbol{z} \right\|^2.$$

It remains to simply rearrange the terms. $\qquad \square$

**Lemma B.10.** *Let* $\{a_k\}_{k \geq 0}$, $\{b_k\}_{k \geq 0}$, $\{c_k\}_{k \geq 0}$, *and* $\{d_k\}_{k \geq 0}$ *be sequences of nonnegative numbers satisfying the recurrence relation*

$$b_k \leq (1 + a_k)d_k - d_{k+1} + c_k \qquad \forall k \geq 0.$$

*Then for any* $k \geq 0$ *it holds that*

$$d_{k+1} + \sum_{j=0}^{k} b_j \leq \left( \prod_{j=0}^{k} (1 + a_j) \right) \left( d_0 + \sum_{j=0}^{k} c_j \right).$$

*Proof.* Because $a_k \geq 0$, it suffices to show that

$$\sum_{j=0}^{k} (b_j - c_j) \prod_{i=j+1}^{k} (1 + a_i) \leq -d_{k+1} + d_0 \prod_{j=0}^{k} (1 + a_j), \tag{27}$$

as this implies

$$\sum_{j=0}^{k} b_j \leq \sum_{j=0}^{k} b_j \prod_{i=j+1}^{k} (1 + a_i)$$
$$\leq \left( \sum_{j=0}^{k} c_j \prod_{i=j+1}^{k} (1 + a_i) \right) - d_{k+1} + d_0 \prod_{j=0}^{k} (1 + a_j)$$
$$\leq -d_{k+1} + \left( d_0 + \sum_{j=0}^{k} c_k \right) \prod_{j=0}^{k} (1 + a_j).$$

So, we show that (27) holds, by induction on $k$. For the base case $k = 0$, the recurrence relation tells us that

$$b_0 - c_0 \leq (1 + a_0)d_0 - d_1$$

which is exactly (27) when $k = 0$. Now suppose that (27) holds for some $k \geq 0$. Using the induction hypothesis and the recurrence relation we get

$$\sum_{j=0}^{k+1}(b_j - c_j)\prod_{i=j+1}^{k+1}(1 + a_i) = b_{k+1} - c_{k+1} + (1 + a_{k+1})\left(\sum_{j=0}^{k}(b_j - c_j)\prod_{i=j+1}^{k}(1 + a_i)\right)$$

$$\leq b_{k+1} - c_{k+1} - (1 + a_{k+1})d_{k+1} + d_0\prod_{j=0}^{k+1}(1 + a_j)$$

$$\leq -d_{k+2} + d_0\prod_{j=0}^{k+1}(1 + a_j).$$

This shows that (27) holds also for $k + 1$, and we are done. $\qquad\square$

The subsequent lemma is technical, but it can be derived from elementary calculus.

**Lemma B.11.** *For any $K \geq 1$,*

$$\sum_{k=2}^{K+2}\frac{1}{k^{2/3}(\log k)^2} \geq \frac{(K+3)^{1/3}}{(\log(K+3))^2}.$$

*Proof.* Consider integrating the function $x \mapsto \frac{1}{x^{2/3}(\log x)^2}$ over the interval $[2, K+3]$, where the function is decreasing. An upper Riemann sum is an upper bound for an integral, so we have

$$\sum_{k=2}^{K+2}\frac{1}{k^{2/3}(\log k)^2} \geq \int_2^{K+3}\frac{1}{x^{2/3}(\log x)^2}\,\mathrm{d}x. \tag{28}$$

Now consider a function $g : [1, \infty) \to \mathbb{R}$, defined as

$$g(y) := \int_2^{y+3}\frac{1}{x^{2/3}(\log x)^2}\,\mathrm{d}x - \frac{(y+3)^{1/3}}{(\log(y+3))^2}.$$

Differentiating, we get

$$g'(y) = \frac{2}{(y+3)^{2/3}(\log(y+3))^3} + \frac{2}{3(y+3)^{2/3}(\log(y+3))^2} > 0$$

whenever $y \geq 1$. That is, $g$ is increasing on $y \geq 1$.

It is easy to verify that $g(1) \geq 0$; one way to do so is, for $\mathrm{Ei}(\,\cdot\,)$ denoting the *exponential integral* function (Abramowitz & Stegun, 1965, Chapter 5), to use the indefinite integral

$$\int\frac{1}{x^{2/3}(\log x)^2}\,\mathrm{d}x = \frac{1}{3}\,\mathrm{Ei}\left(\frac{\log x}{3}\right) - \frac{\sqrt[3]{x}}{\log x} \tag{29}$$

to numerically evaluate $g(1)$. Since $g$ is increasing, we have $g(y) \geq g(1) \geq 0$ for all $y \geq 1$. This, with (28), implies that

$$\sum_{k=2}^{K+2}\frac{1}{k^{2/3}(\log k)^2} \geq \int_2^{K+3}\frac{1}{x^{2/3}(\log x)^2}\,\mathrm{d}x \geq \frac{(K+3)^{1/3}}{(\log(K+3))^2}$$

holds whenever $K \geq 1$, which is exactly the claimed. $\qquad\square$

# C  MISSING PROOFS FOR SECTION 5

## C.1  UNRAVELLING THE RECURRENCE OF THE GENERALIZED SEG IN (11) AND (13)

In Section 5.1, we considered the method where, in an epoch, the iterates are generated following the recurrence

$$
\begin{aligned}
\boldsymbol{w}_i &= \boldsymbol{z}_i - \alpha \boldsymbol{T}_i \boldsymbol{z}_i \\
\boldsymbol{z}_{i+1} &= \boldsymbol{z}_i - \beta \boldsymbol{T}_i \boldsymbol{w}_i
\end{aligned}
\tag{30}
$$

for $i = 0, 1, \ldots, N-1$, where each $\boldsymbol{T}_i$ are sampled from the set $\{\boldsymbol{F}_1, \ldots, \boldsymbol{F}_n\}$, and an additional anchoring step

$$
\boldsymbol{z}^\sharp := \frac{\boldsymbol{z}_N + \theta \boldsymbol{z}_0}{1 + \theta}
\tag{31}
$$

is performed so that $\boldsymbol{z}^\sharp$ is used as the initial point of the next epoch. In this section we would like to prove the following two statements regarding this update rule.

**Proposition C.1** (Proposition 5.1). *It holds that*

$$
\boldsymbol{z}^\sharp = \boldsymbol{z}_0 - \frac{\beta}{1+\theta} \sum_{j=0}^{N-1} \boldsymbol{T}_j \boldsymbol{z}_0 + \frac{\alpha\beta}{1+\theta} \sum_{j=0}^{N-1} D\boldsymbol{T}_j(\boldsymbol{z}_0)\boldsymbol{T}_j \boldsymbol{z}_0 + \frac{\beta^2}{1+\theta} \sum_{0 \le i < j \le N-1} D\boldsymbol{T}_j(\boldsymbol{z}_0)\boldsymbol{T}_i \boldsymbol{z}_0 + \frac{\boldsymbol{\epsilon}_N}{1+\theta}
\tag{32}
$$

*for some $\boldsymbol{\epsilon}_N = o\left((\alpha + \beta)^2\right)$.*

*Proof.* Equation (32) immediately follows from Proposition C.2, with (34) giving us the precise definition of $\boldsymbol{\epsilon}_N$. To show that $\boldsymbol{\epsilon}_N = o\left((\alpha + \beta)^2\right)$, we begin with noting that both $\|\boldsymbol{z}_j - \boldsymbol{z}_0\|$ and $\|\boldsymbol{w}_j - \boldsymbol{z}_0\|$ are of $\mathcal{O}(\alpha + \beta)$, because both $\boldsymbol{z}_j$ and $\boldsymbol{w}_j$ are obtained from $\boldsymbol{z}_0$ by performing at most $j$ updates following (30). Thus, the first term in the right hand side of (34) is of $\mathcal{O}(\beta(\alpha + \beta)^2)$ by Lemma B.6, and the remaining terms are of $\mathcal{O}((\alpha + \beta)^3)$ by the $L$-smoothness of the operators $\boldsymbol{F}_1, \ldots, \boldsymbol{F}_n$. $\qquad\square$

**Proposition C.2.** *For any $i = 0, 1, \ldots, N$, it holds that*

$$
\boldsymbol{z}_i = \boldsymbol{z}_0 - \beta \sum_{j=0}^{i-1} \boldsymbol{T}_j \boldsymbol{z}_0 + \alpha\beta \sum_{j=0}^{i-1} D\boldsymbol{T}_j(\boldsymbol{z}_0)\boldsymbol{T}_j \boldsymbol{z}_0 + \beta^2 \sum_{0 \le k < j \le i-1} D\boldsymbol{T}_j(\boldsymbol{z}_0)\boldsymbol{T}_k \boldsymbol{z}_0 + \boldsymbol{\epsilon}_i
\tag{33}
$$

*where we denote*

$$
\begin{aligned}
\boldsymbol{\epsilon}_i := &- \beta \sum_{j=0}^{i-1} \Big( \boldsymbol{T}_j \boldsymbol{w}_j - \boldsymbol{T}_j \boldsymbol{z}_0 - D\boldsymbol{T}_j(\boldsymbol{z}_0)(\boldsymbol{w}_j - \boldsymbol{z}_0) \Big) \\
&+ \alpha\beta \sum_{j=0}^{i-1} D\boldsymbol{T}_j(\boldsymbol{z}_0)(\boldsymbol{T}_j \boldsymbol{z}_j - \boldsymbol{T}_j \boldsymbol{z}_0) + \beta^2 \sum_{j=0}^{i-1} D\boldsymbol{T}_j(\boldsymbol{z}_0) \sum_{k=0}^{j-1} (\boldsymbol{T}_k \boldsymbol{w}_k - \boldsymbol{T}_k \boldsymbol{z}_0).
\end{aligned}
\tag{34}
$$

*Proof.* We use induction on $i$. There is nothing to show for the base case $i = 0$. Now, suppose that (33) and (34) hold for some $i < N$, and write

$$
\begin{aligned}
\boldsymbol{z}_{i+1} &= \boldsymbol{z}_i - \beta \boldsymbol{T}_i \boldsymbol{w}_i \\
&= \boldsymbol{z}_i - \beta \boldsymbol{T}_i \boldsymbol{z}_0 - \beta D\boldsymbol{T}_i(\boldsymbol{z}_0)(\boldsymbol{w}_i - \boldsymbol{z}_0) - \beta \Big( \boldsymbol{T}_i \boldsymbol{w}_i - \boldsymbol{T}_i \boldsymbol{z}_0 - D\boldsymbol{T}_i(\boldsymbol{z}_0)(\boldsymbol{w}_i - \boldsymbol{z}_0) \Big).
\end{aligned}
$$

Here, notice that by the update rule we have

$$
\begin{aligned}
\boldsymbol{w}_i &= \boldsymbol{z}_i - \alpha \boldsymbol{T}_i \boldsymbol{z}_i \\
&= \boldsymbol{z}_0 - \beta \sum_{j=0}^{i-1} \boldsymbol{T}_j \boldsymbol{w}_j - \alpha \boldsymbol{T}_i \boldsymbol{z}_i.
\end{aligned}
$$

Using this identity and the induction hypothesis we get

$$
\boldsymbol{z}_{i+1} = \boldsymbol{z}_0 - \beta \sum_{j=0}^{i-1} \boldsymbol{T}_j \boldsymbol{z}_0 + \alpha\beta \sum_{j=0}^{i-1} D\boldsymbol{T}_j(\boldsymbol{z}_0)\boldsymbol{T}_j \boldsymbol{z}_0 + \beta^2 \sum_{0 \le k < j \le i-1} D\boldsymbol{T}_j(\boldsymbol{z}_0)\boldsymbol{T}_k \boldsymbol{z}_0 + \boldsymbol{\epsilon}_i
$$

$$- \beta \boldsymbol{T}_i \boldsymbol{z}_0 - \beta D \boldsymbol{T}_i(\boldsymbol{z}_0) \left( -\beta \sum_{j=0}^{i-1} \boldsymbol{T}_j \boldsymbol{w}_j - \alpha \boldsymbol{T}_i \boldsymbol{z}_i \right)$$

$$- \beta \Big( \boldsymbol{T}_i \boldsymbol{w}_i - \boldsymbol{T}_i \boldsymbol{z}_0 - D \boldsymbol{T}_i(\boldsymbol{z}_0)(\boldsymbol{w}_i - \boldsymbol{z}_0) \Big)$$

$$= \boldsymbol{z}_0 - \beta \sum_{j=0}^{i-1} \boldsymbol{T}_j \boldsymbol{z}_0 + \alpha\beta \sum_{j=0}^{i-1} D\boldsymbol{T}_j(\boldsymbol{z}_0)\boldsymbol{T}_j \boldsymbol{z}_0 + \beta^2 \sum_{0 \le k < j \le i-1} D\boldsymbol{T}_j(\boldsymbol{z}_0)\boldsymbol{T}_k \boldsymbol{z}_0 + \boldsymbol{\epsilon}_i$$

$$- \beta \boldsymbol{T}_i \boldsymbol{z}_0 + \beta^2 D\boldsymbol{T}_i(\boldsymbol{z}_0) \sum_{j=0}^{i-1} \boldsymbol{T}_j \boldsymbol{z}_0 + \beta^2 D\boldsymbol{T}_i(\boldsymbol{z}_0) \sum_{j=0}^{i-1} (\boldsymbol{T}_j \boldsymbol{w}_j - \boldsymbol{T}_j \boldsymbol{z}_0)$$

$$+ \alpha\beta D\boldsymbol{T}_i(\boldsymbol{z}_0)(\boldsymbol{T}_i \boldsymbol{z}_i - \boldsymbol{T}_i \boldsymbol{z}_0) + \alpha\beta D\boldsymbol{T}_i(\boldsymbol{z}_0)\boldsymbol{T}_i \boldsymbol{z}_0$$

$$- \beta \Big( \boldsymbol{T}_i \boldsymbol{w}_i - \boldsymbol{T}_i \boldsymbol{z}_0 - D\boldsymbol{T}_i(\boldsymbol{z}_0)(\boldsymbol{w}_i - \boldsymbol{z}_0) \Big)$$

$$= \boldsymbol{z}_0 - \beta \sum_{j=0}^{i} \boldsymbol{T}_j \boldsymbol{z}_0 + \alpha\beta \sum_{j=0}^{i} D\boldsymbol{T}_j(\boldsymbol{z}_0)\boldsymbol{T}_j \boldsymbol{z}_0 + \beta^2 \sum_{0 \le k < j \le i} D\boldsymbol{T}_j(\boldsymbol{z}_0)\boldsymbol{T}_k \boldsymbol{z}_0 + \boldsymbol{\epsilon}_i$$

$$+ \beta^2 D\boldsymbol{T}_i(\boldsymbol{z}_0) \sum_{j=0}^{i-1} (\boldsymbol{T}_j \boldsymbol{w}_j - \boldsymbol{T}_j \boldsymbol{z}_0) + \alpha\beta D\boldsymbol{T}_i(\boldsymbol{z}_0)(\boldsymbol{T}_i \boldsymbol{z}_i - \boldsymbol{T}_i \boldsymbol{z}_0)$$

$$- \beta \Big( \boldsymbol{T}_i \boldsymbol{w}_i - \boldsymbol{T}_i \boldsymbol{z}_0 - D\boldsymbol{T}_i(\boldsymbol{z}_0)(\boldsymbol{w}_i - \boldsymbol{z}_0) \Big)$$

$$= \boldsymbol{z}_0 - \beta \sum_{j=0}^{i} \boldsymbol{T}_j \boldsymbol{z}_0 + \alpha\beta \sum_{j=0}^{i} D\boldsymbol{T}_j(\boldsymbol{z}_0)\boldsymbol{T}_j \boldsymbol{z}_0 + \beta^2 \sum_{0 \le k < j \le i} D\boldsymbol{T}_j(\boldsymbol{z}_0)\boldsymbol{T}_k \boldsymbol{z}_0 + \boldsymbol{\epsilon}_{i+1}$$

which asserts that (33) also holds for $i + 1$. $\qquad\square$

## C.2 Insufficiency of Only Using Flip-Flop Sampling

Here we prove the following.

**Proposition C.3** (Proposition 5.2). *Suppose we use flip-flop sampling. Without anchoring, or equivalently when $\theta = 0$, in order to make (14) and (15) hold, we have to choose $\beta = \eta_1/n$ and $\alpha = \beta/2$. However, such a choice leads to $\eta_2 = 2\eta_1$, which is the set of parameters that fails to make EG+ converge.*

*Proof.* Suppose that we have already established the upcoming Lemma C.4. Then, according to Lemma C.4, for (15) to hold with $\theta = 0$, the following system of equations should be satisfied:

$$\begin{cases} \eta_1 \eta_2 = 2n^2 \beta^2, \\ \eta_1 \eta_2 = n^2 (2\alpha\beta + \beta^2), \\ \eta_2 = 2n\beta. \end{cases}$$

Solving this system of equations, we get $\eta_1 = n\beta$, $\eta_2 = 2n\beta$, and $\alpha = \beta/2$.

For the latter part of the statement on the divergence of EG+ with $\eta_2 = 2\eta_1$, consider the $(1+1)$-dimensional bilinear problem

$$\min_x \max_y \; xy$$

whose unique optimum is $\boldsymbol{z}^* = (0, 0)$. A simple computation shows that

$$\boldsymbol{F}\boldsymbol{z} = \begin{bmatrix} 0 & 1 \\ -1 & 0 \end{bmatrix} \boldsymbol{z}.$$

Consequently, for any $\eta > 0$, the update rule of EG+ with $\eta_1 = \eta$ and $\eta_2 = 2\eta$ amounts to

$$\boldsymbol{z}^+ = \boldsymbol{z} - 2\eta \boldsymbol{F}(\boldsymbol{z} - \eta \boldsymbol{F}\boldsymbol{z}) = \begin{bmatrix} 1 - 2\eta^2 & -2\eta \\ 2\eta & 1 - 2\eta^2 \end{bmatrix} \boldsymbol{z}.$$

It follows that

$$
\begin{aligned}
\left\|\boldsymbol{z}^+ - \boldsymbol{z}^*\right\|^2 &= \left\|\begin{bmatrix} 1 - 2\eta^2 & -2\eta \\ 2\eta & 1 - 2\eta^2 \end{bmatrix} \begin{bmatrix} x \\ y \end{bmatrix}\right\|^2 \\
&= \left((1 - 2\eta^2)x - 2\eta y\right)^2 + \left(2\eta x + (1 - 2\eta^2)y\right)^2 \\
&= (1 + 4\eta^4)(x^2 + y^2) \\
&= (1 + 4\eta^4)\left\|\boldsymbol{z} - \boldsymbol{z}^*\right\|^2.
\end{aligned}
$$

Therefore, the distance from the optimal solution strictly increases every iterate. $\quad\square$

It remains to actually prove Lemma C.4.

**Lemma C.4.** *When flip-flop sampling is used, it holds that*

$$
\frac{\alpha\beta}{1 + \theta} \sum_{j=0}^{N-1} D\boldsymbol{T}_j(\boldsymbol{z}_0)\boldsymbol{T}_j\boldsymbol{z}_0 + \frac{\beta^2}{1 + \theta} \sum_{0 \le i < j \le N-1} D\boldsymbol{T}_j(\boldsymbol{z}_0)\boldsymbol{T}_i\boldsymbol{z}_0
$$

$$
= \frac{2\alpha\beta + \beta^2}{1 + \theta} \sum_{j=1}^{n} D\boldsymbol{F}_j(\boldsymbol{z}_0)\boldsymbol{F}_j\boldsymbol{z}_0 + \frac{2\beta^2}{1 + \theta} \sum_{i \ne j} D\boldsymbol{F}_j(\boldsymbol{z}_0)\boldsymbol{F}_i\boldsymbol{z}_0.
$$

*Proof.* As we are using flip-flop sampling, we have $N = 2n$, and it is clear that

$$
\sum_{j=0}^{N-1} D\boldsymbol{T}_j(\boldsymbol{z}_0)\boldsymbol{T}_j\boldsymbol{z}_0 = 2 \sum_{j=1}^{n} D\boldsymbol{F}_j(\boldsymbol{z}_0)\boldsymbol{F}_j\boldsymbol{z}_0.
$$

For the second term, as $\boldsymbol{T}_i = \boldsymbol{T}_{2n-1-i}$, we have

$$
\begin{aligned}
\sum_{0 \le i < j \le 2n-1} D\boldsymbol{T}_j(\boldsymbol{z}_0)\boldsymbol{T}_i\boldsymbol{z}_0 &= \sum_{0 \le i < j \le n-1} D\boldsymbol{T}_j(\boldsymbol{z}_0)\boldsymbol{T}_i\boldsymbol{z}_0 + \sum_{n \le i < j \le 2n-1} D\boldsymbol{T}_j(\boldsymbol{z}_0)\boldsymbol{T}_i\boldsymbol{z}_0 \\
&\quad + \sum_{i=0}^{n-1}\sum_{j=n}^{2n-2-i} D\boldsymbol{T}_j(\boldsymbol{z}_0)\boldsymbol{T}_i\boldsymbol{z}_0 + \sum_{i=0}^{n-1}\sum_{j=2n-i}^{2n-1} D\boldsymbol{T}_j(\boldsymbol{z}_0)\boldsymbol{T}_i\boldsymbol{z}_0 \\
&\quad + \sum_{i=0}^{n-1} D\boldsymbol{T}_{2n-1-i}(\boldsymbol{z}_0)\boldsymbol{T}_i\boldsymbol{z}_0 \\
&= \sum_{0 \le i < j \le n-1} D\boldsymbol{T}_j(\boldsymbol{z}_0)\boldsymbol{T}_i\boldsymbol{z}_0 + \sum_{0 \le j < i \le n-1} D\boldsymbol{T}_j(\boldsymbol{z}_0)\boldsymbol{T}_i\boldsymbol{z}_0 \\
&\quad + \sum_{i=0}^{n-1}\sum_{j=i+1}^{n-1} D\boldsymbol{T}_j(\boldsymbol{z}_0)\boldsymbol{T}_i\boldsymbol{z}_0 + \sum_{i=0}^{n-1}\sum_{j=0}^{i-1} D\boldsymbol{T}_j(\boldsymbol{z}_0)\boldsymbol{T}_i\boldsymbol{z}_0 \\
&\quad + \sum_{i=0}^{n-1} D\boldsymbol{T}_i(\boldsymbol{z}_0)\boldsymbol{T}_i\boldsymbol{z}_0 \\
&= 2 \sum_{0 \le i < j \le n-1} D\boldsymbol{T}_j(\boldsymbol{z}_0)\boldsymbol{T}_i\boldsymbol{z}_0 + 2 \sum_{0 \le j < i \le n-1} D\boldsymbol{T}_j(\boldsymbol{z}_0)\boldsymbol{T}_i\boldsymbol{z}_0 \\
&\quad + \sum_{i=0}^{n-1} D\boldsymbol{T}_i(\boldsymbol{z}_0)\boldsymbol{T}_i\boldsymbol{z}_0.
\end{aligned}
$$

The claimed identity can be obtained by taking the weighted sum of the two results. $\quad\square$

## D  MISSING PROOFS FOR SECTION 6.1

In this section, although it is an abuse of notation, for convenience we will write $\boldsymbol{F}_i$ to denote the saddle gradient of the component function chosen in the $i^{\text{th}}$ iteration. More precisely, for indices

$i = 0, 1, \ldots, n - 1$ we denote $\boldsymbol{F}_{\tau(i+1)}$ by $\boldsymbol{F}_i$. Similarly, in cases of considering SEG-FF or SEG-FFA, for $i \geq n$ we denote $\boldsymbol{F}_{\tau(2n-i)}$ by $\boldsymbol{F}_i$. Also, we omit the superscripts and subscripts denoting the epoch number $k$ unless strictly necessary, as all the iterates that we consider will be from the same epoch.

We consider the iterates generated by the update rule

$$
\begin{aligned}
\boldsymbol{w}_i &= \boldsymbol{z}_i - \xi\eta\boldsymbol{F}\boldsymbol{z}_i, \\
\boldsymbol{z}_{i+1} &= \boldsymbol{z}_i - \eta\boldsymbol{F}\boldsymbol{w}_i.
\end{aligned}
\tag{35}
$$

Note that $\xi = 1/2$ for SEG-FFA, and $\xi = 1$ for SEG-RR and SEG-FF.

### D.1 AUXILIARY LEMMATA

For $j = 1, \ldots, 2n$ we define

$$
\boldsymbol{g}_j := \sum_{i=0}^{j-1} \boldsymbol{F}_i\boldsymbol{z}_0,
\tag{36}
$$

$$
\delta_j := \|\boldsymbol{g}_j - j\boldsymbol{F}\boldsymbol{z}_0\|,
\tag{37}
$$

$$
\Sigma_j := \sum_{i=1}^{j} \delta_i,
\tag{38}
$$

$$
\Psi_j := \sum_{i=1}^{j} \delta_i^2.
\tag{39}
$$

We set $\Sigma_0 = \Psi_0 = 0$, as they are empty sums. Notice that $\delta_j$ is a random variable that depends on the permutation $\tau$.

Meanwhile, by triangle inequality it is immediate that

$$
\|\boldsymbol{g}_j\| \leq j\|\boldsymbol{F}\boldsymbol{z}_0\| + \delta_j,
$$

and by Young's inequality it holds that

$$
\|\boldsymbol{g}_j\|^2 \leq 2j^2\|\boldsymbol{F}\boldsymbol{z}_0\|^2 + 2\delta_j^2.
$$

**Lemma D.1.** *For any index $i$, it holds that*

$$
\|\boldsymbol{z}_i - \boldsymbol{z}_0\| \leq \eta(1 + \xi\eta L)\|\boldsymbol{g}_i\|
$$

$$
+ \eta^2 L \left(2\xi + 2\xi\eta L + \xi^2\eta^2 L^2\right) \sum_{\ell=0}^{i-2} \left(1 + \eta L + \xi\eta^2 L^2\right)^{i-\ell-2} \|\boldsymbol{g}_{\ell+1}\|,
\tag{40}
$$

$$
\|\boldsymbol{w}_i - \boldsymbol{z}_0\| \leq \xi\eta\|\boldsymbol{g}_{i+1}\| + \xi\eta\left((1 - \xi^{-1}) + 2\eta L + \xi\eta^2 L^2\right)\|\boldsymbol{g}_i\|
$$

$$
+ \eta(1 + \xi\eta L)\left(2\xi\eta L + 2\xi\eta^2 L^2 + \xi^2\eta^3 L^3\right) \sum_{\ell=0}^{i-2} \left(1 + \eta L + \xi\eta^2 L^2\right)^{i-\ell-2} \|\boldsymbol{g}_{\ell+1}\|.
\tag{41}
$$

*Proof.* By the fundamental theorem of calculus for line integrals and the update rule (35), we have

$$
\begin{aligned}
\boldsymbol{w}_i &= \boldsymbol{z}_i - \xi\eta\boldsymbol{F}_i\boldsymbol{z}_i \\
&= \boldsymbol{z}_i - \xi\eta\boldsymbol{F}_i\boldsymbol{z}_0 - \xi\eta(\boldsymbol{F}_i\boldsymbol{z}_i - \boldsymbol{F}_i\boldsymbol{z}_0) \\
&= \boldsymbol{z}_i - \xi\eta\boldsymbol{F}_i\boldsymbol{z}_0 - \xi\eta\int_0^1 D\boldsymbol{F}_i(\boldsymbol{z}_0 + t(\boldsymbol{z}_i - \boldsymbol{z}_0))\,\mathrm{d}t\,(\boldsymbol{z}_i - \boldsymbol{z}_0)
\end{aligned}
$$

and similarly

$$
\begin{aligned}
\boldsymbol{z}_{i+1} &= \boldsymbol{z}_i - \eta\boldsymbol{F}_i\boldsymbol{w}_i \\
&= \boldsymbol{z}_i - \eta\boldsymbol{F}_i\boldsymbol{z}_0 - \eta(\boldsymbol{F}_i\boldsymbol{w}_i - \boldsymbol{F}_i\boldsymbol{z}_0) \\
&= \boldsymbol{z}_i - \eta\boldsymbol{F}_i\boldsymbol{z}_0 - \eta\int_0^1 D\boldsymbol{F}_i(\boldsymbol{z}_0 + t(\boldsymbol{w}_i - \boldsymbol{z}_0))\,\mathrm{d}t\,(\boldsymbol{w}_i - \boldsymbol{z}_0).
\end{aligned}
$$

Hence, by defining

$$
\begin{aligned}
\boldsymbol{A}_i &:= \int_0^1 D\boldsymbol{F}_i(\boldsymbol{z}_0 + t(\boldsymbol{z}_i - \boldsymbol{z}_0))\, \mathrm{d}t \\
\boldsymbol{B}_i &:= \int_0^1 D\boldsymbol{F}_i(\boldsymbol{z}_0 + t(\boldsymbol{w}_i - \boldsymbol{z}_0))\, \mathrm{d}t
\end{aligned}
\tag{42}
$$

the update rule can be rewritten using these quantities as

$$
\boldsymbol{w}_i = \boldsymbol{z}_i - \xi\eta\boldsymbol{F}_i\boldsymbol{z}_0 - \xi\eta\boldsymbol{A}_i(\boldsymbol{z}_i - \boldsymbol{z}_0), \tag{43}
$$
$$
\boldsymbol{z}_{i+1} = \boldsymbol{z}_i - \eta\boldsymbol{F}_i\boldsymbol{z}_0 - \eta\boldsymbol{B}_i(\boldsymbol{w}_i - \boldsymbol{z}_0). \tag{44}
$$

Subtracting $\boldsymbol{z}_0$ from both sides of (43) we get

$$
\begin{aligned}
\boldsymbol{w}_i - \boldsymbol{z}_0 &= \boldsymbol{z}_i - \boldsymbol{z}_0 - \xi\eta\boldsymbol{F}_i\boldsymbol{z}_0 - \xi\eta\boldsymbol{A}_i(\boldsymbol{z}_i - \boldsymbol{z}_0) \\
&= \left(\boldsymbol{I} - \xi\eta\boldsymbol{A}_i\right)(\boldsymbol{z}_i - \boldsymbol{z}_0) - \xi\eta\boldsymbol{F}_i\boldsymbol{z}_0,
\end{aligned}
\tag{45}
$$

and plugging this into (44) gives us

$$
\begin{aligned}
\boldsymbol{z}_{i+1} - \boldsymbol{z}_0 &= \boldsymbol{z}_i - \boldsymbol{z}_0 - \eta\boldsymbol{F}_i\boldsymbol{z}_0 - \eta\boldsymbol{B}_i(\boldsymbol{w}_i - \boldsymbol{z}_0) \\
&= \boldsymbol{z}_i - \boldsymbol{z}_0 - \eta\boldsymbol{F}_i\boldsymbol{z}_0 - \eta\boldsymbol{B}_i\left(\left(\boldsymbol{I} - \xi\eta\boldsymbol{A}_i\right)(\boldsymbol{z}_i - \boldsymbol{z}_0) - \xi\eta\boldsymbol{F}_i\boldsymbol{z}_0\right) \\
&= \left(\boldsymbol{I} - \eta\boldsymbol{B}_i + \xi\eta^2\boldsymbol{B}_i\boldsymbol{A}_i\right)(\boldsymbol{z}_i - \boldsymbol{z}_0) - \eta\left(\boldsymbol{I} - \xi\eta\boldsymbol{B}_i\right)\boldsymbol{F}_i\boldsymbol{z}_0.
\end{aligned}
\tag{46}
$$

For convenience let us define

$$
\begin{aligned}
\boldsymbol{C}_i &:= \boldsymbol{I} - \eta\boldsymbol{B}_i + \xi\eta^2\boldsymbol{B}_i\boldsymbol{A}_i, \\
\boldsymbol{P}_{i,\ell} &:= \boldsymbol{C}_i\boldsymbol{C}_{i-1}\ldots\boldsymbol{C}_{\ell+2}\boldsymbol{C}_{\ell+1}
\end{aligned}
$$

and $\boldsymbol{P}_{i,i} := \boldsymbol{I}$ as it denotes an empty product. Observe that for any $j$ we have

$$
\|\boldsymbol{C}_j\| = \left\|\boldsymbol{I} - \eta\boldsymbol{B}_i + \xi\eta^2\boldsymbol{B}_i\boldsymbol{A}_i\right\| \leq 1 + \eta L + \xi\eta^2 L^2. \tag{47}
$$

Also note that for any $\ell$ it holds that

$$
\begin{aligned}
&(\boldsymbol{I} - \xi\eta\boldsymbol{B}_{\ell+1}) - \boldsymbol{C}_{\ell+1}(\boldsymbol{I} - \xi\eta\boldsymbol{B}_\ell) \\
&\quad = (\boldsymbol{I} - \xi\eta\boldsymbol{B}_{\ell+1}) - \left(\boldsymbol{I} - \eta\boldsymbol{B}_{\ell+1} + \xi\eta^2\boldsymbol{B}_{\ell+1}\boldsymbol{A}_{\ell+1}\right)(\boldsymbol{I} - \xi\eta\boldsymbol{B}_\ell) \\
&\quad = \xi\eta(\boldsymbol{B}_{\ell+1} + \boldsymbol{B}_\ell) - \xi\eta^2\boldsymbol{B}_{\ell+1}(\boldsymbol{A}_{\ell+1} + \boldsymbol{B}_\ell) + \xi^2\eta^3\boldsymbol{B}_{\ell+1}\boldsymbol{A}_{\ell+1}\boldsymbol{B}_\ell
\end{aligned}
$$

and hence

$$
\|(\boldsymbol{I} - \xi\eta\boldsymbol{B}_{\ell+1}) - \boldsymbol{C}_{\ell+1}(\boldsymbol{I} - \xi\eta\boldsymbol{B}_\ell)\| \leq 2\xi\eta L + 2\xi\eta^2 L^2 + \xi^2\eta^3 L^3. \tag{48}
$$

Unravelling the recurrence relation (46) we get

$$
\begin{aligned}
\boldsymbol{z}_{i+1} - \boldsymbol{z}_0 &= \boldsymbol{C}_i(\boldsymbol{z}_i - \boldsymbol{z}_0) - \eta\left(\boldsymbol{I} - \xi\eta\boldsymbol{B}_i\right)\boldsymbol{F}_i\boldsymbol{z}_0 \\
&= \boldsymbol{C}_i\big(\boldsymbol{C}_{i-1}(\boldsymbol{z}_{i-1} - \boldsymbol{z}_0) - \eta\left(\boldsymbol{I} - \xi\eta\boldsymbol{B}_{i-1}\right)\boldsymbol{F}_{i-1}\boldsymbol{z}_0\big) - \eta\left(\boldsymbol{I} - \xi\eta\boldsymbol{B}_i\right)\boldsymbol{F}_i\boldsymbol{z}_0 \\
&= \boldsymbol{P}_{i,i-2}(\boldsymbol{z}_{i-1} - \boldsymbol{z}_0) - \eta\sum_{\ell=i-1}^{i}\boldsymbol{P}_{i,\ell}\left(\boldsymbol{I} - \xi\eta\boldsymbol{B}_\ell\right)\boldsymbol{F}_\ell\boldsymbol{z}_0 \\
&= \boldsymbol{P}_{i,i-2}\big(\boldsymbol{C}_{i-2}(\boldsymbol{z}_{i-2} - \boldsymbol{z}_0) - \eta\left(\boldsymbol{I} - \xi\eta\boldsymbol{B}_{i-2}\right)\boldsymbol{F}_{i-2}\boldsymbol{z}_0\big) - \eta\sum_{\ell=i-1}^{i}\boldsymbol{P}_{i,\ell}\left(\boldsymbol{I} - \xi\eta\boldsymbol{B}_\ell\right)\boldsymbol{F}_\ell\boldsymbol{z}_0 \\
&= \boldsymbol{P}_{i,i-3}(\boldsymbol{z}_{i-2} - \boldsymbol{z}_0) - \eta\sum_{\ell=i-2}^{i}\boldsymbol{P}_{i,\ell}\left(\boldsymbol{I} - \xi\eta\boldsymbol{B}_\ell\right)\boldsymbol{F}_\ell\boldsymbol{z}_0 \\
&\;\;\vdots \\
&= \boldsymbol{P}_{i,-1}(\boldsymbol{z}_0 - \boldsymbol{z}_0) - \eta\sum_{\ell=0}^{i}\boldsymbol{P}_{i,\ell}\left(\boldsymbol{I} - \xi\eta\boldsymbol{B}_\ell\right)\boldsymbol{F}_\ell\boldsymbol{z}_0
\end{aligned}
$$

and therefore

$$z_i - z_0 = -\eta \sum_{\ell=0}^{i-1} P_{i-1,\ell} \left( I - \xi\eta B_\ell \right) F_\ell z_0. \tag{49}$$

In order to compute the bound for $\|z_i - z_0\|$, we use summation by parts to get

$$\frac{1}{\eta}(z_0 - z_i) = \sum_{\ell=0}^{i-1} P_{i-1,\ell} \left( I - \xi\eta B_\ell \right) F_\ell z_0$$

$$= P_{i-1,i-1} \left( I - \xi\eta B_{i-1} \right) \sum_{\ell=0}^{i-1} F_\ell z_0$$

$$- \sum_{\ell=0}^{i-2} \left( P_{i-1,\ell+1} \left( I - \xi\eta B_{\ell+1} \right) - P_{i-1,\ell} \left( I - \xi\eta B_\ell \right) \right) \sum_{j=0}^{\ell} F_\ell z_0$$

$$= \left( I - \xi\eta B_{i-1} \right) g_i - \sum_{\ell=0}^{i-2} \left( P_{i-1,\ell+1} \left( I - \xi\eta B_{\ell+1} \right) - P_{i-1,\ell} \left( I - \xi\eta B_\ell \right) \right) g_{\ell+1}.$$

Here, observe that

$$P_{i-1,\ell+1} \left( I - \xi\eta B_{\ell+1} \right) - P_{i-1,\ell} \left( I - \xi\eta B_\ell \right)$$
$$= C_{i-1} C_{i-2} \dots C_{\ell+2} \left( \left( I - \xi\eta B_{\ell+1} \right) - C_{\ell+1} \left( I - \xi\eta B_\ell \right) \right)$$

so by using (47) and (48) we obtain

$$\|P_{i-1,\ell+1} \left( I - \xi\eta B_{\ell+1} \right) - P_{i-1,\ell} \left( I - \xi\eta B_\ell \right)\|$$
$$\leq \left( 2\xi\eta L + 2\xi\eta^2 L^2 + \xi^2\eta^3 L^3 \right) \left( 1 + \eta L + \xi\eta^2 L^2 \right)^{i-\ell-2}.$$

Therefore, we conclude that

$$\|z_i - z_0\| \leq \eta \left( 1 + \xi\eta L \right) \|g_i\| + \eta^2 L \left( 2\xi + 2\xi\eta L + \xi^2\eta^2 L^2 \right) \sum_{\ell=0}^{i-2} \left( 1 + \eta L + \xi\eta^2 L^2 \right)^{i-\ell-2} \|g_{\ell+1}\|.$$

Meanwhile, substituting (49) back to (45) gives us

$$w_i - z_0 = -\xi\eta F_i z_0 - \eta \sum_{\ell=0}^{i-1} \left( I - \xi\eta A_i \right) P_{i-1,\ell} \left( I - \xi\eta B_\ell \right) F_\ell z_0. \tag{50}$$

For $i > \ell$ let us define

$$R_{i,\ell} := \xi^{-1} \left( I - \xi\eta A_i \right) P_{i-1,\ell} \left( I - \xi\eta B_\ell \right)$$
$$= \xi^{-1} \left( I - \xi\eta A_i \right) C_{i-1} C_{i-2} \dots C_{\ell+2} C_{\ell+1} \left( I - \xi\eta B_\ell \right)$$

and for convenience $R_{i,i} := I$ so that (50) can be rewritten as

$$\frac{1}{\xi\eta}(z_0 - w_i) = \sum_{\ell=0}^{i} R_{i,\ell} F_\ell z_0. \tag{51}$$

Applying summation by parts on the above, we obtain

$$\frac{1}{\xi\eta}(z_0 - w_i) = R_{i,i} \sum_{\ell=0}^{i} F_\ell z_0 - \sum_{\ell=0}^{i-1} (R_{i,\ell+1} - R_{i,\ell}) \sum_{j=0}^{\ell} F_j z_0$$

$$= g_{i+1} - \sum_{\ell=0}^{i-1} (R_{i,\ell+1} - R_{i,\ell}) g_{\ell+1}$$

and as a consequence we get

$$\frac{1}{\xi\eta} \|w_i - z_0\| \leq \|g_{i+1}\| + \sum_{\ell=0}^{i-1} \|R_{i,\ell+1} - R_{i,\ell}\| \|g_{\ell+1}\|. \tag{52}$$

It remains to bound $\|\boldsymbol{R}_{i,\ell+1} - \boldsymbol{R}_{i,\ell}\|$. For the special case where $\ell = i - 1$, a direct computation leads to

$$\boldsymbol{R}_{i,i} - \boldsymbol{R}_{i,i-1} = \boldsymbol{I} - \xi^{-1}\left(\boldsymbol{I} - \xi\eta\boldsymbol{A}_i\right)\left(\boldsymbol{I} - \xi\eta\boldsymbol{B}_{i-1}\right)$$
$$= (1 - \xi^{-1})\boldsymbol{I} + \eta\boldsymbol{A}_i + \eta\boldsymbol{B}_{i-1} - \xi\eta^2\boldsymbol{A}_i\boldsymbol{B}_{i-1}$$

and thus we have

$$\|\boldsymbol{R}_{i,i} - \boldsymbol{R}_{i,i-1}\| \le (1 - \xi^{-1}) + 2\eta L + \xi\eta^2 L^2. \tag{53}$$

For the other cases; that is, when $\ell < i - 1$, we have

$$\boldsymbol{R}_{i,\ell+1} - \boldsymbol{R}_{i,\ell} = \xi^{-1}\left(\boldsymbol{I} - \xi\eta\boldsymbol{A}_i\right)\boldsymbol{C}_{i-1}\boldsymbol{C}_{i-2}\ldots\boldsymbol{C}_{\ell+2}\left(\left(\boldsymbol{I} - \xi\eta\boldsymbol{B}_{\ell+1}\right) - \boldsymbol{C}_{\ell+1}\left(\boldsymbol{I} - \xi\eta\boldsymbol{B}_\ell\right)\right)$$

so by using (47) and (48) we get the bound

$$\|\boldsymbol{R}_{i,\ell+1} - \boldsymbol{R}_{i,\ell}\| \le \xi^{-1}(1 + \xi\eta L)\left(2\xi\eta L + 2\xi\eta^2 L^2 + \xi^2\eta^3 L^3\right)\left(1 + \eta L + \xi\eta^2 L^2\right)^{i-\ell-2}. \tag{54}$$

Applying (53) and (54) on (52) gives the bound for $\|\boldsymbol{w}_i - \boldsymbol{z}_0\|$. $\qquad\square$

**Proposition D.2.** *Suppose that* SEG-FFA *is used,* $\eta < \frac{1}{nL}$*, and let* $\nu := 1 + \frac{1}{2n}$*. Then for any* $i = 1, \ldots, 2n - 1$ *we have the bounds*

$$\|\boldsymbol{z}_i - \boldsymbol{z}_0\| \le \left(\eta\nu i + \frac{\eta\nu^2 e^2 i(i-1)}{2n}\right)\|\boldsymbol{F}\boldsymbol{z}_0\| + \eta\nu\delta_i + \eta^2 L\nu^2 e^2 \Sigma_{i-1},$$

$$\|\boldsymbol{w}_i - \boldsymbol{z}_0\| \le \frac{\eta}{2}\left(1 + 2\nu^2 i + \frac{\nu^3 e^2 i(i-1)}{n}\right)\|\boldsymbol{F}\boldsymbol{z}_0\| + \frac{\eta}{2}\delta_{i+1} + \frac{\eta(2\nu^2 - 1)}{2}\delta_i + \eta^2 L\nu^3 e^2 \Sigma_{i-1},$$

$$\|\boldsymbol{w}_i - \boldsymbol{z}_0\|^2 \le \left(\frac{3\eta^2(i+1)^2}{2} + \frac{3\eta^2(2\nu^2 - 1)^2 i^2}{2} + \frac{\eta^2\nu^6 e^4 i(i-1)^2(2i-1)}{n^2}\right)\|\boldsymbol{F}\boldsymbol{z}_0\|^2$$
$$+ \frac{3\eta^2}{2}\delta_{i+1}^2 + \frac{3\eta^2(2\nu^2 - 1)^2}{2}\delta_i^2 + \frac{6\eta^2\nu^6 e^4(i-1)}{n^2}\Psi_{i-1}.$$

*Proof.* Using elementary calculus one can show that $x \mapsto (1 + \frac{1}{x} + \frac{1}{2x^2})^x$ increases on $x > 0$ and is bounded above by $e$. Hence for all $0 \le \ell < i \le 2n$ we have

$$\left(1 + \eta L + \frac{\eta^2 L^2}{2}\right)^{i-\ell-2} \le \left(1 + \frac{1}{n} + \frac{1}{2n^2}\right)^{2n} \le e^2.$$

Applying the definitions (37) and (38) on (40) and then substituting $\xi = 1/2$ we get

$$\|\boldsymbol{z}_i - \boldsymbol{z}_0\| \le \eta\left(1 + \frac{\eta L}{2}\right)\|\boldsymbol{g}_i\| + \eta^2 L\left(1 + \frac{\eta L}{2}\right)^2 \sum_{\ell=0}^{i-2}\left(1 + \eta L + \frac{\eta^2 L^2}{2}\right)^{i-\ell-2}\|\boldsymbol{g}_{\ell+1}\|$$

$$\le \eta\nu\left(i\|\boldsymbol{F}\boldsymbol{z}_0\| + \delta_i\right) + \eta^2 L\nu^2 \sum_{\ell=0}^{i-2} e^2\left((\ell+1)\|\boldsymbol{F}\boldsymbol{z}_0\| + \delta_{\ell+1}\right)$$

$$\le \eta\nu\left(i\|\boldsymbol{F}\boldsymbol{z}_0\| + \delta_i\right) + \frac{\eta\nu^2 e^2 i(i-1)}{2n}\|\boldsymbol{F}\boldsymbol{z}_0\| + \eta^2 L\nu^2 e^2 \Sigma_{i-1}.$$

Similarly, from (41) we get

$$\|\boldsymbol{w}_i - \boldsymbol{z}_0\| \le \frac{\eta}{2}\|\boldsymbol{g}_{i+1}\| + \frac{\eta}{2}\left(1 + 2\eta L + \frac{\eta^2 L^2}{2}\right)\|\boldsymbol{g}_i\|$$

$$+ \eta^2 L\left(1 + \frac{\eta L}{2}\right)^3 \sum_{\ell=0}^{i-2}\left(1 + \eta L + \frac{\eta^2 L^2}{2}\right)^{i-\ell-2}\|\boldsymbol{g}_{\ell+1}\|$$

$$\le \frac{\eta}{2}\left((i+1)\|\boldsymbol{F}\boldsymbol{z}_0\| + \delta_{i+1}\right) + \frac{\eta}{2}\left(1 + \frac{2}{n} + \frac{1}{2n^2}\right)\left(i\|\boldsymbol{F}\boldsymbol{z}_0\| + \delta_i\right)$$

$$+ \eta^2 L\nu^3 \sum_{\ell=0}^{i-2} e^2\left((\ell+1)\|\boldsymbol{F}\boldsymbol{z}_0\| + \delta_{\ell+1}\right)$$

$$\le \frac{\eta}{2}(1 + 2i\nu^2)\|\boldsymbol{F}\boldsymbol{z}_0\| + \frac{\eta\nu^3 e^2 i(i-1)}{2n}\|\boldsymbol{F}\boldsymbol{z}_0\| + \frac{\eta}{2}\delta_{i+1} + \frac{\eta(2\nu^2 - 1)}{2}\delta_i + \eta^2 L\nu^3 e^2 \Sigma_{i-1}.$$

Finally, applying generalized Young's inequality on (41) we get

$$\|\boldsymbol{w}_i - \boldsymbol{z}_0\|^2 \le \frac{3\eta^2}{4} \|\boldsymbol{g}_{i+1}\|^2 + \frac{3\eta^2}{4} \left(1 + 2\eta L + \frac{\eta^2 L^2}{2}\right)^2 \|\boldsymbol{g}_i\|^2$$

$$+ 3 \left(\eta^2 L \left(1 + \frac{\eta L}{2}\right)^3 \sum_{\ell=0}^{i-2} \left(1 + \eta L + \frac{\eta^2 L^2}{2}\right)^{i-\ell-2} \|\boldsymbol{g}_{\ell+1}\|\right)^2.$$

Using generalized Young's inequality once more on the last term gives us

$$3 \left(\eta^2 L \left(1 + \frac{\eta L}{2}\right)^3 \sum_{\ell=0}^{i-2} \left(1 + \eta L + \frac{\eta^2 L^2}{2}\right)^{i-\ell-2} \|\boldsymbol{g}_{\ell+1}\|\right)^2 \le 3 \left(\frac{\eta\nu^3 e^2}{n} \sum_{\ell=0}^{i-2} \|\boldsymbol{g}_{\ell+1}\|\right)^2$$

$$\le \frac{3\eta^2 \nu^6 e^4 (i-1)}{n^2} \sum_{\ell=0}^{i-2} \|\boldsymbol{g}_{\ell+1}\|^2.$$

Plugging this back yields

$$\|\boldsymbol{w}_i - \boldsymbol{z}_0\|^2 \le \frac{3\eta^2}{4} \|\boldsymbol{g}_{i+1}\|^2 + \frac{3\eta^2}{4} \left(1 + 2\eta L + \frac{\eta^2 L^2}{2}\right)^2 \|\boldsymbol{g}_i\|^2 + \frac{3\eta^2 \nu^6 e^4 (i-1)}{n^2} \sum_{\ell=0}^{i-2} \|\boldsymbol{g}_{\ell+1}\|^2$$

$$\le \frac{3\eta^2}{4} \left(2(i+1)^2 \|\boldsymbol{F}\boldsymbol{z}_0\|^2 + 2\delta_{i+1}^2\right) + \frac{3\eta^2}{4} \left(2\nu^2 - 1\right)^2 \left(2i^2 \|\boldsymbol{F}\boldsymbol{z}_0\|^2 + 2\delta_i^2\right)$$

$$+ \frac{3\eta^2 \nu^6 e^4 (i-1)}{n^2} \sum_{\ell=0}^{i-2} \left(2(\ell+1)^2 \|\boldsymbol{F}\boldsymbol{z}_0\|^2 + 2\delta_{\ell+1}^2\right)$$

$$\le \frac{3\eta^2}{2} \left((i+1)^2 \|\boldsymbol{F}\boldsymbol{z}_0\|^2 + \delta_{i+1}^2\right) + \frac{3\eta^2}{2} \left(2\nu^2 - 1\right)^2 \left(i^2 \|\boldsymbol{F}\boldsymbol{z}_0\|^2 + \delta_i^2\right)$$

$$+ \frac{\eta^2 \nu^6 e^4 i(i-1)^2(2i-1)}{n^2} \|\boldsymbol{F}\boldsymbol{z}_0\|^2 + \frac{6\eta^2 \nu^6 e^4 (i-1)}{n^2} \Psi_{i-1}.$$

Now the claimed inequalities can be obtained simply by rearranging the terms appropriately. □

**Proposition D.3.** *Suppose that either* SEG-RR *or* SEG-FF *is used with* $\alpha = \beta = \eta < \frac{1}{nL}$, *and let* $\tilde{\nu} := 1 + \frac{1}{n}$. *Then for any* $i = 1, \dots, 2n-1$ *we have the bounds*

$$\|\boldsymbol{z}_i - \boldsymbol{z}_0\| \le \left(\eta\tilde{\nu}i + \frac{16\eta\tilde{\nu}^2 i(i-1)}{n}\right) \|\boldsymbol{F}\boldsymbol{z}_0\| + \eta\tilde{\nu}\delta_i + 32\eta^2 L\tilde{\nu}^2 \Sigma_{i-1},$$

$$\|\boldsymbol{w}_i - \boldsymbol{z}_0\| \le \eta \left(1 + i\tilde{\nu}^2 + \frac{16\tilde{\nu}^3 i(i-1)}{n}\right) \|\boldsymbol{F}\boldsymbol{z}_0\| + \eta\delta_{i+1} + \eta(\tilde{\nu}^2 - 1)\delta_i + 32\eta^2 L\tilde{\nu}^3 \Sigma_{i-1},$$

$$\|\boldsymbol{w}_i - \boldsymbol{z}_0\|^2 \le \left(6\eta^2(i+1)^2 + \frac{6\eta^2 (1+\tilde{\nu})^2 i^2}{n^2} + \frac{1024\eta^2 \tilde{\nu}^6 i(i-1)^2(2i-1)}{n^2}\right) \|\boldsymbol{F}\boldsymbol{z}_0\|^2$$

$$+ 6\eta^2 \delta_{i+1}^2 + \frac{6\eta^2 (1+\tilde{\nu})^2}{n^2} \delta_i^2 + \frac{6144\eta^2 \tilde{\nu}^6 (i-1)}{n^2} \Psi_{i-1}.$$

*Proof.* One can verify that $x \mapsto (1 + \frac{4}{3x})^x$ increases on $x \ge 3$ and is bounded above by $e^{4/3} < 4$. With noting that $(1 + \frac{1}{1} + \frac{1}{1^2})^1 = 3 < 4$, $(1 + \frac{1}{2} + \frac{1}{2^2})^2 = \frac{49}{16} < 4$, and $1 + \frac{1}{x} + \frac{1}{x^2} \le 1 + \frac{4}{3x}$ whenever $x \ge 3$, we see that for all $0 \le \ell < i \le 2n$ it holds that

$$\left(1 + \eta L + \eta^2 L^2\right)^{i-\ell-2} \le \left(1 + \frac{1}{n} + \frac{1}{n^2}\right)^{2n} \le 4^2 = 16.$$

Also, we have

$$2 + 2\eta L + \eta^2 L^2 \le 2 + \frac{2}{n} + \frac{1}{n^2} = 1 + \tilde{\nu}^2 \le 2\tilde{\nu}^2.$$

Applying the definitions (37) and (38) on (40) and then substituting $\xi = 1$ we get

$$\|\boldsymbol{z}_i - \boldsymbol{z}_0\| \leq \eta\left(1 + \eta L\right)\|\boldsymbol{g}_i\| + \eta^2 L\left(2 + 2\eta L + \eta^2 L^2\right)\sum_{\ell=0}^{i-2}\left(1 + \eta L + \eta^2 L^2\right)^{i-\ell-2}\|\boldsymbol{g}_{\ell+1}\|$$

$$\leq \eta\tilde{\nu}\left(i\|\boldsymbol{F}\boldsymbol{z}_0\| + \delta_i\right) + 2\eta^2 L\tilde{\nu}^2\sum_{\ell=0}^{i-2}16\left((\ell+1)\|\boldsymbol{F}\boldsymbol{z}_0\| + \delta_{\ell+1}\right)$$

$$\leq \eta\tilde{\nu}\left(i\|\boldsymbol{F}\boldsymbol{z}_0\| + \delta_i\right) + \frac{16\eta\tilde{\nu}^2 i(i-1)}{n}\|\boldsymbol{F}\boldsymbol{z}_0\| + 32\eta^2 L\tilde{\nu}^2\Sigma_{i-1}.$$

Similarly, from (41) we get

$$\|\boldsymbol{w}_i - \boldsymbol{z}_0\| \leq \eta\|\boldsymbol{g}_{i+1}\| + \eta^2 L\left(2 + \eta L\right)\|\boldsymbol{g}_i\|$$

$$+ \eta^2 L(1 + \eta L)\left(2 + 2\eta L + \eta^2 L^2\right)\sum_{\ell=0}^{i-2}\left(1 + \eta L + \eta^2 L^2\right)^{i-\ell-2}\|\boldsymbol{g}_{\ell+1}\|.$$

$$\leq \eta\left((i+1)\|\boldsymbol{F}\boldsymbol{z}_0\| + \delta_{i+1}\right) + \frac{\eta}{n}\left(2 + \frac{1}{n}\right)\left(i\|\boldsymbol{F}\boldsymbol{z}_0\| + \delta_i\right)$$

$$+ \eta^2 L\tilde{\nu}\left(2\tilde{\nu}^2\right)\sum_{\ell=0}^{i-2}16\left(\ell\|\boldsymbol{F}\boldsymbol{z}_0\| + \delta_\ell\right).$$

$$\leq \eta(1 + i\tilde{\nu}^2)\|\boldsymbol{F}\boldsymbol{z}_0\| + \frac{16\eta\tilde{\nu}^3 i(i-1)}{n}\|\boldsymbol{F}\boldsymbol{z}_0\| + \eta\delta_{i+1} + \eta(\tilde{\nu}^2 - 1)\delta_i + 32\eta^2 L\tilde{\nu}^3\Sigma_{i-1}.$$

Finally, applying Young's inequality on (41) we get

$$\|\boldsymbol{w}_i - \boldsymbol{z}_0\|^2 \leq 3\eta^2\|\boldsymbol{g}_{i+1}\|^2 + \frac{3\eta^2\left(2 + \eta L\right)^2}{n^2}\|\boldsymbol{g}_i\|$$

$$+ 3\left(\eta^2 L(1 + \eta L)\left(2 + 2\eta L + \eta^2 L^2\right)\sum_{\ell=0}^{i-2}\left(1 + \eta L + \eta^2 L^2\right)^{i-\ell-2}\|\boldsymbol{g}_{\ell+1}\|\right)^2$$

$$\leq 3\eta^2\|\boldsymbol{g}_{i+1}\|^2 + \frac{3\eta^2\left(2 + \eta L\right)^2}{n^2}\|\boldsymbol{g}_i\| + 3\left(\frac{2\eta\tilde{\nu}^3}{n}\sum_{\ell=0}^{i-2}16\|\boldsymbol{g}_{\ell+1}\|\right)^2.$$

Using Young's inequality once more on the last term gives us

$$3\left(\frac{32\eta\tilde{\nu}^3}{n}\sum_{\ell=0}^{i-2}\|\boldsymbol{g}_{\ell+1}\|\right)^2 \leq \frac{3072\eta^2\tilde{\nu}^6(i-1)}{n^2}\sum_{\ell=0}^{i-2}\|\boldsymbol{g}_{\ell+1}\|^2.$$

Plugging this back yields

$$\|\boldsymbol{w}_i - \boldsymbol{z}_0\|^2 \leq 3\eta^2\|\boldsymbol{g}_{i+1}\|^2 + \frac{3\eta^2\left(2 + \eta L\right)^2}{n^2}\|\boldsymbol{g}_i\| + \frac{3072\eta^2\tilde{\nu}^6(i-1)}{n^2}\sum_{\ell=0}^{i-2}\|\boldsymbol{g}_{\ell+1}\|^2$$

$$\leq 6\eta^2\left((i+1)^2\|\boldsymbol{F}\boldsymbol{z}_0\|^2 + \delta_{i+1}^2\right) + \frac{6\eta^2\left(2 + \eta L\right)^2}{n^2}\left(i^2\|\boldsymbol{F}\boldsymbol{z}_0\|^2 + \delta_i^2\right)$$

$$+ \frac{6144\eta^2\tilde{\nu}^6(i-1)}{n^2}\sum_{\ell=0}^{i-2}\left((\ell+1)^2\|\boldsymbol{F}\boldsymbol{z}_0\|^2 + \delta_{\ell+1}^2\right)$$

$$\leq 6\eta^2\left((i+1)^2\|\boldsymbol{F}\boldsymbol{z}_0\|^2 + \delta_{i+1}^2\right) + \frac{6\eta^2\left(1 + \tilde{\nu}\right)^2}{n^2}\left(i^2\|\boldsymbol{F}\boldsymbol{z}_0\|^2 + \delta_i^2\right)$$

$$+ \frac{1024\eta^2\tilde{\nu}^6 i(i-1)^2(2i-1)}{n^2}\|\boldsymbol{F}\boldsymbol{z}_0\|^2 + \frac{6144\eta^2\tilde{\nu}^6(i-1)}{n^2}\Psi_{i-1}.$$

Now the claimed inequalities can be obtained simply by rearranging the terms appropriately. $\qquad\square$

**Lemma D.4.** *For any* $j = 1, \ldots, 2n$, *it deterministically holds that*

$$\delta_j \leq n(\rho\|\boldsymbol{F}\boldsymbol{z}_0\| + \sigma). \tag{55}$$

*Proof.* For any set of indices $\mathcal{J} \subset \{0, \ldots, n-1\}$, by Assumption 4 it holds that

$$\sum_{i \in \mathcal{J}} \|\boldsymbol{F}_i \boldsymbol{z}_0 - \boldsymbol{F} \boldsymbol{z}_0\|^2 \leq \sum_{i=0}^{n-1} \|\boldsymbol{F}_i \boldsymbol{z}_0 - \boldsymbol{F} \boldsymbol{z}_0\|^2 \leq n(\rho \|\boldsymbol{F} \boldsymbol{z}_0\| + \sigma)^2.$$

Hence, for any $j = 1, \ldots, n$ we have

$$\begin{aligned}
\|\boldsymbol{g}_j - j\boldsymbol{F} \boldsymbol{z}_0\|^2 &= \left\| \sum_{i=0}^{j-1} \boldsymbol{F}_i \boldsymbol{z}_0 - j\boldsymbol{F} \boldsymbol{z}_0 \right\|^2 \\
&\leq j \sum_{i=0}^{j-1} \|\boldsymbol{F}_i \boldsymbol{z}_0 - \boldsymbol{F} \boldsymbol{z}_0\|^2 \\
&\leq jn(\rho \|\boldsymbol{F} \boldsymbol{z}_0\| + \sigma)^2 \\
&\leq n^2 (\rho \|\boldsymbol{F} \boldsymbol{z}_0\| + \sigma)^2,
\end{aligned}$$

and for any $j = n+1, \ldots, 2n$ we have

$$\begin{aligned}
\|\boldsymbol{g}_j - j\boldsymbol{F} \boldsymbol{z}_0\|^2 &= \left\| \sum_{i=0}^{n-1} \boldsymbol{F}_i \boldsymbol{z}_0 + \sum_{i=n}^{j-1} \boldsymbol{F}_i \boldsymbol{z}_0 - j\boldsymbol{F} \boldsymbol{z}_0 \right\|^2 \\
&= \left\| \sum_{i=n}^{j-1} \boldsymbol{F}_i \boldsymbol{z}_0 - (j-n)\boldsymbol{F} \boldsymbol{z}_0 \right\|^2 \\
&= \left\| \sum_{i=2n-j}^{n-1} \boldsymbol{F}_i \boldsymbol{z}_0 - (j-n)\boldsymbol{F} \boldsymbol{z}_0 \right\|^2 \quad (56) \\
&\leq (j-n) \sum_{i=0}^{j-1} \|\boldsymbol{F}_i \boldsymbol{z}_0 - \boldsymbol{F} \boldsymbol{z}_0\|^2 \\
&\leq n^2 (\rho \|\boldsymbol{F} \boldsymbol{z}_0\| + \sigma)^2.
\end{aligned}$$

Therefore, in any case we have

$$\|\boldsymbol{g}_j - j\boldsymbol{F} \boldsymbol{z}_0\|^2 \leq n^2 (\rho \|\boldsymbol{F} \boldsymbol{z}_0\| + \sigma)^2.$$

Taking square roots on both sides gives us the desired bound. $\qquad \square$

**Lemma D.5.** *For any $j = 1, \ldots, 2n$, it holds that*

$$\mathbb{E}_\tau [\delta_j^2] \leq \frac{n(\rho \|\boldsymbol{F} \boldsymbol{z}_0\| + \sigma)^2}{2}, \quad (57)$$

*Proof.* If $n = 1$ then the left hand side is always 0, so there is nothing to show. So, we may assume that $n \geq 2$. Then, for any $j = 1, \ldots, n$, using (Mishchenko et al., 2020a, Lemma 1) we obtain

$$\mathbb{E}_\tau \left\| \frac{1}{j} \boldsymbol{g}_j - \boldsymbol{F} \boldsymbol{z}_0 \right\|^2 \leq \frac{n-j}{j(n-1)} (\rho \|\boldsymbol{F} \boldsymbol{z}_0\| + \sigma)^2.$$

Multiplying both sides by $j^2$ and applying AM-GM inequality leads to

$$\mathbb{E}_\tau \|\boldsymbol{g}_j - j\boldsymbol{F} \boldsymbol{z}_0\|^2 \leq \frac{j(n-j)}{n-1} (\rho \|\boldsymbol{F} \boldsymbol{z}_0\| + \sigma)^2 \leq \frac{n^2}{4(n-1)} (\rho \|\boldsymbol{F} \boldsymbol{z}_0\| + \sigma)^2 \leq \frac{n}{2} (\rho \|\boldsymbol{F} \boldsymbol{z}_0\| + \sigma)^2.$$

Meanwhile, for $j = n+1, \ldots, 2n$, following the first few steps in (56) we get

$$\|\boldsymbol{g}_j - j\boldsymbol{F} \boldsymbol{z}_0\|^2 = \left\| \sum_{i=2n-j}^{n-1} \boldsymbol{F}_i \boldsymbol{z}_0 - (j-n)\boldsymbol{F} \boldsymbol{z}_0 \right\|^2$$

Applying ([Mishchenko et al., 2020a](#), Lemma 1) here, we get

$$
\begin{aligned}
\mathbb{E}_\tau \left\| \boldsymbol{g}_j - j\boldsymbol{F}\boldsymbol{z}_0 \right\|^2 &= \mathbb{E}_\tau \left\| \sum_{i=2n-j}^{n-1} \boldsymbol{F}_i \boldsymbol{z}_0 - (j-n)\boldsymbol{F}\boldsymbol{z}_0 \right\|^2 \\
&= (j-n)^2 \, \mathbb{E}_\tau \left\| \frac{1}{j-n} \sum_{i=2n-j}^{n-1} \boldsymbol{F}_i \boldsymbol{z}_0 - \boldsymbol{F}\boldsymbol{z}_0 \right\|^2 \\
&\leq (j-n)^2 \cdot \frac{n-(j-n)}{(j-n)(n-1)} (\rho \left\| \boldsymbol{F}\boldsymbol{z}_0 \right\| + \sigma)^2 \\
&\leq \frac{(j-n)(2n-j)}{n-1} (\rho \left\| \boldsymbol{F}\boldsymbol{z}_0 \right\| + \sigma)^2.
\end{aligned}
$$

Using AM-GM inequality on the last line gives us

$$
\mathbb{E}_\tau \left\| \boldsymbol{g}_j - j\boldsymbol{F}\boldsymbol{z}_0 \right\|^2 \leq \frac{(j-n)(2n-j)}{n-1} (\rho \left\| \boldsymbol{F}\boldsymbol{z}_0 \right\| + \sigma)^2 \leq \frac{n^2}{4(n-1)} (\rho \left\| \boldsymbol{F}\boldsymbol{z}_0 \right\| + \sigma)^2 \leq \frac{n}{2} (\rho \left\| \boldsymbol{F}\boldsymbol{z}_0 \right\| + \sigma)^2.
$$

Thus, for any case, we have ([57](#)). $\qquad\square$

**Lemma D.6.** *For any $k, \ell \in \{0, 1, \ldots, 2n\}$, it holds that*

$$
\mathbb{E}_\tau [\Sigma_k \Sigma_\ell] \leq \frac{k\ell n (\rho \left\| \boldsymbol{F}\boldsymbol{z}_0 \right\| + \sigma)^2}{2}. \tag{58}
$$

*Proof.* Expanding the product $\Sigma_k \Sigma_\ell$ and writing in terms of $\delta$, we get

$$
\begin{aligned}
\Sigma_k \Sigma_\ell = \left( \sum_{i=1}^{k} \delta_i \right) \left( \sum_{j=1}^{\ell} \delta_j \right) &= \sum_{i=1}^{k} \sum_{j=1}^{\ell} \delta_i \delta_j \\
&\leq \sum_{i=1}^{k} \sum_{j=1}^{\ell} \frac{\delta_i^2 + \delta_j^2}{2}
\end{aligned}
$$

where the last line follows from the AM-GM inequality. Taking the expectation with respect to $\tau$ and using the bound from Lemma [D.5](#), we obtain

$$
\begin{aligned}
\mathbb{E}_\tau [\Sigma_k \Sigma_\ell] &\leq \frac{1}{2} \sum_{i=1}^{k} \sum_{j=1}^{\ell} \left( \mathbb{E}_\tau [\delta_i^2] + \mathbb{E}_\tau [\delta_j^2] \right) \\
&\leq \frac{1}{2} \sum_{i=1}^{k} \sum_{j=1}^{\ell} \left( \frac{n(\rho \left\| \boldsymbol{F}\boldsymbol{z}_0 \right\| + \sigma)^2}{2} + \frac{n(\rho \left\| \boldsymbol{F}\boldsymbol{z}_0 \right\| + \sigma)^2}{2} \right) \\
&= \frac{k\ell n (\rho \left\| \boldsymbol{F}\boldsymbol{z}_0 \right\| + \sigma)^2}{2}
\end{aligned}
$$

which is exactly the claimed. $\qquad\square$

**Lemma D.7.** *For any $k, \ell \in \{0, 1, \ldots, 2n\}$, it holds that*

$$
\mathbb{E}_\tau \left[ \left( \sum_{i=1}^{k} \Sigma_i \right) \left( \sum_{j=1}^{\ell} \Sigma_j \right) \right] \leq \frac{k(k+1)\ell(\ell+1)n(\rho \left\| \boldsymbol{F}\boldsymbol{z}_0 \right\| + \sigma)^2}{8}. \tag{59}
$$

*Proof.* Expanding the product in the left hand side of (59) and applying (58), we get

$$\mathbb{E}_{\tau}\left[\left(\sum_{i=1}^{k}\Sigma_i\right)\left(\sum_{j=1}^{\ell}\Sigma_j\right)\right] = \mathbb{E}_{\tau}\left[\sum_{i=1}^{k}\sum_{j=1}^{\ell}\Sigma_i\Sigma_j\right] = \sum_{i=1}^{k}\sum_{j=1}^{\ell}\mathbb{E}_{\tau}[\Sigma_i\Sigma_j]$$

$$\leq \sum_{i=1}^{k}\sum_{j=1}^{\ell}\frac{ijn(\rho\|\boldsymbol{F}\boldsymbol{z}_0\|+\sigma)^2}{2}$$

$$\leq \frac{k(k+1)\ell(\ell+1)n(\rho\|\boldsymbol{F}\boldsymbol{z}_0\|+\sigma)^2}{8}. \qquad \square$$

### D.2 UPPER BOUNDS OF THE WITHIN-EPOCH ERRORS

The full proof of Theorem 6.1 is quite long and technical, so we divide it into several parts. First we show that (19) and (20) holds with $a = 3$ when SEG-FFA is in use. Then we show that Theorem 6.1 also holds for SEG-FF in Appendix D.2.3, and for SEG-RR in Appendix D.2.4.

#### D.2.1 PROOF OF EQUATION (19) FOR SEG-FFA

In this section we prove the following.

**Theorem D.8.** *Say we use SEG-FFA. Then, as long as the stepsize used in an epoch satisfies $\eta < \frac{1}{nL}$, it holds that*

$$\|\boldsymbol{r}\| \leq \eta^3 n^3 C_{1A}\|\boldsymbol{F}\boldsymbol{z}_0\| + \eta^3 n^3 D_{1A}\|\boldsymbol{F}\boldsymbol{z}_0\|^2 + \eta^3 n^3 V_{1A} \tag{60}$$

*for constants*

$$C_{1A} := L^2\left(\frac{1}{2}\left(1+\frac{2e^2}{3}\right)+\frac{6+e^2}{3}+15\rho\right), \tag{61}$$

$$D_{1A} := M\left(\frac{83}{4}+\frac{24e^4}{5}+\rho^2\left(\frac{243}{16}+27e^4\right)\right), \tag{62}$$

$$V_{1A} := M\sigma^2\left(\frac{243}{16}+27e^4\right)+15L^2\sigma. \tag{63}$$

We first list the intermediate results. The actual proof of Theorem D.8 is in page 37, at the end of this section.

**Proposition D.9.** *For using SEG-FFA, the within-epoch update $\boldsymbol{z}^{\sharp}$ (12) satisfies*

$$\boldsymbol{z}^{\sharp} = \boldsymbol{z}_0 - n\eta\boldsymbol{F}(\boldsymbol{z}_0 - n\eta\boldsymbol{F}\boldsymbol{z}_0) + \boldsymbol{r}$$

*where we denote*

$$\boldsymbol{r} := n\eta\boldsymbol{F}(\boldsymbol{z}_0 - n\eta\boldsymbol{F}\boldsymbol{z}_0) - n\eta\boldsymbol{F}\boldsymbol{z}_0 + n^2\eta^2 D\boldsymbol{F}(\boldsymbol{z}_0)\boldsymbol{F}\boldsymbol{z}_0 \tag{64a}$$

$$-\frac{\eta}{2}\sum_{j=0}^{2n-1}\left(\boldsymbol{F}_j\boldsymbol{w}_j - \boldsymbol{F}_j\boldsymbol{z}_0 - D\boldsymbol{F}_j(\boldsymbol{z}_0)(\boldsymbol{w}_j - \boldsymbol{z}_0)\right) \tag{64b}$$

$$+\frac{\eta^2}{4}\sum_{j=0}^{2n-1}D\boldsymbol{F}_j(\boldsymbol{z}_0)(\boldsymbol{F}_j\boldsymbol{z}_j - \boldsymbol{F}_j\boldsymbol{z}_0) \tag{64c}$$

$$+\frac{\eta^2}{2}\sum_{j=0}^{2n-1}D\boldsymbol{F}_j(\boldsymbol{z}_0)\sum_{k=0}^{j-1}(\boldsymbol{F}_k\boldsymbol{w}_k - \boldsymbol{F}_k\boldsymbol{z}_0). \tag{64d}$$

*Proof.* Setting $\alpha = \eta/2$, $\beta = \eta$, and $\theta = 1$ in (13), we get

$$\boldsymbol{z}^{\sharp} = \boldsymbol{z}_0 - \frac{\eta}{2}\sum_{j=0}^{2n-1}\boldsymbol{F}_j\boldsymbol{z}_0 + \frac{\eta^2}{4}\sum_{j=0}^{2n-1}D\boldsymbol{F}_j(\boldsymbol{z}_0)\boldsymbol{F}_j\boldsymbol{z}_0 + \frac{\eta^2}{2}\sum_{0\leq k<j\leq 2n-1}D\boldsymbol{F}_j(\boldsymbol{z}_0)\boldsymbol{F}_k\boldsymbol{z}_0 + \frac{1}{2}\boldsymbol{\epsilon}_{2n} \tag{65}$$

where $\epsilon_{2n}$ is defined as in (34). Recall that $\boldsymbol{F}_i = \boldsymbol{F}_{2n-1-i}$ for all $i = 0, 1, \ldots, 2n-1$, and moreover, $\sum_{i=0}^{n-1} \boldsymbol{F}_i = \sum_{i=n}^{2n-1} \boldsymbol{F}_i = n\boldsymbol{F}$. Thus, the first sum in the above is equal to $2n\boldsymbol{F}\boldsymbol{z}_0$, and the second sum is equal to $2\sum_{j=0}^{n-1} D\boldsymbol{F}_j(\boldsymbol{z}_0)\boldsymbol{F}_j\boldsymbol{z}_0$. For the last sum, observe that

$$
\sum_{0 \le k < j \le 2n-1} D\boldsymbol{F}_j(\boldsymbol{z}_0)\boldsymbol{F}_k\boldsymbol{z}_0 = \sum_{0 \le k < j \le n-1} D\boldsymbol{F}_j(\boldsymbol{z}_0)\boldsymbol{F}_k\boldsymbol{z}_0 + \sum_{n \le k < j \le 2n-1} D\boldsymbol{F}_j(\boldsymbol{z}_0)\boldsymbol{F}_k\boldsymbol{z}_0
$$
$$
+ \sum_{\substack{0 \le k \le n-1 \\ n \le j \le 2n-1}} D\boldsymbol{F}_j(\boldsymbol{z}_0)\boldsymbol{F}_k\boldsymbol{z}_0
$$
$$
= \sum_{0 \le k < j \le n-1} D\boldsymbol{F}_j(\boldsymbol{z}_0)\boldsymbol{F}_k\boldsymbol{z}_0 + \sum_{n-1 \ge k > j \ge 0} D\boldsymbol{F}_j(\boldsymbol{z}_0)\boldsymbol{F}_k\boldsymbol{z}_0
$$
$$
+ \sum_{\substack{0 \le k \le n-1 \\ n-1 \ge j \ge 0}} D\boldsymbol{F}_j(\boldsymbol{z}_0)\boldsymbol{F}_k\boldsymbol{z}_0
$$
$$
= 2\sum_{k \ne j} D\boldsymbol{F}_j(\boldsymbol{z}_0)\boldsymbol{F}_k\boldsymbol{z}_0 + \sum_{j=0}^{n-1} D\boldsymbol{F}_j(\boldsymbol{z}_0)\boldsymbol{F}_j\boldsymbol{z}_0.
$$

Hence, (65) is equivalent to

$$
\boldsymbol{z}^\sharp = \boldsymbol{z}_0 - n\eta\boldsymbol{F}\boldsymbol{z}_0 + \frac{\eta^2}{2}\sum_{j=0}^{n-1} D\boldsymbol{F}_j(\boldsymbol{z}_0)\boldsymbol{F}_j\boldsymbol{z}_0 + \frac{\eta^2}{2}\sum_{0 \le k < j \le 2n-1} D\boldsymbol{F}_j(\boldsymbol{z}_0)\boldsymbol{F}_k\boldsymbol{z}_0 + \frac{1}{2}\boldsymbol{\epsilon}_{2n}
$$
$$
= \boldsymbol{z}_0 - n\eta\boldsymbol{F}\boldsymbol{z}_0 + \eta^2\sum_{j=0}^{n-1} D\boldsymbol{F}_j(\boldsymbol{z}_0)\boldsymbol{F}_j\boldsymbol{z}_0 + \eta^2\sum_{k \ne j} D\boldsymbol{F}_j(\boldsymbol{z}_0)\boldsymbol{F}_k\boldsymbol{z}_0 + \frac{1}{2}\boldsymbol{\epsilon}_{2n}
$$
$$
= \boldsymbol{z}_0 - n\eta\boldsymbol{F}\boldsymbol{z}_0 + \eta^2\left(\sum_{j=0}^{n-1} D\boldsymbol{F}_j(\boldsymbol{z}_0)\right)\left(\sum_{j=0}^{n-1}\boldsymbol{F}_j\boldsymbol{z}_0\right) + \frac{1}{2}\boldsymbol{\epsilon}_{2n}
$$
$$
= \boldsymbol{z}_0 - n\eta\boldsymbol{F}\boldsymbol{z}_0 + n^2\eta^2 D\boldsymbol{F}(\boldsymbol{z}_0)\boldsymbol{F}\boldsymbol{z}_0 + \frac{1}{2}\boldsymbol{\epsilon}_{2n}.
$$

Observing that the terms (64b), (64c), and (64d) add up to $\frac{1}{2}\boldsymbol{\epsilon}_{2n}$ completes the proof. $\qquad\square$

**Proposition D.10.** *Suppose that $\eta < \frac{1}{nL}$, and let $\nu := 1 + \frac{1}{2n}$. Then the noise term satisfies the bound*

$$
\|\boldsymbol{r}\| \le \eta^3 n^3 L^2 \|\boldsymbol{F}\boldsymbol{z}_0\| \left( \frac{1}{2n}\left(1 + \frac{2e^2}{3}\right) + \frac{4\nu + e^2}{3} \right)
$$
$$
+ \eta^3 n^3 M \|\boldsymbol{F}\boldsymbol{z}_0\|^2 \left( \frac{1}{2} + 4\nu^4 + \frac{16\nu e^4}{5} \right)
$$
$$
+ \frac{3\eta^3 M}{8}\left( \Psi_{2n} + (2\nu^2 - 1)^2 \Psi_{2n-1} + \frac{4\nu^6 e^4}{n^2}\sum_{j=1}^{2n-2} j\Psi_j \right)
$$
$$
+ \frac{\eta^3 L^2(\nu+1)}{4}\Sigma_{2n-1} + \frac{\eta^3 L^2\nu^2(1 + \eta L e^2)}{2}\sum_{j=1}^{2n-2}\Sigma_j + \frac{\eta^4 L^3\nu^3 e^2}{2}\sum_{k=1}^{2n-2}(2n-k-1)\Sigma_{k-1}.
$$

*Proof.* We bound each line in equation (64). For (64a), we use Lemma B.6 to get

$$
\left\| n\eta\boldsymbol{F}(\boldsymbol{z}_0 - n\eta\boldsymbol{F}\boldsymbol{z}_0) - n\eta\boldsymbol{F}\boldsymbol{z}_0 + n^2\eta^2 D\boldsymbol{F}(\boldsymbol{z}_0)\boldsymbol{F}\boldsymbol{z}_0 \right\| \le \frac{n\eta M}{2}\left\| -n\eta\boldsymbol{F}\boldsymbol{z}_0 \right\|^2
$$
$$
= \frac{n^3\eta^3 M}{2}\left\| \boldsymbol{F}\boldsymbol{z}_0 \right\|^2.
$$

In bounding the remaining three lines we repeatedly use the bounds obtained in Proposition D.2. We will also use the following bounds, which follows from (35), (37), and Young's inequality:

$$\|\boldsymbol{w}_0 - \boldsymbol{z}_0\| = \frac{\eta}{2}\|\boldsymbol{g}_1\| \le \frac{\eta}{2}\|\boldsymbol{F}\boldsymbol{z}_0\| + \frac{\eta}{2}\delta_1,$$

$$\|\boldsymbol{w}_0 - \boldsymbol{z}_0\|^2 = \frac{\eta^2}{4}\|\boldsymbol{g}_1\|^2 \le \frac{\eta^2}{2}\|\boldsymbol{F}\boldsymbol{z}_0\|^2 + \frac{\eta^2}{2}\delta_1^2.$$

For (64b), observe that Lemma B.6 gives us

$$\|\boldsymbol{F}_j\boldsymbol{w}_j - \boldsymbol{F}_j\boldsymbol{z}_0 - D\boldsymbol{F}_j(\boldsymbol{z}_0)(\boldsymbol{w}_j - \boldsymbol{z}_0)\| \le \frac{M}{2}\|\boldsymbol{w}_j - \boldsymbol{z}_0\|^2.$$

Thus, by using the bound obtained in Proposition D.2, we get

$$\left\|-\frac{\eta}{2}\sum_{j=0}^{2n-1}\Big(\boldsymbol{F}_j\boldsymbol{w}_j - \boldsymbol{F}_j\boldsymbol{z}_0 - D\boldsymbol{F}_j(\boldsymbol{z}_0)(\boldsymbol{w}_j - \boldsymbol{z}_0)\Big)\right\|$$

$$\le \frac{\eta}{2}\sum_{j=0}^{2n-1}\|\boldsymbol{F}_j\boldsymbol{w}_j - \boldsymbol{F}_j\boldsymbol{z}_0 - D\boldsymbol{F}_j(\boldsymbol{z}_0)(\boldsymbol{w}_j - \boldsymbol{z}_0)\|$$

$$\le \frac{\eta M}{4}\sum_{j=0}^{2n-1}\|\boldsymbol{w}_j - \boldsymbol{z}_0\|^2$$

$$\le \frac{\eta M}{4}\sum_{j=1}^{2n-1}\left(\frac{3\eta^2(j+1)^2}{2} + \frac{3\eta^2(2\nu^2-1)^2j^2}{2} + \frac{\eta^2\nu^6e^4 j(j-1)^2(2j-1)}{n^2}\right)\|\boldsymbol{F}\boldsymbol{z}_0\|^2$$

$$+ \frac{\eta M}{4}\sum_{j=1}^{2n-1}\left(\frac{3\eta^2}{2}\delta_{j+1}^2 + \frac{3\eta^2(2\nu^2-1)^2}{2}\delta_j^2 + \frac{6\eta^2\nu^6e^4(j-1)}{n^2}\Psi_{j-1}\right)$$

$$+ \frac{\eta M}{4}\|\boldsymbol{w}_0 - \boldsymbol{z}_0\|^2$$

$$= \frac{\eta M}{4}\left(\frac{\eta^2 n(1+2n)(1+4n) - 3\eta^2}{2} + \frac{\eta^2(2\nu^2-1)^2 n(2n-1)(4n-1)}{2}\right.$$

$$\left.+ \frac{\eta^2\nu^6e^4(n-1)(2n-1)(32n^2 - 42n + 11)}{5n}\right)\|\boldsymbol{F}\boldsymbol{z}_0\|^2$$

$$+ \frac{3\eta^3 M}{8}(\Psi_{2n} - \delta_1^2) + \frac{3\eta^3 M(2\nu^2-1)^2}{8}\Psi_{2n-1} + \frac{3\eta^3 M\nu^6e^4}{2n^2}\sum_{j=1}^{2n-1}(j-1)\Psi_{j-1}$$

$$+ \frac{\eta M}{4}\left(\frac{\eta^2}{2}\|\boldsymbol{F}\boldsymbol{z}_0\|^2 + \frac{\eta^2}{2}\delta_1^2\right)$$

$$\le \eta^3 n^3 M\left(\nu^2 + (2\nu^2-1)^2 + \frac{16\nu e^4}{5}\right)\|\boldsymbol{F}\boldsymbol{z}_0\|^2$$

$$+ \frac{3\eta^3 M}{8}\Psi_{2n} + \frac{3\eta^3 M(2\nu^2-1)^2}{8}\Psi_{2n-1} + \frac{3\eta^3 M\nu^6e^4}{2n^2}\sum_{j=1}^{2n-2}j\Psi_j$$

$$\le \eta^3 n^3 M\left(4\nu^4 + \frac{16\nu e^4}{5}\right)\|\boldsymbol{F}\boldsymbol{z}_0\|^2$$

$$+ \frac{3\eta^3 M}{8}\left(\Psi_{2n} + (2\nu^2-1)^2\Psi_{2n-1} + \frac{4\nu^6e^4}{n^2}\sum_{j=1}^{2n-2}j\Psi_j\right)$$

where along the derivation we used the inequality

$$\nu^5(n-1)(2n-1)(32n^2 - 42n + 11) \le 64n^4$$

which holds for all $n \ge 1$. From now on, we will keep on using similar techniques to reduce the exponents of $\nu$, without explicitly stating the inequalities used, but recovering the inequalities that are used should be clear from context.

For (64c), we use $L$-smoothness of $\boldsymbol{F}_j$, and also the fact that it implies $\|D\boldsymbol{F}_j(\boldsymbol{z}_0)\| \le L$, to get

$$
\left\| \frac{\eta^2}{4} \sum_{j=0}^{2n-1} D\boldsymbol{F}_j(\boldsymbol{z}_0)(\boldsymbol{F}_j\boldsymbol{z}_j - \boldsymbol{F}_j\boldsymbol{z}_0) \right\|
$$

$$
\le \frac{\eta^2}{4} \sum_{j=0}^{2n-1} \|D\boldsymbol{F}_j(\boldsymbol{z}_0)\| \, \|\boldsymbol{F}_j\boldsymbol{z}_j - \boldsymbol{F}_j\boldsymbol{z}_0\|
$$

$$
\le \frac{\eta^2 L^2}{4} \sum_{j=0}^{2n-1} \|\boldsymbol{z}_j - \boldsymbol{z}_0\|
$$

$$
\le \frac{\eta^2 L^2}{4} \sum_{j=1}^{2n-1} \left( \left( \eta\nu j + \frac{\eta\nu^2 e^2 j(j-1)}{2n} \right) \|\boldsymbol{F}\boldsymbol{z}_0\| + \eta\nu\delta_j + \eta^2 L\nu^2 e^2 \Sigma_{j-1} \right)
$$

$$
= \frac{\eta^2 L^2}{4} \left( \eta\nu n(2n-1) + \frac{2\eta\nu^2 e^2 (n-1)(2n-1)}{3} \right) \|\boldsymbol{F}\boldsymbol{z}_0\|
$$

$$
+ \frac{\eta^3 L^2 \nu}{4} \Sigma_{2n-1} + \frac{\eta^4 L^3 \nu^2 e^2}{4} \sum_{j=1}^{2n-1} \Sigma_{j-1}
$$

$$
\le \frac{\eta^3 n^2 L^2}{2} \left( 1 + \frac{2e^2}{3} \right) \|\boldsymbol{F}\boldsymbol{z}_0\| + \frac{\eta^3 L^2 \nu}{4} \Sigma_{2n-1} + \frac{\eta^4 L^3 \nu^2 e^2}{4} \sum_{j=1}^{2n-2} \Sigma_j.
$$

By the same logic, each summand in (64d) with $j > 0$ can be bounded as

$$
\left\| D\boldsymbol{F}_j(\boldsymbol{z}_0) \sum_{k=0}^{j-1} (\boldsymbol{F}_k\boldsymbol{w}_k - \boldsymbol{F}_k\boldsymbol{z}_0) \right\|
$$

$$
\le \|D\boldsymbol{F}_j(\boldsymbol{z}_0)\| \sum_{k=0}^{j-1} \|\boldsymbol{F}_k\boldsymbol{w}_k - \boldsymbol{F}_k\boldsymbol{z}_0\|
$$

$$
\le L^2 \sum_{k=0}^{j-1} \|\boldsymbol{w}_k - \boldsymbol{z}_0\|
$$

$$
\le L^2 \left( \frac{\eta}{2} \|\boldsymbol{F}\boldsymbol{z}_0\| + \frac{\eta}{2}\delta_1 \right)
$$

$$
+ L^2 \sum_{k=1}^{j-1} \frac{\eta}{2} \left( 1 + 2\nu^2 k + \frac{\nu^3 e^2 k(k-1)}{n} \right) \|\boldsymbol{F}\boldsymbol{z}_0\|
$$

$$
+ L^2 \sum_{k=1}^{j-1} \left( \frac{\eta}{2}\delta_{k+1} + \frac{\eta(2\nu^2 - 1)}{2}\delta_k + \eta^2 L\nu^3 e^2 \Sigma_{k-1} \right)
$$

$$
= \frac{\eta L^2}{2} (\|\boldsymbol{F}\boldsymbol{z}_0\| + \delta_1) + \frac{\eta L^2}{2} \left( j - 1 + \nu^2 j(j-1) + \frac{\nu^3 e^2 j(j-1)(j-2)}{3n} \right) \|\boldsymbol{F}\boldsymbol{z}_0\|
$$

$$
+ \frac{\eta L^2}{2} (\Sigma_j - \delta_1) + \frac{\eta L^2 (2\nu^2 - 1)}{2} \Sigma_{j-1} + \eta^2 L^3 \nu^3 e^2 \sum_{k=1}^{j-1} \Sigma_{k-1}
$$

$$
= \frac{\eta L^2}{2} \left( j + \nu^2 j(j-1) + \frac{\nu^3 e^2 j(j-1)(j-2)}{3n} \right) \|\boldsymbol{F}\boldsymbol{z}_0\|
$$

$$
+ \frac{\eta L^2}{2} \Sigma_j + \frac{\eta L^2 (2\nu^2 - 1)}{2} \Sigma_{j-1} + \eta^2 L^3 \nu^3 e^2 \sum_{k=1}^{j-1} \Sigma_{k-1},
$$

and when $j = 0$ the sum with respect to $k$ becomes an empty sum. Thus, (64d) in total satisfies the bound

$$
\left\| \frac{\eta^2}{2} \sum_{j=0}^{2n-1} D\boldsymbol{F}_j(\boldsymbol{z}_0) \sum_{k=0}^{j-1} (\boldsymbol{F}_k \boldsymbol{w}_k - \boldsymbol{F}_k \boldsymbol{z}_0) \right\|
$$

$$
\leq \frac{\eta^2}{2} \sum_{j=0}^{2n-1} \left\| D\boldsymbol{F}_j(\boldsymbol{z}_0) \sum_{k=0}^{j-1} (\boldsymbol{F}_k \boldsymbol{w}_k - \boldsymbol{F}_k \boldsymbol{z}_0) \right\|
$$

$$
\leq \frac{\eta^3 L^2}{4} \sum_{j=1}^{2n-1} \left( j + \nu^2 j(j-1) + \frac{\nu^3 e^2 j(j-1)(j-2)}{3n} \right) \|\boldsymbol{F}\boldsymbol{z}_0\|
$$

$$
+ \frac{\eta^2}{2} \sum_{j=1}^{2n-1} \left( \frac{\eta L^2}{2} \Sigma_j + \frac{\eta L^2 (2\nu^2 - 1)}{2} \Sigma_{j-1} + \eta^2 L^3 \nu^3 e^2 \sum_{k=1}^{j-1} \Sigma_{k-1} \right)
$$

$$
= \frac{\eta^3 L^2}{4} \left( n(2n-1) + \frac{4\nu^2 n(n-1)(2n-1)}{3} + \frac{\nu^3 e^2 (n-1)(2n-1)(2n-3)}{3} \right) \|\boldsymbol{F}\boldsymbol{z}_0\|
$$

$$
+ \frac{\eta^3 L^2}{4} \sum_{j=1}^{2n-1} \Sigma_j + \frac{\eta^3 L^2 (2\nu^2 - 1)}{4} \sum_{j=1}^{2n-1} \Sigma_{j-1} + \frac{\eta^4 L^3 \nu^3 e^2}{2} \sum_{j=1}^{2n-1} \sum_{k=1}^{j-1} \Sigma_{k-1}
$$

$$
\leq \frac{\eta^3 L^2}{2} \left( n^2 + \frac{4n^3}{3} + \frac{2e^2 n^3}{3} \right) \|\boldsymbol{F}\boldsymbol{z}_0\|
$$

$$
+ \frac{\eta^3 L^2}{4} \sum_{j=1}^{2n-1} \Sigma_j + \frac{\eta^3 L^2 (2\nu^2 - 1)}{4} \sum_{j=1}^{2n-2} \Sigma_j + \frac{\eta^4 L^3 \nu^3 e^2}{2} \sum_{k=1}^{2n-2} \sum_{j=k+1}^{2n-1} \Sigma_{k-1}
$$

$$
\leq \eta^3 n^3 L^2 \left( \frac{4\nu + e^2}{3} \right) \|\boldsymbol{F}\boldsymbol{z}_0\|
$$

$$
+ \frac{\eta^3 L^2}{4} \Sigma_{2n-1} + \frac{\eta^3 L^2 \nu^2}{2} \sum_{j=1}^{2n-2} \Sigma_j + \frac{\eta^4 L^3 \nu^3 e^2}{2} \sum_{k=1}^{2n-2} (2n - k - 1) \Sigma_{k-1}.
$$

Simply collecting all the inequalities and rearranging the terms leads to the claimed bound. $\qquad\square$

Before we proceed, let us write

$$
X_1 := \frac{3\eta^3 M}{8} \left( \Psi_{2n} + (2\nu^2 - 1)^2 \Psi_{2n-1} + \frac{4\nu^6 e^4}{n^2} \sum_{j=1}^{2n-2} j\Psi_j \right), \tag{66}
$$

$$
X_2 := \frac{\eta^3 L^2 (\nu + 1)}{4} \Sigma_{2n-1} + \frac{\eta^3 L^2 \nu^2 (1 + \eta Le^2)}{2} \sum_{j=1}^{2n-2} \Sigma_j + \frac{\eta^4 L^3 \nu^3 e^2}{2} \sum_{k=1}^{2n-2} (2n - k - 1) \Sigma_{k-1} \tag{67}
$$

so that the bound on $\|\boldsymbol{r}\|$ obtained in Proposition D.10 can be written as

$$
\|\boldsymbol{r}\| \leq \eta^3 n^3 L^2 \|\boldsymbol{F}\boldsymbol{z}_0\| \left( \frac{1}{2n} \left( 1 + \frac{2e^2}{3} \right) + \frac{4\nu + e^2}{3} \right)
$$
$$
+ \eta^3 n^3 M \|\boldsymbol{F}\boldsymbol{z}_0\|^2 \left( \frac{1}{2} + 4\nu^4 + \frac{16\nu e^4}{5} \right) \tag{68}
$$
$$
+ X_1 + X_2.
$$

**Theorem D.11.** *Suppose that* $\eta < \frac{1}{nL}$, *and let* $\nu := 1 + \frac{1}{2n}$. *Then the noise term* deterministically *satisfies the bound*

$$\|\boldsymbol{r}\| \leq \eta^3 n^3 L^2 \|\boldsymbol{F}\boldsymbol{z}_0\| \left( \frac{1}{2n} \left(1 + \frac{2e^2}{3}\right) + \frac{4\nu + e^2}{3} + 10\nu\rho \right)$$

$$+ \eta^3 n^3 M \|\boldsymbol{F}\boldsymbol{z}_0\|^2 \left( \frac{1}{2} + 4\nu^4 + \frac{16\nu e^4}{5} + \rho^2 \left(3\nu^4 + 8\nu^3 e^4\right) \right)$$

$$+ \eta^3 n^3 M \sigma^2 \left(3\nu^4 + 8\nu^3 e^4\right) + 10\nu\eta^3 n^3 L^2 \sigma.$$

*Proof.* From (38), (39), and Lemma D.4, it holds that

$$\Sigma_j = \sum_{i=1}^{j} \delta_i \leq jn(\rho \|\boldsymbol{F}\boldsymbol{z}_0\| + \sigma), \tag{69}$$

$$\Psi_j = \sum_{i=1}^{j} \delta_i^2 \leq jn^2(\rho \|\boldsymbol{F}\boldsymbol{z}_0\| + \sigma)^2. \tag{70}$$

Plugging the bound for $\Psi_j$ into (66) we get

$$X_1 \leq \frac{3\eta^3 M}{8} \left( 2n^3(\rho \|\boldsymbol{F}\boldsymbol{z}_0\| + \sigma)^2 + (2\nu^2 - 1)^2(2n-1)n^2(\rho \|\boldsymbol{F}\boldsymbol{z}_0\| + \sigma)^2 + 4\nu^6 e^4 \sum_{j=1}^{2n-2} j^2(\rho \|\boldsymbol{F}\boldsymbol{z}_0\| + \sigma)^2 \right)$$

$$= \frac{3\eta^3 M}{8} \left( \left(2n^3 + (2\nu^2 - 1)^2(2n-1)n^2\right)(\rho \|\boldsymbol{F}\boldsymbol{z}_0\| + \sigma)^2 + \frac{4\nu^6 e^4}{3}(n-1)(2n-1)(4n-3)(\rho \|\boldsymbol{F}\boldsymbol{z}_0\| + \sigma)^2 \right)$$

$$\leq \frac{3\eta^3 M}{8} \left( 4\nu^4 n^3(\rho \|\boldsymbol{F}\boldsymbol{z}_0\| + \sigma)^2 + \frac{32\nu^3 e^4 n^3}{3}(\rho \|\boldsymbol{F}\boldsymbol{z}_0\| + \sigma)^2 \right)$$

$$= \frac{\eta^3 n^3 M(\rho \|\boldsymbol{F}\boldsymbol{z}_0\| + \sigma)^2}{2} \left(3\nu^4 + 8\nu^3 e^4\right).$$

By Young's inequality, it holds that

$$\frac{(\rho \|\boldsymbol{F}\boldsymbol{z}_0\| + \sigma)^2}{2} \leq \rho^2 \|\boldsymbol{F}\boldsymbol{z}_0\|^2 + \sigma^2,$$

from which we get

$$X_1 \leq \eta^3 n^3 M \rho^2 \|\boldsymbol{F}\boldsymbol{z}_0\|^2 \left(3\nu^4 + 8\nu^3 e^4\right) + \eta^3 n^3 M \sigma^2 \left(3\nu^4 + 8\nu^3 e^4\right). \tag{71}$$

Meanwhile, plugging the bound for $\Sigma_j$ into (67) we get

$$X_2 \leq \frac{\eta^3 L^2(\nu+1)}{4}(2n-1)n(\rho \|\boldsymbol{F}\boldsymbol{z}_0\| + \sigma) + \frac{\eta^3 L^2\nu^2(1+\eta Le^2)}{2} \sum_{j=1}^{2n-2} jn(\rho \|\boldsymbol{F}\boldsymbol{z}_0\| + \sigma)$$

$$+ \frac{\eta^4 L^3\nu^3 e^2}{2} \sum_{k=1}^{2n-2} (2n-k-1)(k-1)n(\rho \|\boldsymbol{F}\boldsymbol{z}_0\| + \sigma)$$

$$= \frac{\eta^3 L^2(\nu+1)}{4}(2n-1)n(\rho \|\boldsymbol{F}\boldsymbol{z}_0\| + \sigma) + \frac{\eta^3 L^2\nu^2(1+\eta Le^2)}{2}(n-1)(2n-1)n(\rho \|\boldsymbol{F}\boldsymbol{z}_0\| + \sigma)$$

$$+ \frac{\eta^4 L^3\nu^3 e^2}{6} \left(-3 + 11n - 12n^2 + 4n^3\right) n(\rho \|\boldsymbol{F}\boldsymbol{z}_0\| + \sigma)$$

$$\leq \eta^3 L^2 \left( n^2(\rho \|\boldsymbol{F}\boldsymbol{z}_0\| + \sigma) + (1+\eta Le^2)n^3(\rho \|\boldsymbol{F}\boldsymbol{z}_0\| + \sigma) + \frac{2\eta Le^2}{3}n^4(\rho \|\boldsymbol{F}\boldsymbol{z}_0\| + \sigma) \right)$$

$$\leq \eta^3 n^3 L^2(\rho \|\boldsymbol{F}\boldsymbol{z}_0\| + \sigma) \left( \frac{1}{n} + 1 + \frac{e^2}{n} + \frac{2e^2}{3} \right)$$

where in the last line we used that $\eta < \frac{1}{nL}$. Because the inequality

$$\frac{1}{n} + 1 + \frac{e^2}{n} + \frac{2e^2}{3} \leq 10\nu$$

holds for all $n \geq 1$, continuing from above we obtain

$$
\begin{aligned}
X_2 &\leq 10\nu\eta^3 n^3 L^2 (\rho \|\boldsymbol{F}\boldsymbol{z}_0\| + \sigma) \\
&\leq 10\nu\eta^3 n^3 L^2 \rho \|\boldsymbol{F}\boldsymbol{z}_0\| + 10\nu\eta^3 n^3 L^2 \sigma.
\end{aligned}
\tag{72}
$$

Rearranging (68) with applying the bounds (71) and (72) gives us the claimed result. □

*Proof of Theorem D.8.* As $n \geq 1$, we notice that $1/n \leq 1$ and $\nu \leq 3/2$. Then the bound (60) is immediate from Theorem D.11. □

### D.2.2 Proof of Equation (20) for SEG-FFA

In this section, we prove the following.

**Theorem D.12.** *Say we use SEG-FFA. Then, as long as the stepsize used in an epoch satisfies $\eta < \frac{1}{nL}$, it holds that*

$$
\mathbb{E}\left[\|\boldsymbol{r}\|^2 \mid \boldsymbol{z}_0\right] \leq \eta^6 n^6 C_{2A} \|\boldsymbol{F}\boldsymbol{z}_0\|^2 + \eta^6 n^6 D_{2A} \|\boldsymbol{F}\boldsymbol{z}_0\|^4 + \eta^6 n^5 V_{2A}
\tag{73}
$$

*for constants*

$$
C_{2A} := 4L^4 \left( \left( \frac{1}{2}\left(1 + \frac{2e^2}{3}\right) + \frac{6 + e^2}{3} \right)^2 + 36\rho^2 e^4 \right),
\tag{74}
$$

$$
D_{2A} := 4M^2 \left( \left( \frac{83}{4} + \frac{24e^4}{5} \right)^2 + \rho^4 \left( \frac{243}{16} + 27e^4 \right)^2 \right),
\tag{75}
$$

$$
V_{2A} := 4M^2 \sigma^4 \left( \frac{243}{16} + 27e^4 \right)^2 + 144 e^4 L^4 \sigma^2.
\tag{76}
$$

*Proof.* As $n \geq 1$, we notice that $1/n \leq 1$ and $\nu \leq 3/2$. The bound is then immediate from the following Theorem D.13. □

**Theorem D.13.** *Suppose that $\eta < \frac{1}{nL}$, and let $\nu := 1 + \frac{1}{2n}$. Then,* in expectation, *the noise term satisfies the bound*

$$
\begin{aligned}
\mathbb{E}\left[\|\boldsymbol{r}\|^2 \mid \boldsymbol{z}_0\right] &\leq 4\eta^6 n^6 L^4 \|\boldsymbol{F}\boldsymbol{z}_0\|^2 \left( \left( \frac{1}{2n}\left(1 + \frac{2e^2}{3}\right) + \frac{4\nu + e^2}{3} \right)^2 + \frac{36\rho^2 e^4}{n} \right) \\
&\quad + 4\eta^6 n^6 M^2 \|\boldsymbol{F}\boldsymbol{z}_0\|^4 \left( \left( \frac{1}{2} + 4\nu^4 + \frac{16\nu e^4}{5} \right)^2 + \frac{\rho^4 \left(3\nu^4 + 8\nu^3 e^4\right)^2}{n} \right) \\
&\quad + 4\eta^6 n^5 M^2 \sigma^4 \left(3\nu^4 + 8\nu^3 e^4\right)^2 + 144 e^4 \eta^6 n^5 L^4 \sigma^2.
\end{aligned}
$$

*Proof.* Notice that, when conditioned on $\boldsymbol{z}_0$, the only source of randomness included in $\Psi_j$ is the random permutation $\tau$ selected for the epoch. Hence, we can use Lemma D.5 to get

$$
\mathbb{E}\left[\Psi_j \mid \boldsymbol{z}_0\right] = \mathbb{E}\left[\sum_{i=1}^{j} \delta_i^2 \,\Big|\, \boldsymbol{z}_0\right] = \sum_{i=1}^{j} \mathbb{E}\left[\delta_i^2 \mid \boldsymbol{z}_0\right] \leq \frac{jn(\rho \|\boldsymbol{F}\boldsymbol{z}_0\| + \sigma)^2}{2}.
$$

Applying Young's inequality on (68) we get

$$
\begin{aligned}
\|\boldsymbol{r}\|^2 &\leq 4\eta^6 n^6 L^4 \|\boldsymbol{F}\boldsymbol{z}_0\|^2 \left( \frac{1}{2n}\left(1 + \frac{2e^2}{3}\right) + \frac{4\nu + e^2}{3} \right)^2 \\
&\quad + 4\eta^6 n^6 M^2 \|\boldsymbol{F}\boldsymbol{z}_0\|^4 \left( \frac{1}{2} + 4\nu^4 + \frac{16\nu e^4}{5} \right)^2 \\
&\quad + 4X_1^2 + 4X_2^2.
\end{aligned}
\tag{77}
$$

When conditioned on $\boldsymbol{z}_0$, the first two lines are not random quantities. Thus, it suffices to derive the bounds for $\mathbb{E}\left[X_i^2 \mid \boldsymbol{z}_0\right]$, $i = 1, 2$.

Recall that the bound (71) on $X_1$ holds *deterministically*. Hence, it holds that

$$
\mathbb{E}\left[X_1^2 \mid \boldsymbol{z}_0\right] \leq \mathbb{E}\left[X_1 \left(\eta^3 n^3 M \rho^2 \left\|\boldsymbol{F}\boldsymbol{z}_0\right\|^2 \left(3\nu^4 + 8\nu^3 e^4\right) + \eta^3 n^3 M \sigma^2 \left(3\nu^4 + 8\nu^3 e^4\right)\right) \,\Big|\, \boldsymbol{z}_0\right]
$$
$$
= \eta^3 n^3 M \left(3\nu^4 + 8\nu^3 e^4\right) \left(\rho^2 \left\|\boldsymbol{F}\boldsymbol{z}_0\right\|^2 + \sigma^2\right) \mathbb{E}\left[X_1 \mid \boldsymbol{z}_0\right].
$$

Now, to compute $\mathbb{E}\left[X_1 \mid \boldsymbol{z}_0\right]$, we apply the linearity of expectation on (66) to get

$$
\mathbb{E}\left[X_1 \mid \boldsymbol{z}_0\right] = \frac{3\eta^3 M}{8} \left(\mathbb{E}\left[\Psi_{2n} \mid \boldsymbol{z}_0\right] + (2\nu^2 - 1)^2 \mathbb{E}\left[\Psi_{2n-1} \mid \boldsymbol{z}_0\right] + \frac{4\nu^6 e^4}{n^2} \sum_{j=1}^{2n-2} j\, \mathbb{E}\left[\Psi_j \mid \boldsymbol{z}_0\right]\right)
$$
$$
\leq \frac{3\eta^3 M}{8} \left(n^2(\rho \left\|\boldsymbol{F}\boldsymbol{z}_0\right\| + \sigma)^2 + \frac{(2\nu^2 - 1)^2(2n-1)n\sigma^2}{2} + \frac{4\nu^6 e^4}{n^2} \sum_{j=1}^{2n-2} \frac{j^2 n(\rho \left\|\boldsymbol{F}\boldsymbol{z}_0\right\| + \sigma)^2}{2}\right)
$$
$$
= \frac{3\eta^3 M}{8} \left(\frac{2n^2 + (2\nu^2 - 1)^2(2n-1)n}{2}(\rho \left\|\boldsymbol{F}\boldsymbol{z}_0\right\| + \sigma)^2 + \frac{2\nu^6 e^4(n-1)(2n-1)(4n-3)(\rho \left\|\boldsymbol{F}\boldsymbol{z}_0\right\| + \sigma)^2}{3n}\right)
$$
$$
\leq \frac{3\eta^3 M}{8} \left(2\nu^4 n^2(\rho \left\|\boldsymbol{F}\boldsymbol{z}_0\right\| + \sigma)^2 + \frac{16\nu^3 e^4 n^2(\rho \left\|\boldsymbol{F}\boldsymbol{z}_0\right\| + \sigma)^2}{3}\right)
$$
$$
= \frac{\eta^3 n^2 M(\rho \left\|\boldsymbol{F}\boldsymbol{z}_0\right\| + \sigma)^2}{4} \left(3\nu^4 + 8\nu^3 e^4\right).
$$

Young's inequality gives us the bound

$$
\frac{(\rho \left\|\boldsymbol{F}\boldsymbol{z}_0\right\| + \sigma)^2}{2} \leq \rho^2 \left\|\boldsymbol{F}\boldsymbol{z}_0\right\|^2 + \sigma^2 \tag{78}
$$

which, with the inequality derived above, leads to

$$
\mathbb{E}\left[X_1 \mid \boldsymbol{z}_0\right] \leq \frac{\eta^3 n^2 M}{2} \left(3\nu^4 + 8\nu^3 e^4\right) \left(\rho^2 \left\|\boldsymbol{F}\boldsymbol{z}_0\right\|^2 + \sigma^2\right).
$$

As a consequence, with using Young's inequality once again, we obtain

$$
\mathbb{E}\left[X_1^2 \mid \boldsymbol{z}_0\right] \leq \frac{\eta^6 n^5 M^2}{2} \left(3\nu^4 + 8\nu^3 e^4\right)^2 \left(\rho^2 \left\|\boldsymbol{F}\boldsymbol{z}_0\right\|^2 + \sigma^2\right)^2
$$
$$
\leq \eta^6 n^5 M^2 \left(3\nu^4 + 8\nu^3 e^4\right)^2 \left(\rho^4 \left\|\boldsymbol{F}\boldsymbol{z}_0\right\|^4 + \sigma^4\right). \tag{79}
$$

To get the bound of $\mathbb{E}\left[X_2^2 \mid \boldsymbol{z}_0\right]$, we begin by using

$$
\eta L \nu^2 (2n - k - 1) \leq \left(1 + \frac{1}{2n}\right)^2 \frac{2n - k - 1}{n}
$$
$$
= -\frac{k}{4n^3} - \frac{k}{n^2} - \frac{k}{n} - \frac{1}{4n^3} - \frac{1}{2n^2} + \frac{1}{n} + 2 \leq 2,
$$

which holds for all $1 \leq k \leq 2n - 2$, to (67) to obtain

$$
X_2 \leq \frac{\eta^3 L^2(\nu + 1)}{4} \Sigma_{2n-1} + \frac{\eta^3 L^2 \nu^2(1 + \eta L e^2)}{2} \sum_{j=1}^{2n-2} \Sigma_j + \frac{\eta^3 L^2 \nu^3 e^2}{2} \sum_{k=1}^{2n-2} \frac{2n - k - 1}{n} \Sigma_{k-1}
$$
$$
\leq \frac{\eta^3 L^2(\nu + 1)}{4} \Sigma_{2n-1} + \frac{\eta^3 L^2 \nu^2(1 + \eta L e^2)}{2} \sum_{j=1}^{2n-2} \Sigma_j + \eta^3 L^2 \nu e^2 \sum_{k=1}^{2n-2} \Sigma_{k-1}
$$
$$
\leq \frac{\eta^3 L^2(\nu + 1)}{4} \Sigma_{2n-1} + \left(\frac{\eta^3 L^2 \nu^2(1 + \eta L e^2)}{2} + \eta^3 L^2 \nu e^2\right) \sum_{j=1}^{2n-2} \Sigma_j
$$
$$
\leq \frac{\eta^3 L^2(\nu + 1)}{4} \Sigma_{2n-1} + 3\eta^3 L^2 e^2 \sum_{j=1}^{2n-2} \Sigma_j.
$$

Then we directly square both sides and expand them to get

$$X_2^2 \leq \left( \frac{\eta^3 L^2 (\nu + 1)}{4} \Sigma_{2n-1} + 3\eta^3 L^2 e^2 \sum_{j=1}^{2n-2} \Sigma_j \right)^2$$

$$= \frac{\eta^6 L^4 (\nu+1)^2}{16} \Sigma_{2n-1}^2 + 9\eta^6 L^4 e^4 \left( \sum_{j=1}^{2n-2} \Sigma_j \right)^2 + \frac{3\eta^6 L^4 e^2 (\nu+1)}{2} \sum_{j=1}^{2n-2} \Sigma_{2n-1} \Sigma_j.$$

Here, using Lemma D.6 and Lemma D.7 on the right hand side leads to

$$\mathbb{E}\left[ X_2^2 \,\big|\, \boldsymbol{z}_0 \right] \leq \frac{\eta^6 L^4 (\nu+1)^2 n (2n-1)^2 (\rho \|\boldsymbol{F}\boldsymbol{z}_0\| + \sigma)^2}{32} + \frac{9\eta^6 L^4 e^4 n (2n-2)^2 (2n-1)^2 (\rho \|\boldsymbol{F}\boldsymbol{z}_0\| + \sigma)^2}{8}$$

$$+ \frac{3\eta^6 L^4 e^2 (\nu+1)}{2} \sum_{j=1}^{2n-2} \frac{jn(2n-1)(\rho \|\boldsymbol{F}\boldsymbol{z}_0\| + \sigma)^2}{2}$$

$$\leq \frac{\eta^6 L^4 (\nu+1)^2 n (2n-1)^2 (\rho \|\boldsymbol{F}\boldsymbol{z}_0\| + \sigma)^2}{32} + \frac{9\eta^6 L^4 e^4 n (2n-2)^2 (2n-1)^2 (\rho \|\boldsymbol{F}\boldsymbol{z}_0\| + \sigma)^2}{8}$$

$$+ \frac{3\eta^6 L^4 e^2 (\nu+1) n (n-1)(2n-1)^2 (\rho \|\boldsymbol{F}\boldsymbol{z}_0\| + \sigma)^2}{4}$$

$$\leq \frac{\eta^6 L^4 n^3 (\rho \|\boldsymbol{F}\boldsymbol{z}_0\| + \sigma)^2}{2} + \frac{9\eta^6 L^4 e^4 n (2n-2)^2 (2n-1)^2 (\rho \|\boldsymbol{F}\boldsymbol{z}_0\| + \sigma)^2}{8}$$

$$+ 6\eta^6 L^4 e^2 n^3 (n-1)(\rho \|\boldsymbol{F}\boldsymbol{z}_0\| + \sigma)^2$$

$$= \eta^6 L^4 (\rho \|\boldsymbol{F}\boldsymbol{z}_0\| + \sigma)^2 \left( \frac{n^3}{2} + \frac{9 e^4 n (2n-2)^2 (2n-1)^2}{8} + 6 e^2 n^3 (n-1) \right)$$

$$\leq 18 e^4 \eta^6 L^4 n^5 (\rho \|\boldsymbol{F}\boldsymbol{z}_0\| + \sigma)^2.$$

As a consequence, with using (78) once again, we obtain

$$\mathbb{E}\left[ X_2^2 \,\big|\, \boldsymbol{z}_0 \right] \leq 36 e^4 \eta^6 L^4 n^5 \left( \rho^2 \|\boldsymbol{F}\boldsymbol{z}_0\|^2 + \sigma^2 \right). \tag{80}$$

Taking the conditional expectation on (77), applying the bounds (79) and (80), and then rearranging the terms leads to the claimed inequality. $\qquad \square$

### D.2.3 Upper Bounds of the Within-Epoch Errors for SEG-FF

**Theorem D.14.** *Say we use* SEG-FF *with* $\alpha = \beta = \eta/2$. *Then, as long as the stepsize used in an epoch satisfies* $\eta < \frac{1}{nL}$, *it holds that*

$$\|\boldsymbol{r}\| \leq \eta^2 n^2 C_{1F} \|\boldsymbol{F}\boldsymbol{z}_0\| + \eta^2 n^2 D_{1F} \|\boldsymbol{F}\boldsymbol{z}_0\|^2 + \eta^2 n^2 V_{1F}$$

$$\mathbb{E}\left[ \|\boldsymbol{r}\|^2 \,\Big|\, \boldsymbol{z}_0 \right] \leq \eta^4 n^4 C_{2F} \|\boldsymbol{F}\boldsymbol{z}_0\|^2 + \eta^4 n^4 D_{2F} \|\boldsymbol{F}\boldsymbol{z}_0\|^4 + \eta^4 n^3 V_{2F}$$

*for constants* $C_{1F}$, $D_{1F}$, $V_{1F}$, $C_{2F}$, $D_{2F}$, *and* $V_{2F}$ *to be determined later in* (83) *and* (84).

*Proof.* As we have discussed in Section 5.1, we already know that aiming to achieve $\mathcal{O}(\eta^3)$ error without anchoring is futile. Instead, we show that error of magnitude $\mathcal{O}(\eta^2)$ is possible with the chosen stepsizes.

By Proposition C.2 and Lemma C.4 we have For any $i = 0, 1, \ldots, N$, it holds that

$$\boldsymbol{z}_{2n} = \boldsymbol{z}_0 - \frac{\eta}{2} \sum_{j=0}^{2n-1} \boldsymbol{T}_j \boldsymbol{z}_0 + \frac{\eta^2}{4} \sum_{j=0}^{2n-1} D\boldsymbol{T}_j(\boldsymbol{z}_0) \boldsymbol{T}_j \boldsymbol{z}_0 + \frac{\eta^2}{4} \sum_{0 \leq k < j \leq 2n-1} D\boldsymbol{T}_j(\boldsymbol{z}_0) \boldsymbol{T}_k \boldsymbol{z}_0 + \boldsymbol{\epsilon}_{2n}$$

$$= \boldsymbol{z}_0 - \eta \sum_{j=0}^{n-1} \boldsymbol{F}_j \boldsymbol{z}_0 + \frac{3\eta^2}{4} \sum_{j=1}^{n} D\boldsymbol{F}_j(\boldsymbol{z}_0) \boldsymbol{F}_j \boldsymbol{z}_0 + \frac{\eta^2}{2} \sum_{i \neq j} D\boldsymbol{F}_j(\boldsymbol{z}_0) \boldsymbol{F}_i \boldsymbol{z}_0 + \boldsymbol{\epsilon}_{2n}$$

$$= \boldsymbol{z}_0 - \eta n \boldsymbol{F} \boldsymbol{z}_0 + \eta^2 n^2 D\boldsymbol{F}(\boldsymbol{z}_0) \boldsymbol{F} \boldsymbol{z}_0 - \frac{\eta^2}{4} \sum_{j=1}^{n} D\boldsymbol{F}_j(\boldsymbol{z}_0) \boldsymbol{F}_j \boldsymbol{z}_0 - \frac{\eta^2}{2} \sum_{i \neq j} D\boldsymbol{F}_j(\boldsymbol{z}_0) \boldsymbol{F}_i \boldsymbol{z}_0 + \boldsymbol{\epsilon}_{2n}$$

where we denote

$$
\begin{aligned}
\boldsymbol{\epsilon}_{2n} &:= -\frac{\eta}{2} \sum_{j=0}^{2n-1} \Big( \boldsymbol{F}_j \boldsymbol{w}_j - \boldsymbol{F}_j \boldsymbol{z}_0 - D\boldsymbol{F}_j(\boldsymbol{z}_0)(\boldsymbol{w}_j - \boldsymbol{z}_0) \Big) \\
&\quad + \frac{\eta^2}{4} \sum_{j=0}^{2n-1} D\boldsymbol{F}_j(\boldsymbol{z}_0)(\boldsymbol{F}_j \boldsymbol{z}_j - \boldsymbol{F}_j \boldsymbol{z}_0) + \frac{\eta^2}{4} \sum_{j=0}^{2n-1} D\boldsymbol{F}_j(\boldsymbol{z}_0) \sum_{k=0}^{j-1} (\boldsymbol{F}_k \boldsymbol{w}_k - \boldsymbol{F}_k \boldsymbol{z}_0).
\end{aligned}
\tag{81}
$$

Comparing $\boldsymbol{z}_{2n}$ to a point that would have been the result of a deterministic EG update with stepsize $\eta n$ we get

$$
\begin{aligned}
\boldsymbol{z}_{2n} - (\boldsymbol{z}_0 - \eta n \boldsymbol{F}(\boldsymbol{z}_0 - \eta n \boldsymbol{F} \boldsymbol{z}_0)) = {}& \eta n \boldsymbol{F}(\boldsymbol{z}_0 - \eta n \boldsymbol{F} \boldsymbol{z}_0) - \eta n \boldsymbol{F} \boldsymbol{z}_0 + \eta^2 n^2 D\boldsymbol{F}(\boldsymbol{z}_0) \boldsymbol{F} \boldsymbol{z}_0 + \boldsymbol{\epsilon}_{2n} \\
& - \frac{\eta^2}{4} \sum_{j=1}^{n} D\boldsymbol{F}_j(\boldsymbol{z}_0) \boldsymbol{F}_j \boldsymbol{z}_0 - \frac{\eta^2}{2} \sum_{i \neq j} D\boldsymbol{F}_j(\boldsymbol{z}_0) \boldsymbol{F}_i \boldsymbol{z}_0.
\end{aligned}
$$

Let us define

$$
\tilde{\boldsymbol{r}} := \eta n \boldsymbol{F}(\boldsymbol{z}_0 - \eta n \boldsymbol{F} \boldsymbol{z}_0) - \eta n \boldsymbol{F} \boldsymbol{z}_0 + \eta^2 n^2 D\boldsymbol{F}(\boldsymbol{z}_0) \boldsymbol{F} \boldsymbol{z}_0 + \boldsymbol{\epsilon}_{2n}.
\tag{82}
$$

Noticing the resemblence between (64) and the equations in (81) and (82), we can repeat the same reasoning used for Theorem D.8 and Theorem D.12, but with replacing the bounds given by Proposition D.2 to those in Proposition D.3 (and plugging in $\eta/2$ in place of $\eta$ in the statement of Proposition D.3) to conclude that

$$
\|\tilde{\boldsymbol{r}}\| \leq \eta^3 n^3 \tilde{C}_{\mathsf{1A}} \|\boldsymbol{F} \boldsymbol{z}_0\| + \eta^3 n^3 \tilde{D}_{\mathsf{1A}} \|\boldsymbol{F} \boldsymbol{z}_0\|^2 + \eta^3 n^3 \tilde{V}_{\mathsf{1A}}
$$

$$
\mathbb{E}\left[ \|\tilde{\boldsymbol{r}}\|^2 \,\Big|\, \boldsymbol{z}_0 \right] \leq \eta^6 n^6 \tilde{C}_{\mathsf{2A}} \|\boldsymbol{F} \boldsymbol{z}_0\|^2 + \eta^6 n^6 \tilde{D}_{\mathsf{2A}} \|\boldsymbol{F} \boldsymbol{z}_0\|^4 + \eta^6 n^5 \tilde{V}_{\mathsf{2A}}
$$

for some constants $\tilde{C}_{\mathsf{1A}}$, $\tilde{D}_{\mathsf{1A}}$, $\tilde{V}_{\mathsf{1A}}$, $\tilde{C}_{\mathsf{2A}}$, $\tilde{D}_{\mathsf{2A}}$, and $\tilde{V}_{\mathsf{2A}}$. Meanwhile, we also have

$$
\begin{aligned}
\left\| \frac{\eta^2}{4} \sum_{j=1}^{n} D\boldsymbol{F}_j(\boldsymbol{z}_0) \boldsymbol{F}_j \boldsymbol{z}_0 \right. & \left. + \frac{\eta^2}{2} \sum_{i \neq j} D\boldsymbol{F}_j(\boldsymbol{z}_0) \boldsymbol{F}_i \boldsymbol{z}_0 \right\| \\
&= \left\| \frac{\eta^2 n^2}{2} D\boldsymbol{F}(\boldsymbol{z}_0) \boldsymbol{F} \boldsymbol{z}_0 - \frac{\eta^2}{4} \sum_{j=1}^{n} D\boldsymbol{F}_j(\boldsymbol{z}_0) \boldsymbol{F}_j \boldsymbol{z}_0 \right\| \\
&\leq \frac{\eta^2 n^2}{2} \|D\boldsymbol{F}(\boldsymbol{z}_0)\| \|\boldsymbol{F} \boldsymbol{z}_0\| + \frac{\eta^2}{4} \sum_{j=1}^{n} \|D\boldsymbol{F}_j(\boldsymbol{z}_0)\| \|\boldsymbol{F}_j \boldsymbol{z}_0\| \\
&\leq \frac{\eta^2 n^2}{2} L \|\boldsymbol{F} \boldsymbol{z}_0\| + \frac{\eta^2}{4} \sum_{j=1}^{n} L \left( \|\boldsymbol{F}_j \boldsymbol{z}_0 - \boldsymbol{F} \boldsymbol{z}_0\| + \|\boldsymbol{F} \boldsymbol{z}_0\| \right) \\
&\leq \frac{\eta^2 (n^2 + n) L}{2} \|\boldsymbol{F} \boldsymbol{z}_0\| + \frac{\eta^2 L}{4} \sum_{j=1}^{n} \|\boldsymbol{F}_j \boldsymbol{z}_0 - \boldsymbol{F} \boldsymbol{z}_0\| \\
&\leq \eta^2 n^2 L \|\boldsymbol{F} \boldsymbol{z}_0\| + \frac{\eta^2 L}{4} \left( \sum_{j=1}^{n} \|\boldsymbol{F}_j \boldsymbol{z}_0 - \boldsymbol{F} \boldsymbol{z}_0\|^2 \right)^{1/2} \left( \sum_{j=1}^{n} 1 \right)^{1/2} \\
&= \eta^2 n^2 L \|\boldsymbol{F} \boldsymbol{z}_0\| + \frac{\eta^2 n L}{4} (\rho \|\boldsymbol{F} \boldsymbol{z}_0\| + \sigma)
\end{aligned}
$$

where in the second to the last line we used the Cauchy-Schwarz inequality. Therefore, as $\eta \leq 1/nL$, we conclude that

$$
\|\boldsymbol{z}_{2n} - (\boldsymbol{z}_0 - \eta n \boldsymbol{F}(\boldsymbol{z}_0 - \eta n \boldsymbol{F} \boldsymbol{z}_0))\| \leq \eta^2 n^2 C_{\mathsf{1F}} \|\boldsymbol{F} \boldsymbol{z}_0\| + \eta^2 n^2 D_{\mathsf{1F}} \|\boldsymbol{F} \boldsymbol{z}_0\|^2 + \eta^2 n^2 V_{\mathsf{1F}}
$$

for constants

$$
C_{\mathsf{1F}} = L + \frac{\rho L}{4} + \frac{\tilde{C}_{\mathsf{1A}}}{L}, \quad D_{\mathsf{1F}} = \frac{\tilde{D}_{\mathsf{1A}}}{L}, \quad V_{\mathsf{1F}} = \frac{\sigma L}{4} + \frac{\tilde{V}_{\mathsf{1A}}}{L}.
\tag{83}
$$

Moreover, using Young's inequality, we see that

$$\left\| \frac{\eta^2}{4} \sum_{j=1}^{n} D\boldsymbol{F}_j(\boldsymbol{z}_0) \boldsymbol{F}_j \boldsymbol{z}_0 + \frac{\eta^2}{2} \sum_{i \neq j} D\boldsymbol{F}_j(\boldsymbol{z}_0) \boldsymbol{F}_i \boldsymbol{z}_0 \right\|^2$$

$$\leq 3\eta^4 n^4 L^2 \left\| \boldsymbol{F}\boldsymbol{z}_0 \right\|^2 + \frac{3\eta^4 n^2 L^2}{16} \rho^2 \left\| \boldsymbol{F}\boldsymbol{z}_0 \right\|^2 + \frac{3\eta^4 n^2 L^2}{16} \sigma^2,$$

so we also conclude that

$$\mathbb{E} \left[ \left\| \boldsymbol{z}_{2n} - (\boldsymbol{z}_0 - \eta n \boldsymbol{F}(\boldsymbol{z}_0 - \eta n \boldsymbol{F}\boldsymbol{z}_0)) \right\|^2 \,\Big|\, \boldsymbol{z}_0 \right] \leq \eta^4 n^4 C_{\mathsf{2F}} \left\| \boldsymbol{F}\boldsymbol{z}_0 \right\|^2 + \eta^4 n^4 D_{\mathsf{2F}} \left\| \boldsymbol{F}\boldsymbol{z}_0 \right\|^4 + \eta^4 n^3 V_{\mathsf{2F}}$$

holds for constants

$$C_{\mathsf{2F}} = 6L^2 + \frac{3\rho^2 L^2}{8} + \frac{2\tilde{C}_{\mathsf{2A}}}{L^2}, \quad D_{\mathsf{2F}} = \frac{2\tilde{D}_{\mathsf{1A}}}{L^2}, \quad V_{\mathsf{2F}} = \frac{3\sigma^2 L^2}{8} + \frac{2\tilde{V}_{\mathsf{1A}}}{L^2}. \tag{84}$$

$\square$

### D.2.4 Upper Bounds of the Within-Epoch Errors for SEG-RR

**Theorem D.15.** *Say we use SEG-RR with $\alpha = \beta = \eta$. Then, as long as the stepsize used in an epoch satisfies $\eta < \frac{1}{nL}$, it holds that*

$$\| \boldsymbol{r} \| \leq \eta^2 n^2 C_{\mathsf{1R}} \| \boldsymbol{F}\boldsymbol{z}_0 \| + \eta^2 n^2 D_{\mathsf{1R}} \| \boldsymbol{F}\boldsymbol{z}_0 \|^2 + \eta^2 n^2 V_{\mathsf{1R}}$$

$$\mathbb{E} \left[ \| \boldsymbol{r} \|^2 \,\Big|\, \boldsymbol{z}_0 \right] \leq \eta^4 n^4 C_{\mathsf{2R}} \| \boldsymbol{F}\boldsymbol{z}_0 \|^2 + \eta^4 n^4 D_{\mathsf{2R}} \| \boldsymbol{F}\boldsymbol{z}_0 \|^4 + \eta^4 n^3 V_{\mathsf{2R}}$$

*for constants $C_{\mathsf{1R}}$, $D_{\mathsf{1R}}$, $V_{\mathsf{1R}}$, $C_{\mathsf{2R}}$, $D_{\mathsf{2R}}$, and $V_{\mathsf{2R}}$ to be determined later in* (88) *and* (89).

*Proof.* As we have discussed in Section 5.1, we already know that aiming to achieve $\mathcal{O}(\eta^3)$ error with only using random reshuffling is futile. Instead, we show that error of magnitude $\mathcal{O}(\eta^2)$ is possible with the chosen stepsizes.

By Proposition C.2 and Lemma C.4 we have For any $i = 0, 1, \ldots, N$, it holds that

$$\boldsymbol{z}_n = \boldsymbol{z}_0 - \eta \sum_{j=0}^{n-1} \boldsymbol{F}_j \boldsymbol{z}_0 + \eta^2 \sum_{j=0}^{n-1} D\boldsymbol{F}_j(\boldsymbol{z}_0) \boldsymbol{F}_j \boldsymbol{z}_0 + \eta^2 \sum_{0 \leq k < j \leq n-1} D\boldsymbol{F}_j(\boldsymbol{z}_0) \boldsymbol{F}_k \boldsymbol{z}_0 + \boldsymbol{\epsilon}_n$$

$$= \boldsymbol{z}_0 - \eta n \boldsymbol{F}\boldsymbol{z}_0 + \eta^2 n^2 D\boldsymbol{F}(\boldsymbol{z}_0) \boldsymbol{F}\boldsymbol{z}_0 - \eta^2 \sum_{0 \leq j < k \leq n-1} D\boldsymbol{F}_j(\boldsymbol{z}_0) \boldsymbol{F}_k \boldsymbol{z}_0 + \boldsymbol{\epsilon}_n$$

where we denote

$$\boldsymbol{\epsilon}_n := -\eta \sum_{j=0}^{n-1} \Big( \boldsymbol{F}_j \boldsymbol{w}_j - \boldsymbol{F}_j \boldsymbol{z}_0 - D\boldsymbol{F}_j(\boldsymbol{z}_0)(\boldsymbol{w}_j - \boldsymbol{z}_0) \Big)$$

$$+ \eta^2 \sum_{j=0}^{n-1} D\boldsymbol{F}_j(\boldsymbol{z}_0)(\boldsymbol{F}_j \boldsymbol{z}_j - \boldsymbol{F}_j \boldsymbol{z}_0) + \eta^2 \sum_{j=0}^{n-1} D\boldsymbol{F}_j(\boldsymbol{z}_0) \sum_{k=0}^{j-1} (\boldsymbol{F}_k \boldsymbol{w}_k - \boldsymbol{F}_k \boldsymbol{z}_0). \tag{85}$$

Comparing $\boldsymbol{z}_n$ to a point that would have been the result of a deterministic EG update with stepsize $\eta n$ we get

$$\boldsymbol{z}_n - (\boldsymbol{z}_0 - \eta n \boldsymbol{F}(\boldsymbol{z}_0 - \eta n \boldsymbol{F}\boldsymbol{z}_0)) = \eta n \boldsymbol{F}(\boldsymbol{z}_0 - \eta n \boldsymbol{F}\boldsymbol{z}_0) - \eta n \boldsymbol{F}\boldsymbol{z}_0 + \eta^2 n^2 D\boldsymbol{F}(\boldsymbol{z}_0) \boldsymbol{F}\boldsymbol{z}_0 + \boldsymbol{\epsilon}_n$$

$$- \eta^2 \sum_{0 \leq j < k \leq n-1} D\boldsymbol{F}_j(\boldsymbol{z}_0) \boldsymbol{F}_k \boldsymbol{z}_0.$$

Let us define

$$\check{\boldsymbol{r}} := \eta n \boldsymbol{F}(\boldsymbol{z}_0 - \eta n \boldsymbol{F}\boldsymbol{z}_0) - \eta n \boldsymbol{F}\boldsymbol{z}_0 + \eta^2 n^2 D\boldsymbol{F}(\boldsymbol{z}_0) \boldsymbol{F}\boldsymbol{z}_0 + \boldsymbol{\epsilon}_n. \tag{86}$$

Comparing the sums (64b)–(64d) to (85), we can repeat the same reasoning used for Theorem D.8 and Theorem D.12, but with replacing the bounds given by Proposition D.2 to those in Proposition D.3, to conclude that

$$\| \check{\boldsymbol{r}} \| \leq \eta^3 n^3 \check{C}_{\mathsf{1A}} \| \boldsymbol{F}\boldsymbol{z}_0 \| + \eta^3 n^3 \check{D}_{\mathsf{1A}} \| \boldsymbol{F}\boldsymbol{z}_0 \|^2 + \eta^3 n^3 \check{V}_{\mathsf{1A}}$$

$$\mathbb{E} \left[ \| \check{\boldsymbol{r}} \|^2 \,\Big|\, \boldsymbol{z}_0 \right] \leq \eta^6 n^6 \check{C}_{\mathsf{2A}} \| \boldsymbol{F}\boldsymbol{z}_0 \|^2 + \eta^6 n^6 \check{D}_{\mathsf{2A}} \| \boldsymbol{F}\boldsymbol{z}_0 \|^4 + \eta^6 n^5 \check{V}_{\mathsf{2A}}$$

for some constants $\check{C}_{1\mathsf{A}}$, $\check{D}_{1\mathsf{A}}$, $\check{V}_{1\mathsf{A}}$, $\check{C}_{2\mathsf{A}}$, $\check{D}_{2\mathsf{A}}$, and $\check{V}_{2\mathsf{A}}$. Meanwhile, we also have

$$\sum_{0 \leq j < k \leq n-1} D\boldsymbol{F}_j(\boldsymbol{z}_0)\boldsymbol{F}_k\boldsymbol{z}_0 = \sum_{j=0}^{n-1} D\boldsymbol{F}_j(\boldsymbol{z}_0)(n\boldsymbol{F}\boldsymbol{z}_0 - \boldsymbol{g}_{j+1})$$

$$= \sum_{j=0}^{n-1}(n-j-1)D\boldsymbol{F}_j(\boldsymbol{z}_0)\boldsymbol{F}\boldsymbol{z}_0 - \sum_{j=0}^{n-1} D\boldsymbol{F}_j(\boldsymbol{z}_0)(\boldsymbol{g}_{j+1} - (j+1)\boldsymbol{F}\boldsymbol{z}_0)$$

which leads to

$$\left\| \sum_{0 \leq j < k \leq n-1} D\boldsymbol{F}_j(\boldsymbol{z}_0)\boldsymbol{F}_k\boldsymbol{z}_0 \right\| \leq \sum_{j=0}^{n-1}(n-j-1)L\|\boldsymbol{F}\boldsymbol{z}_0\| + L\sum_{j=0}^{n-1}\delta_{j+1}$$

$$\leq \frac{n^2 L}{2}\|\boldsymbol{F}\boldsymbol{z}_0\| + L\sum_{j=0}^{n-1}\delta_{j+1}. \tag{87}$$

Therefore, from $\eta \leq 1/nL$ and Lemma D.4, on one hand we obtain

$$\|\boldsymbol{z}_n - (\boldsymbol{z}_0 - \eta n\boldsymbol{F}(\boldsymbol{z}_0 - \eta n\boldsymbol{F}\boldsymbol{z}_0))\| \leq \eta^2 n^2 C_{1\mathsf{R}}\|\boldsymbol{F}\boldsymbol{z}_0\| + \eta^2 n^2 D_{1\mathsf{R}}\|\boldsymbol{F}\boldsymbol{z}_0\|^2 + \eta^2 n^2 V_{1\mathsf{R}}$$

for constants

$$C_{1\mathsf{R}} = \frac{L}{2} + \rho L + \frac{\check{C}_{1\mathsf{A}}}{L}, \quad D_{1\mathsf{R}} = \frac{\check{D}_{1\mathsf{A}}}{L}, \quad V_{1\mathsf{R}} = \sigma L + \frac{\check{V}_{1\mathsf{A}}}{L}. \tag{88}$$

On the other hand, applying Young's inequality on (87) we get

$$\left\| \sum_{0 \leq j < k \leq n-1} D\boldsymbol{F}_j(\boldsymbol{z}_0)\boldsymbol{F}_k\boldsymbol{z}_0 \right\|^2 \leq n^4 L^2 \|\boldsymbol{F}\boldsymbol{z}_0\|^2 + 2L^2 \left(\sum_{j=0}^{n-1}\delta_{j+1}\right)^2$$

$$\leq n^4 L^2 \|\boldsymbol{F}\boldsymbol{z}_0\|^2 + 2nL^2 \sum_{j=1}^{n}\delta_j^2.$$

Taking the expectation conditioned on $\boldsymbol{z}_0$ and applying Lemma D.5, we conclude that

$$\mathbb{E}\left[\|\boldsymbol{z}_{2n} - (\boldsymbol{z}_0 - \eta n\boldsymbol{F}(\boldsymbol{z}_0 - \eta n\boldsymbol{F}\boldsymbol{z}_0))\|^2 \mid \boldsymbol{z}_0\right] \leq \eta^4 n^4 C_{2\mathsf{R}}\|\boldsymbol{F}\boldsymbol{z}_0\|^2 + \eta^4 n^4 D_{2\mathsf{R}}\|\boldsymbol{F}\boldsymbol{z}_0\|^4 + \eta^4 n^3 V_{2\mathsf{R}}$$

holds for constants

$$C_{2\mathsf{R}} = 2L^2 + 4\rho^2 L^2 + \frac{2\check{C}_{2\mathsf{A}}}{L^2}, \quad D_{2\mathsf{R}} = \frac{2\check{D}_{2\mathsf{A}}}{L^2}, \quad V_{2\mathsf{R}} = 4\sigma^2 L^2 + \frac{2\check{V}_{2\mathsf{A}}}{L^2}. \tag{89}$$

$\square$

# E   CONVERGENCE BOUNDS IN THE STRONGLY MONOTONE SETTING

In this section, we provide further details for Theorem 6.2. The precise statement of Theorem 6.2 is presented as Theorem E.4. As we are now interested in the iterates $\{\boldsymbol{z}_0^k\}_{k \geq 0}$, we omit the subscript 0 unless necessary, and simply write $\boldsymbol{z}^k$ instead of $\boldsymbol{z}_0^k$.

For any of SEG-RR, SEG-FF, and SEG-FFA, we can decompose the update across the epoch into a deterministic EG update plus a noise. More precisely, letting $\boldsymbol{w}_\dagger^k := \boldsymbol{z}^k - \eta_k n\boldsymbol{F}\boldsymbol{z}^k$, we define $\widehat{\boldsymbol{F}}^k$ by the relation $\eta_k n\widehat{\boldsymbol{F}}^k = \eta_k n\boldsymbol{F}\boldsymbol{w}_\dagger^k + \boldsymbol{r}^k$ so that

$$\boldsymbol{z}^{k+1} = \boldsymbol{z}^k - \eta_k n\widehat{\boldsymbol{F}}^k. \tag{90}$$

**Proposition E.1.** *Let $\boldsymbol{F}$ be $\mu$-strongly monotone. Then, for any $\eta_k > 0$, it holds that*

$$\eta_k^2 n^2 \left(1 - \frac{3}{2}\mu\eta_k n - \left(1 + \frac{1}{2}\mu\eta_k n\right)\eta_k^2 n^2 L^2\right)\|\boldsymbol{F}\boldsymbol{z}^k\|^2$$

$$\leq \left(1 - \frac{1}{2}\mu\eta_k n\right)\|\boldsymbol{z}^k - \boldsymbol{z}^*\|^2 - \|\boldsymbol{z}^{k+1} - \boldsymbol{z}^*\|^2 + \frac{2 + \mu\eta_k n}{\mu\eta_k n}\|\boldsymbol{r}^k\|^2. \tag{91}$$

*Proof.* From (90), using Lemma B.7 we get

$$\left\|\boldsymbol{z}^{k+1}-\boldsymbol{z}^*\right\|^2 = \left\|\boldsymbol{z}^k-\boldsymbol{z}^*\right\|^2 - 2\left\langle \eta_k n\widehat{\boldsymbol{F}}^k, \boldsymbol{z}^k-\boldsymbol{z}^*\right\rangle + \left\|\eta_k n\widehat{\boldsymbol{F}}^k\right\|^2$$

$$= \left\|\boldsymbol{z}^k-\boldsymbol{z}^*\right\|^2 - 2\eta_k n\left\langle \boldsymbol{F}\boldsymbol{w}_\dagger^k, \boldsymbol{w}_\dagger^k-\boldsymbol{z}^*\right\rangle - 2\eta_k^2 n^2\left\langle \boldsymbol{F}\boldsymbol{w}_\dagger^k, \boldsymbol{F}\boldsymbol{z}^k\right\rangle$$

$$- 2\left\langle \boldsymbol{r}^k, \boldsymbol{z}^k-\boldsymbol{z}^*\right\rangle + \left\|\eta_k n\widehat{\boldsymbol{F}}^k\right\|^2$$

$$\leq \left\|\boldsymbol{z}^k-\boldsymbol{z}^*\right\|^2 - \mu\eta_k n\left\|\boldsymbol{z}^k-\boldsymbol{z}^*\right\|^2 - 2\eta_k^2 n^2\left\langle \boldsymbol{F}\boldsymbol{w}_\dagger^k, \boldsymbol{F}\boldsymbol{z}^k\right\rangle$$

$$- 2\left\langle \boldsymbol{r}^k, \boldsymbol{z}^k-\boldsymbol{z}^*\right\rangle + \left\|\eta_k n\widehat{\boldsymbol{F}}^k\right\|^2 + 2\mu\eta_k^3 n^3\left\|\boldsymbol{F}\boldsymbol{z}^k\right\|^2.$$

Meanwhile, using the polarization identity (Lemma B.1) and the $L$-smoothness of $\boldsymbol{F}$ we get

$$-2\left\langle \boldsymbol{F}\boldsymbol{w}_\dagger^k, \boldsymbol{F}\boldsymbol{z}^k\right\rangle = \left\|\boldsymbol{F}\boldsymbol{w}_\dagger^k - \boldsymbol{F}\boldsymbol{z}^k\right\|^2 - \left\|\boldsymbol{F}\boldsymbol{w}_\dagger^k\right\|^2 - \left\|\boldsymbol{F}\boldsymbol{z}^k\right\|^2$$

$$\leq L^2\left\|\boldsymbol{w}_\dagger^k - \boldsymbol{z}^k\right\|^2 - \left\|\boldsymbol{F}\boldsymbol{w}_\dagger^k\right\|^2 - \left\|\boldsymbol{F}\boldsymbol{z}^k\right\|^2$$

$$\leq -(1-\eta_k^2 n^2 L^2)\left\|\boldsymbol{F}\boldsymbol{z}^k\right\|^2 - \left\|\boldsymbol{F}\boldsymbol{w}_\dagger^k\right\|^2.$$

Combining the two inequalities and using the definition of $\widehat{\boldsymbol{F}}$ we obtain

$$\left\|\boldsymbol{z}^{k+1}-\boldsymbol{z}^*\right\|^2 \leq (1-\mu\eta_k n)\left\|\boldsymbol{z}^k-\boldsymbol{z}^*\right\|^2 - \eta_k^2 n^2(1-\eta_k^2 n^2 L^2)\left\|\boldsymbol{F}\boldsymbol{z}^k\right\|^2 - \eta_k^2 n^2\left\|\boldsymbol{F}\boldsymbol{w}_\dagger^k\right\|^2$$

$$- 2\left\langle \boldsymbol{r}^k, \boldsymbol{z}^k-\boldsymbol{z}^*\right\rangle + \left\|\eta_k n\boldsymbol{F}\boldsymbol{w}_\dagger^k + \boldsymbol{r}^k\right\|^2 + 2\mu\eta_k^3 n^3\left\|\boldsymbol{F}\boldsymbol{z}^k\right\|^2$$

$$\leq (1-\mu\eta_k n)\left\|\boldsymbol{z}^k-\boldsymbol{z}^*\right\|^2 - \eta_k^2 n^2(1-2\mu\eta_k n - \eta_k^2 n^2 L^2)\left\|\boldsymbol{F}\boldsymbol{z}^k\right\|^2$$

$$- 2\left\langle \boldsymbol{r}^k, \boldsymbol{z}^k-\boldsymbol{z}^*\right\rangle + 2\left\langle \boldsymbol{r}^k, \eta_k n\boldsymbol{F}\boldsymbol{w}_\dagger^k\right\rangle + \left\|\boldsymbol{r}^k\right\|^2$$

$$\leq (1-\mu\eta_k n)\left\|\boldsymbol{z}^k-\boldsymbol{z}^*\right\|^2 - \eta_k^2 n^2(1-2\mu\eta_k n - \eta_k^2 n^2 L^2)\left\|\boldsymbol{F}\boldsymbol{z}^k\right\|^2$$

$$- 2\left\langle \boldsymbol{r}^k, \boldsymbol{z}^k-\eta_k n\boldsymbol{F}\boldsymbol{w}_\dagger^k-\boldsymbol{z}^*\right\rangle + \left\|\boldsymbol{r}^k\right\|^2.$$

Let us consider the inner product term in the last line above. By Lemma B.2 and the nonexpansiveness of the EG update (Lemma B.8), for any $\gamma_k > 0$ we have

$$-2\left\langle \boldsymbol{r}^k, \boldsymbol{z}^k-\eta_k n\boldsymbol{F}\boldsymbol{w}_\dagger^k-\boldsymbol{z}^*\right\rangle \leq \frac{1}{\gamma_k}\left\|\boldsymbol{r}^k\right\|^2 + \gamma_k\left\|\boldsymbol{z}^k-\eta_k n\boldsymbol{F}\boldsymbol{w}_\dagger^k-\boldsymbol{z}^*\right\|^2$$

$$\leq \frac{1}{\gamma_k}\left\|\boldsymbol{r}^k\right\|^2 + \gamma_k\left\|\boldsymbol{z}^k-\boldsymbol{z}^*\right\|^2 - \gamma_k\eta_k^2 n^2(1-\eta_k^2 n^2 L^2)\left\|\boldsymbol{F}\boldsymbol{z}^k\right\|^2.$$

Plugging this back we get

$$\eta_k^2 n^2(1+\gamma_k - 2\mu\eta_k n - (1+\gamma_k)\eta_k^2 n^2 L^2)\left\|\boldsymbol{F}\boldsymbol{z}^k\right\|^2$$

$$\leq (1+\gamma_k - \mu\eta_k n)\left\|\boldsymbol{z}^k-\boldsymbol{z}^*\right\|^2 - \left\|\boldsymbol{z}^{k+1}-\boldsymbol{z}^*\right\|^2 + \left(1+\frac{1}{\gamma_k}\right)\left\|\boldsymbol{r}^k\right\|^2. \tag{92}$$

Choosing $\gamma_k = \frac{\mu\eta_k n}{2}$ completes the proof. $\qquad\square$

**Proposition E.2.** *Let $\boldsymbol{F}$ be a $\mu$-strongly monotone and $L$-Lipschitz operator. Then, whenever $\eta_k < \frac{1}{nL\sqrt{2}}$, it holds that*

$$\left\|\boldsymbol{F}\boldsymbol{z}^{k+1}\right\| \leq \left(1-\frac{\mu n\eta_k}{5}\right)\left\|\boldsymbol{F}\boldsymbol{z}^k\right\| + L\left\|\boldsymbol{r}^k\right\|.$$

*Proof.* Let $\boldsymbol{z}_\dagger^{k+1} := \boldsymbol{z}^k - \eta_k n\boldsymbol{F}(\boldsymbol{z}^k - \eta_k n\boldsymbol{F}\boldsymbol{z}^k)$, so that we have $\left\|\boldsymbol{z}^{k+1}-\boldsymbol{z}_\dagger^{k+1}\right\| = \left\|\boldsymbol{r}^k\right\|$. Then, the $L$-smoothness of $\boldsymbol{F}$ and Lemma B.9 implies

$$\left\|\boldsymbol{F}\boldsymbol{z}^{k+1}\right\| \leq \left\|\boldsymbol{F}\boldsymbol{z}^{k+1} - \boldsymbol{F}\boldsymbol{z}_\dagger^{k+1}\right\| + \left\|\boldsymbol{F}\boldsymbol{z}_\dagger^{k+1}\right\|$$

$$\leq L\left\|\boldsymbol{z}^{k+1} - \boldsymbol{z}_\dagger^{k+1}\right\| + \left\|\boldsymbol{F}\boldsymbol{z}_\dagger^{k+1}\right\|$$

$$\leq L\left\|\boldsymbol{r}^k\right\| + \left(1-\frac{2\mu\eta_k n}{5}\right)^{1/2}\left\|\boldsymbol{F}\boldsymbol{z}^k\right\|$$

$$\leq L\left\|\boldsymbol{r}^k\right\| + \left(1-\frac{\mu\eta_k n}{5}\right)\left\|\boldsymbol{F}\boldsymbol{z}^k\right\|$$

where in the last line we apply a simple inequality $1 - 2x \leq (1-x)^2$ which holds for all $x \in \mathbb{R}$. $\quad\square$

**Lemma E.3.** *Let* $\Omega := \left\| \boldsymbol{F} \boldsymbol{z}^0 \right\|$, *and suppose that* (19) *holds. Given a constant* $b \geq 1/a$ *and the number of epochs* $K$, *say we use constant stepsize* $\eta_k = \frac{\omega}{nK^b}$ *where* $\omega$ *is a constant such that* $\omega \leq \frac{1}{L\sqrt{2}}$ *and*

$$\omega^{a-1} \left( C_1 + D_1 \left( \Omega + \omega^a L V_1 \right) \right) \leq \frac{1}{5\kappa}, \tag{93}$$

*where* $\kappa = L/\mu$ *denotes the condition number. Then for any* $k = 0, 1, \ldots, K$, *it holds that*

$$\left\| \boldsymbol{F} \boldsymbol{z}^k \right\| \leq \Omega + \omega^a L V_1. \tag{94}$$

*Proof.* We show a stronger statement that

$$\left\| \boldsymbol{F} \boldsymbol{z}^k \right\| \leq \left\| \boldsymbol{F} \boldsymbol{z}^0 \right\| + \omega^a L V_1 \cdot \frac{k}{K^{ab}} \tag{95}$$

holds for all $k = 0, 1, \ldots, K$, by induction on $k$. For $k = 0$ there is nothing to show. Let us use induction on $k$, and to this end, suppose that (95) holds for some $k \geq 0$. Then, with noting that (95) implies (94), by Proposition E.2 and (19) we have

$$
\begin{aligned}
\left\| \boldsymbol{F} \boldsymbol{z}^{k+1} \right\| &\leq \left( 1 - \frac{\mu \eta_k n}{5} \right) \left\| \boldsymbol{F} \boldsymbol{z}^k \right\| + \eta_k^a n^a L \left( C_1 \left\| \boldsymbol{F} \boldsymbol{z}^k \right\| + D_1 \left\| \boldsymbol{F} \boldsymbol{z}^k \right\|^2 + V_1 \right) \\
&\leq \left( 1 - \frac{\mu \eta_k n}{5} + \eta_k^a n^a L C_1 + \eta_k^a n^a L D_1 \left( \Omega + \omega^a L V_1 \right) \right) \left\| \boldsymbol{F} \boldsymbol{z}^k \right\| + \eta_k^a n^a L V_1.
\end{aligned} \tag{96}
$$

Here, observe that if $\omega$ satisfies (93) then

$$
\begin{aligned}
\eta_k^a n^a L C_1 + \eta_k^a n^a L D_1 \left( \Omega + \omega^a L V_1 \right) &\leq \frac{\eta_k n L \omega^{a-1}}{K^{b(a-1)}} \left( C_1 + D_1 \left( \Omega + \omega^a L V_1 \right) \right) \\
&\leq \frac{\eta_k n L}{5 \kappa K^{b(a-1)}} \\
&\leq \frac{\mu \eta_k n}{5}
\end{aligned} \tag{97}
$$

where the last line follows from $b(a-1) \geq 0$ and $\kappa = L/\mu$. Therefore, combining (96) and (97) and then invoking the induction hypothesis (95) leads to

$$
\begin{aligned}
\left\| \boldsymbol{F} \boldsymbol{z}^{k+1} \right\| &\leq \left\| \boldsymbol{F} \boldsymbol{z}^k \right\| + \eta_k^a n^a L V_1 \\
&= \left\| \boldsymbol{F} \boldsymbol{z}^k \right\| + \frac{\omega^a L V_1}{K^{ab}} \\
&\leq \left\| \boldsymbol{F} \boldsymbol{z}^0 \right\| + \omega^a L V_1 \cdot \frac{k+1}{K^{ab}}.
\end{aligned}
$$

That is, (95) also holds for $k + 1$, so we are done. $\quad\square$

**Theorem E.4** (Theorem 6.2). *Let* $\Omega := \left\| \boldsymbol{F} \boldsymbol{z}^0 \right\|$, *and suppose that both* (19) *and* (20) *hold. Given a constant* $b \in (1/a, 1)$ *and the number of epochs* $K$, *say we use constant stepsize* $\eta_k \equiv \eta = \frac{\omega}{nK^b}$ *where* $\omega$ *is a constant such that* $\omega \leq \frac{1}{L\sqrt{2}}$ *and*

$$\omega^{a-1} \left( C_1 + D_1 \left( \Omega + \omega^a L V_1 \right) \right) \leq \frac{1}{5\kappa}.$$

*Furthermore, for a constant*

$$\Phi := C_2 + D_2 \left( \Omega + \omega^a L V_1 \right)^2,$$

*suppose that* $\omega$ *is chosen sufficiently small so that the constant stepsize* $\eta$ *further satisfies*

$$\frac{3}{2} \mu \eta n + \left( 1 + \frac{1}{2} \mu \eta n \right) \eta^2 n^2 L^2 + \frac{\eta^{2a-3} n^{2a-3} \Phi (2 + \mu \eta n)}{\mu} \leq 1. \tag{98}$$

*Then for* $\varepsilon = 1 - b$, *it holds that*

$$\mathbb{E} \left[ \left\| \boldsymbol{z}^K - \boldsymbol{z}^* \right\|^2 \right] \leq \exp \left( -\frac{\mu \omega K^\varepsilon}{2} \right) \left\| \boldsymbol{z}^0 - \boldsymbol{z}^* \right\|^2 + \mathcal{O} \left( \frac{1}{nK^{2a-2-(2a-1)\varepsilon}} \right). \tag{99}$$

*Proof.* From (20) and Lemma E.3 it follows that

$$
\mathbb{E}\left[\left\|\boldsymbol{r}\right\|^2 \mid \boldsymbol{z}^k\right] \leq \eta_k^{2a} n^{2a} C_2 \left\|\boldsymbol{F}\boldsymbol{z}^k\right\|^2 + \eta_k^{2a} n^{2a} D_2 \left\|\boldsymbol{F}\boldsymbol{z}^k\right\|^4 + \eta_k^{2a} n^{2a-1} V_2
$$

$$
\leq \eta_k^{2a} n^{2a} C_2 \left\|\boldsymbol{F}\boldsymbol{z}^k\right\|^2 + \eta_k^{2a} n^{2a} D_2 \left(\Omega + \omega^a Z L V_1\right)^2 \left\|\boldsymbol{F}\boldsymbol{z}^k\right\|^2 + \eta_k^{2a} n^{2a-1} V_2
$$

$$
= \eta_k^{2a} n^{2a} \Phi \left\|\boldsymbol{F}\boldsymbol{z}^k\right\|^2 + \eta_k^{2a} n^{2a-1} V_2.
$$

Thus, taking the conditional expectation on (91) we get

$$
\eta_k^2 n^2 \left(1 - \frac{3}{2}\mu\eta_k n - \left(1 + \frac{1}{2}\mu\eta_k n\right)\eta_k^2 n^2 L^2\right)\left\|\boldsymbol{F}\boldsymbol{z}^k\right\|^2
$$

$$
\leq \left(1 - \frac{1}{2}\mu\eta_k n\right)\left\|\boldsymbol{z}^k - \boldsymbol{z}^*\right\|^2 - \mathbb{E}\left[\left\|\boldsymbol{z}^{k+1} - \boldsymbol{z}^*\right\|^2 \mid \boldsymbol{z}^k\right]
$$

$$
+ \frac{2 + \mu\eta_k n}{\mu}\left(\eta_k^{2a-1} n^{2a-1}\Phi\left\|\boldsymbol{F}\boldsymbol{z}^k\right\|^2 + \eta_k^{2a-1} n^{2a-2} V_2\right).
$$

Rearranging the terms, we further get

$$
\eta_k^2 n^2 \left(1 - \frac{3}{2}\mu\eta_k n - \left(1 + \frac{1}{2}\mu\eta_k n\right)\eta_k^2 n^2 L^2 - \frac{\eta_k^{2a-3} n^{2a-3}\Phi(2 + \mu\eta_k n)}{\mu}\right)\left\|\boldsymbol{F}\boldsymbol{z}^k\right\|^2
$$

$$
\leq \left(1 - \frac{1}{2}\mu\eta_k n\right)\left\|\boldsymbol{z}^k - \boldsymbol{z}^*\right\|^2 - \mathbb{E}\left[\left\|\boldsymbol{z}^{k+1} - \boldsymbol{z}^*\right\|^2 \mid \boldsymbol{z}^k\right] + \frac{\eta_k^{2a-1} n^{2a-2} V_2(2 + \mu\eta_k n)}{\mu}.
$$

$$(100)$$

Because $\eta_k = \eta$, by assuming (98), we can guarantee that the left hand side of (100) is nonnegative, so we obtain

$$
\mathbb{E}\left[\left\|\boldsymbol{z}^{k+1} - \boldsymbol{z}^*\right\|^2 \mid \boldsymbol{z}^k\right] \leq \left(1 - \frac{1}{2}\mu\eta_k n\right)\left\|\boldsymbol{z}^k - \boldsymbol{z}^*\right\|^2 + \frac{\eta_k^{2a-1} n^{2a-2} V_2(2 + \mu\eta_k n)}{\mu}.
$$

Applying the law of total expectation, and using the simple identity $1 - x \leq e^{-x}$ which holds for all $x \in \mathbb{R}$, from the above we get

$$
\mathbb{E}\left\|\boldsymbol{z}^{k+1} - \boldsymbol{z}^*\right\|^2 \leq e^{-\frac{1}{2}\mu\eta_k n}\mathbb{E}\left\|\boldsymbol{z}^k - \boldsymbol{z}^*\right\|^2 + \frac{\eta_k^{2a-1} n^{2a-2} V_2(2 + \mu\eta_k n)}{\mu}. \tag{101}
$$

As $\eta_k n L \leq 1$ and $\kappa = {}^L/\mu$, we have $L(2 + \mu\eta_k n) \leq 2\mu\kappa + \mu$, so (101) in turn implies

$$
\mathbb{E}\left\|\boldsymbol{z}^{k+1} - \boldsymbol{z}^*\right\|^2 \leq e^{-\frac{1}{2}\mu\eta_k n}\mathbb{E}\left\|\boldsymbol{z}^k - \boldsymbol{z}^*\right\|^2 + \frac{\eta_k^{2a-1} n^{2a-2} V_2(2\kappa + 1)}{L}.
$$

Noting that $e^{-\frac{1}{2}\mu\eta_k n} \leq 1$, simply unravelling this recurrence leads to

$$
\mathbb{E}\left[\left\|\boldsymbol{z}^K - \boldsymbol{z}^*\right\|^2\right] \leq e^{-\frac{n\mu}{2}\sum_{k=0}^{K-1}\eta_k}\left\|\boldsymbol{z}^0 - \boldsymbol{z}^*\right\|^2 + \sum_{k=0}^{K-1}\frac{\eta_k^{2a-1} n^{2a-2} V_2(2\kappa + 1)}{L}
$$

$$
\leq e^{-\frac{\mu\omega K^{1-b}}{2}}\left\|\boldsymbol{z}^0 - \boldsymbol{z}^*\right\|^2 + \frac{\omega^{2a-1} V_2(2\kappa + 1)}{LnK^{b(2a-1)-1}}. \tag{102}
$$

Now, for the first term to get smaller as the choice of the number of epochs $K$ gets larger, we want $1 - b > 0$. Choosing $b = 1 - \varepsilon$, we get

$$
\mathbb{E}\left[\left\|\boldsymbol{z}^K - \boldsymbol{z}^*\right\|^2\right] \leq \exp\left(-\frac{\mu\omega K^\varepsilon}{2}\right)\left\|\boldsymbol{z}^0 - \boldsymbol{z}^*\right\|^2 + \mathcal{O}\left(\frac{1}{nK^{2a-2-(2a-1)\varepsilon}}\right)
$$

as claimed. $\qquad\square$

## F  CONVERGENCE RATE OF SEG-FFA IN THE MONOTONE SETTING

In this section, we consider SEG-FFA. As in the previous section, we decompose the update across the epoch into a deterministic EG update plus a noise, as

$$
\begin{aligned}
\boldsymbol{w}_\dagger^k &:= \boldsymbol{z}^k - \eta_k n \boldsymbol{F}\boldsymbol{z}^k, \\
\boldsymbol{z}^{k+1} &= \boldsymbol{z}^k - \eta_k n \widehat{\boldsymbol{F}}^k.
\end{aligned} \tag{103}
$$

for $\widehat{\boldsymbol{F}}^k$ defined by the equation

$$\eta_k n \widehat{\boldsymbol{F}}^k = \eta_k n \boldsymbol{F} \boldsymbol{w}_\dagger^k + \boldsymbol{r}^k. \tag{104}$$

**Lemma F.1.** *Let $\boldsymbol{u}$ be any point in $\mathbb{R}^{d_1+d_2}$. Then, for any $\eta_k > 0$ and $\gamma_k > 0$, it holds that*

$$
\begin{aligned}
2\eta_k n \left\langle \boldsymbol{F} \boldsymbol{u}, \boldsymbol{w}_\dagger^k - \boldsymbol{u} \right\rangle &\leq \left\| \boldsymbol{z}^k - \boldsymbol{u} \right\|^2 - \frac{1}{1+\gamma_k} \left\| \boldsymbol{z}^{k+1} - \boldsymbol{u} \right\|^2 \\
&\quad - \eta_k^2 n^2 (1 - \eta_k^2 n^2 L^2) \left\| \boldsymbol{F} \boldsymbol{z}^k \right\|^2 + \frac{1}{\gamma_k} \left\| \boldsymbol{r}^k \right\|^2.
\end{aligned}
\tag{105}
$$

*Proof.* By (103) and (104) we get

$$
\begin{aligned}
\left\| \boldsymbol{z}^{k+1} - \boldsymbol{u} \right\|^2 &= \left\| \boldsymbol{z}^k - \eta_k n \widehat{\boldsymbol{F}}^k - \boldsymbol{u} \right\|^2 \\
&= \left\| \boldsymbol{z}^k - \boldsymbol{u} \right\|^2 - 2 \left\langle \eta_k n \widehat{\boldsymbol{F}}^k, \boldsymbol{z}^k - \boldsymbol{u} \right\rangle + \left\| \eta_k n \widehat{\boldsymbol{F}}^k \right\|^2 \\
&= \left\| \boldsymbol{z}^k - \boldsymbol{u} \right\|^2 - 2 \left\langle \eta_k n \boldsymbol{F} \boldsymbol{w}_\dagger^k, \boldsymbol{w}_\dagger^k - \boldsymbol{u} \right\rangle - 2 \left\langle \eta_k n \boldsymbol{F} \boldsymbol{w}_\dagger^k, \boldsymbol{z}^k - \boldsymbol{w}_\dagger^k \right\rangle - 2 \left\langle \boldsymbol{r}^k, \boldsymbol{z}^k - \boldsymbol{u} \right\rangle \\
&\quad + \left\| \eta_k n \boldsymbol{F} \boldsymbol{w}_\dagger^k \right\|^2 + 2 \left\langle \boldsymbol{r}^k, \eta_k n \boldsymbol{F} \boldsymbol{w}_\dagger^k \right\rangle + \left\| \boldsymbol{r}^k \right\|^2 \\
&= \left\| \boldsymbol{z}^k - \boldsymbol{u} \right\|^2 - 2\eta_k n \left\langle \boldsymbol{F} \boldsymbol{w}_\dagger^k, \boldsymbol{w}_\dagger^k - \boldsymbol{u} \right\rangle - 2 \left\langle \eta_k n \boldsymbol{F} \boldsymbol{w}_\dagger^k, \eta_k n \boldsymbol{F} \boldsymbol{z}^k \right\rangle \\
&\quad + \left\| \eta_k n \boldsymbol{F} \boldsymbol{w}_\dagger^k \right\|^2 - 2 \left\langle \boldsymbol{r}^k, \boldsymbol{z}^k - \eta_k n \boldsymbol{F} \boldsymbol{w}_\dagger^k - \boldsymbol{u} \right\rangle + \left\| \boldsymbol{r}^k \right\|^2 \\
&= \left\| \boldsymbol{z}^k - \boldsymbol{u} \right\|^2 - 2\eta_k n \left\langle \boldsymbol{F} \boldsymbol{w}_\dagger^k, \boldsymbol{w}_\dagger^k - \boldsymbol{u} \right\rangle - 2\eta_k^2 n^2 \left\langle \boldsymbol{F} \boldsymbol{w}_\dagger^k, \boldsymbol{F} \boldsymbol{z}^k \right\rangle \\
&\quad + \eta_k^2 n^2 \left\| \boldsymbol{F} \boldsymbol{w}_\dagger^k \right\|^2 - 2 \left\langle \boldsymbol{r}^k, \boldsymbol{z}^{k+1} - \boldsymbol{u} \right\rangle - \left\| \boldsymbol{r}^k \right\|^2.
\end{aligned}
$$

We now bound the inner products. On one hand, by the polarization identity (Lemma B.1) and the $L$-smoothness of $f$, we have

$$
\begin{aligned}
-2 \left\langle \boldsymbol{F} \boldsymbol{w}_\dagger^k, \boldsymbol{F} \boldsymbol{z}^k \right\rangle &= \left\| \boldsymbol{F} \boldsymbol{w}_\dagger^k - \boldsymbol{F} \boldsymbol{z}^k \right\|^2 - \left\| \boldsymbol{F} \boldsymbol{w}_\dagger^k \right\|^2 - \left\| \boldsymbol{F} \boldsymbol{z}^k \right\|^2 \\
&\leq L^2 \left\| -\eta_k n \boldsymbol{F} \boldsymbol{z}^k \right\|^2 - \left\| \boldsymbol{F} \boldsymbol{w}_\dagger^k \right\|^2 - \left\| \boldsymbol{F} \boldsymbol{z}^k \right\|^2 \\
&= -(1 - \eta_k^2 n^2 L^2) \left\| \boldsymbol{F} \boldsymbol{z}^k \right\|^2 - \left\| \boldsymbol{F} \boldsymbol{w}_\dagger^k \right\|^2.
\end{aligned}
$$

On the other hand, by the weighted AM-GM inequality (Lemma B.2), for any number $a_k \in (0,1)$ it holds that

$$
-2 \left\langle \boldsymbol{r}^k, \boldsymbol{z}^{k+1} - \boldsymbol{u} \right\rangle \leq \frac{1}{a_k} \left\| \boldsymbol{r}^k \right\|^2 + a_k \left\| \boldsymbol{z}^{k+1} - \boldsymbol{u} \right\|^2.
$$

Using these two bounds, we get

$$
\begin{aligned}
\left\| \boldsymbol{z}^{k+1} - \boldsymbol{u} \right\|^2 &\leq \left\| \boldsymbol{z}^k - \boldsymbol{u} \right\|^2 - 2\eta_k n \left\langle \boldsymbol{F} \boldsymbol{w}_\dagger^k, \boldsymbol{w}_\dagger^k - \boldsymbol{u} \right\rangle - \eta_k^2 n^2 (1 - \eta_k^2 n^2 L^2) \left\| \boldsymbol{F} \boldsymbol{z}^k \right\|^2 \\
&\quad - \eta_k^2 n^2 \left\| \boldsymbol{F} \boldsymbol{w}_\dagger^k \right\|^2 + \eta_k^2 n^2 \left\| \boldsymbol{F} \boldsymbol{w}_\dagger^k \right\|^2 + a_k \left\| \boldsymbol{z}^{k+1} - \boldsymbol{u} \right\|^2 + \left( \frac{1}{a_k} - 1 \right) \left\| \boldsymbol{r}^k \right\|^2.
\end{aligned}
$$

Choosing $a_k = \frac{\gamma_k}{1+\gamma_k}$ and rearranging the terms, we obtain

$$
\begin{aligned}
2\eta_k n \left\langle \boldsymbol{F} \boldsymbol{w}_\dagger^k, \boldsymbol{w}_\dagger^k - \boldsymbol{u} \right\rangle &\leq \left\| \boldsymbol{z}^k - \boldsymbol{u} \right\|^2 - \frac{1}{1+\gamma_k} \left\| \boldsymbol{z}^{k+1} - \boldsymbol{u} \right\|^2 \\
&\quad - \eta_k^2 n^2 (1 - \eta_k^2 n^2 L^2) \left\| \boldsymbol{F} \boldsymbol{z}^k \right\|^2 + \frac{1}{\gamma_k} \left\| \boldsymbol{r}^k \right\|^2.
\end{aligned}
\tag{106}
$$

As the monotonicity of $\boldsymbol{F}$ implies

$$
\left\langle \boldsymbol{F} \boldsymbol{w}_\dagger^k, \boldsymbol{w}_\dagger^k - \boldsymbol{u} \right\rangle \geq \left\langle \boldsymbol{F} \boldsymbol{u}, \boldsymbol{w}_\dagger^k - \boldsymbol{u} \right\rangle,
$$

plugging this to (106) gives us the claimed inequality. $\qquad \square$

Now we show that choosing the appropriate stepsizes leads to $\left\| \boldsymbol{F} \boldsymbol{z}^k \right\|$ being bounded uniformly over $k$.

**Proposition F.2.** *Say we are using* *SEG-FFA*. *Let the sequence of stepsizes* $\{\eta_k\}_{k\geq 0}$ *be nonincreasing, with*

$$S := \sum_{k=0}^{\infty} \eta_k^3 n^3 L^3 < \infty. \tag{107}$$

*Suppose that initial stepsize* $\eta_0$ *is chosen sufficiently small so that*

$$\eta_0^2 n^2 L^2 + \frac{3\eta_0 n C_{1A}^2}{L^3} + \frac{3\eta_0 n D_{1A}^2}{L} \cdot e^S \left( \left\| z^0 - z^* \right\|^2 + \frac{6SV_{1A}^2}{L^6} \right) \leq 1 \tag{108}$$

*for constants* $C_{1A}$, $D_{1A}$, *and* $V_{1A}$ *defined in* (61)–(63). *Then for all* $k \geq 0$,

$$\left\| Fz^k \right\|^2 \leq e^S L^2 \left( \left\| z^0 - z^* \right\|^2 + \frac{6SV_{1A}^2}{L^6} \right). \tag{109}$$

*Proof.* We use induction on $k$, to establish a stronger inequality

$$\left\| z^k - z^* \right\|^2 \leq e^S \left( \left\| z^0 - z^* \right\|^2 + \frac{6SV_{1A}^2}{L^6} \right). \tag{110}$$

To see that (110) indeed implies (109), notice that by the $L$-smoothness of $f$ it holds that

$$\left\| Fz^k \right\|^2 = \left\| Fz^k - Fz^* \right\|^2 \leq L^2 \left\| z^k - z^* \right\|^2.$$

For the case when $k = 0$, as $S > 0$, it is clear that (110) holds. Now suppose that (110) holds for some $k \geq 0$. Applying Young's inequality on (19) leads to

$$\left\| r^k \right\|^2 \leq 3\eta_k^6 n^6 \left( C_{1A}^2 \left\| Fz^k \right\|^2 + D_{1A}^2 \left\| Fz^k \right\|^4 + V_{1A}^2 \right).$$

Taking $u = z^*$ in (105) and then using the bound on $\left\| r^k \right\|^2$ above, we obtain

$$\eta_k^2 n^2 \left( 1 - \eta_k^2 n^2 L^2 - \frac{3\eta_k^4 n^4 C_{1A}^2}{\gamma_k} - \frac{3\eta_k^4 n^4 D_{1A}^2}{\gamma_k} \left\| Fz^k \right\|^2 \right) \left\| Fz^k \right\|^2$$
$$\leq \left\| z^k - z^* \right\|^2 - \frac{1}{1 + \gamma_k} \left\| z^{k+1} - z^* \right\|^2 + \frac{3\eta_k^6 n^6 V_{1A}^2}{\gamma_k}. \tag{111}$$

Choose $\gamma_k = \eta_k^3 n^3 L^3$. Notice that (108) implies $\eta_0 n L \leq 1$, henceforth $\eta_k \leq \eta_0 \leq 1/nL$. This, with the induction hypothesis (109), implies

$$\eta_k^2 n^2 L^2 + \frac{3\eta_k^4 n^4 C_{1A}^2}{\gamma_k} + \frac{3\eta_k^4 n^4 D_{1A}^2}{\gamma_k} \left\| Fz^k \right\|^2$$
$$= \eta_k^2 n^2 L^2 + \frac{3\eta_k n C_{1A}^2}{L^3} + \frac{3\eta_k n D_{1A}^2}{L^3} \left\| Fz^k \right\|^2$$
$$\leq \eta_0^2 n^2 L^2 + \frac{3\eta_0 n C_{1A}^2}{L^3} + \frac{3\eta_0 n D_{1A}^2}{L^3} \left\| Fz^k \right\|^2$$
$$\leq \eta_0^2 n^2 L^2 + \frac{3\eta_0 n C_{1A}^2}{L^3} + \frac{3\eta_0 n D_{1A}^2}{L^3} \cdot e^S L^2 \left( \left\| z^0 - z^* \right\|^2 + \frac{6SV_{1A}^2}{L^6} \right)$$
$$\leq 1.$$

That is, the left hand side of (111) becomes nonnegative. Then it is immediate from (111) that

$$\left\| z^{k+1} - z^* \right\|^2 \leq (1 + \gamma_k) \left\| z^k - z^* \right\|^2 + \frac{3\eta_k^6 n^6 (1 + \gamma_k) V_{1A}^2}{\gamma_k}$$
$$\leq (1 + \eta_k^3 n^3 L^3) \left\| z^k - z^* \right\|^2 + \frac{6\eta_k^3 n^3 V_{1A}^2}{L^3}.$$

Using Lemma B.10 to unravel this recurrence relation, we obtain

$$\left\|\boldsymbol{z}^{k+1} - \boldsymbol{z}^*\right\|^2 \leq \left(\prod_{j=0}^{k}\left(1 + \eta_j^3 n^3 L^3\right)\right)\left(\left\|\boldsymbol{z}^0 - \boldsymbol{z}^*\right\|^2 + \sum_{j=0}^{k}\frac{6\eta_j^3 n^3 V_{1A}^2}{L^3}\right)$$

$$\leq e^{\sum_{j=0}^{k}\eta_j^3 n^3 L^3}\left(\left\|\boldsymbol{z}^0 - \boldsymbol{z}^*\right\|^2 + \frac{6V_{1A}^2}{L^6}\sum_{j=0}^{k}\eta_j^3 n^3 L^3\right)$$

$$\leq e^S\left(\left\|\boldsymbol{z}^0 - \boldsymbol{z}^*\right\|^2 + \frac{6S V_{1A}^2}{L^6}\right)$$

which shows that (110) also holds when $k$ is replaced by $k+1$. This completes the proof. $\qquad\square$

**Theorem F.3** (Formal version of Theorem 5.4)**.** *Suppose that we are using* **SEG-FFA** *with* $\eta_k = \frac{\eta_0 \sqrt[3]{2}\log 2}{(k+2)^{1/3}\log(k+2)}$ *for* $k = 0, 1, \ldots$, *where, for* $S := \sum_{k=0}^{\infty}\eta_k^3 n^3 L^3$, *the initial stepsize* $\eta_0$ *is chosen so that*

$$\eta_0^2 n^2 L^2 + \frac{3\eta_0 n C_{1A}^2}{L^3} + \frac{3\eta_0 n D_{1A}^2}{L}\cdot e^S\left(\left\|\boldsymbol{z}^0 - \boldsymbol{z}^*\right\|^2 + \frac{6S V_{1A}^2}{L^6}\right) \leq 1 \qquad (112)$$

*for constants* $C_{1A}$, $D_{1A}$, *and* $V_{1A}$ *defined in* (61)–(63), *and there exists a positive constant* $\lambda > 0$ *such that*

$$\eta_0^2 n^2 L^2 + \frac{\eta_0 n C_{2A}}{L^3} + \frac{\eta_0 n D_{2A}}{L}\cdot e^S\left(\left\|\boldsymbol{z}^0 - \boldsymbol{z}^*\right\|^2 + \frac{6S V_{1A}^2}{L^6}\right) \leq 1 - \lambda \qquad (113)$$

*for constants* $C_{2A}$, $D_{2A}$, *and* $V_{2A}$ *defined in* (74)–(76). *Then for* $\Lambda := \lambda e^{-3/2}(\sqrt[3]{2}\log 2)^2$ *it holds that*

$$\min_{k=0,1,\ldots,K}\mathbb{E}\left\|\boldsymbol{F}\boldsymbol{z}^k\right\|^2 \leq \frac{(\log(K+3))^2}{(K+3)^{1/3}}\cdot\left(\frac{\left\|\boldsymbol{z}^0 - \boldsymbol{z}^*\right\|^2 + \frac{3V_{2A}}{nL^6}}{\Lambda\eta_0^2 n^2}\right). \qquad (114)$$

*Proof.* As the sequence of stepsizes $\{\eta_k\}_{k\geq 0}$ is nonincreasing and (112) asserts that $\eta_0 \leq 1/nL$, we can use the bounds established in Theorem D.8 and Theorem D.12. Also, the premises required for Proposition F.2 are also satisfied, so the bound (109) holds.

Taking $\boldsymbol{u} = \boldsymbol{z}^*$ and $\gamma_k = \eta_k^3 n^3 L^3$ in (105), with using (73) and (113), we obtain

$$0 = 2\eta_k n\left\langle \boldsymbol{F}\boldsymbol{z}^*, \boldsymbol{w}_\dagger^k - \boldsymbol{z}^*\right\rangle$$

$$\leq \left\|\boldsymbol{z}^k - \boldsymbol{z}^*\right\|^2 - \frac{1}{1+\gamma_k}\mathbb{E}\left[\left\|\boldsymbol{z}^{k+1} - \boldsymbol{z}^*\right\|^2\Big|\boldsymbol{z}^k\right] - \eta_k^2 n^2(1 - \eta_k^2 n^2 L^2)\left\|\boldsymbol{F}\boldsymbol{z}^k\right\|^2 + \frac{1}{\gamma_k}\mathbb{E}\left[\left\|\boldsymbol{r}^k\right\|^2\Big|\boldsymbol{z}^k\right]$$

$$\leq \left\|\boldsymbol{z}^k - \boldsymbol{z}^*\right\|^2 - \frac{1}{1+\gamma_k}\mathbb{E}\left[\left\|\boldsymbol{z}^{k+1} - \boldsymbol{z}^*\right\|^2\Big|\boldsymbol{z}^k\right]$$

$$- \eta_k^2 n^2(1 - \eta_k^2 n^2 L^2)\left\|\boldsymbol{F}\boldsymbol{z}^k\right\|^2 + \frac{1}{L^3}\left(\eta_k^3 n^3 C_{2A}\left\|\boldsymbol{F}\boldsymbol{z}^k\right\|^2 + \eta_k^3 n^3 D_{2A}\left\|\boldsymbol{F}\boldsymbol{z}^k\right\|^4 + \eta_k^3 n^2 V_{2A}\right)$$

$$\leq \left\|\boldsymbol{z}^k - \boldsymbol{z}^*\right\|^2 - \frac{1}{1+\gamma_k}\mathbb{E}\left[\left\|\boldsymbol{z}^{k+1} - \boldsymbol{z}^*\right\|^2\Big|\boldsymbol{z}^k\right]$$

$$- \eta_k^2 n^2\left(1 - \eta_k^2 n^2 L^2 - \frac{\eta_k n C_{2A}}{L^3} - \frac{\eta_k n D_{2A}}{L^3}\left\|\boldsymbol{F}\boldsymbol{z}^k\right\|^2\right)\left\|\boldsymbol{F}\boldsymbol{z}^k\right\|^2 + \frac{\eta_k^3 n^2 V_{2A}}{L^3}$$

$$\leq \left\|\boldsymbol{z}^k - \boldsymbol{z}^*\right\|^2 - \frac{1}{1+\gamma_k}\mathbb{E}\left[\left\|\boldsymbol{z}^{k+1} - \boldsymbol{z}^*\right\|^2\Big|\boldsymbol{z}^k\right] - \lambda\eta_k^2 n^2\left\|\boldsymbol{F}\boldsymbol{z}^k\right\|^2 + \frac{\eta_k^3 n^2 V_{2A}}{L^3}.$$

By the law of total expectation, and that $\gamma_k = \eta_k^3 n^3 L^3 < 1$, from the above we get

$$(1+\gamma_k)\lambda\eta_k^2 n^2\,\mathbb{E}\left\|\boldsymbol{F}\boldsymbol{z}^k\right\|^2 \leq (1+\gamma_k)\,\mathbb{E}\left\|\boldsymbol{z}^k - \boldsymbol{z}^*\right\|^2 - \mathbb{E}\left\|\boldsymbol{z}^{k+1} - \boldsymbol{z}^*\right\|^2 + \frac{(1+\gamma_k)\eta_k^3 n^2 V_{2A}}{L^3}$$

$$\leq (1+\gamma_k)\,\mathbb{E}\left\|\boldsymbol{z}^k - \boldsymbol{z}^*\right\|^2 - \mathbb{E}\left\|\boldsymbol{z}^{k+1} - \boldsymbol{z}^*\right\|^2 + \frac{2\eta_k^3 n^2 V_{2A}}{L^3}.$$

This recurrence can be unraveled using Lemma B.10, giving us

$$\mathbb{E}\left\|\boldsymbol{z}^{K+1}-\boldsymbol{z}^*\right\|^2+\sum_{k=0}^{K}(1+\gamma_k)\lambda\eta_j^2 n^2\,\mathbb{E}\left\|\boldsymbol{F}\boldsymbol{z}^k\right\|^2$$
$$\leq\left(\prod_{k=0}^{K}(1+\gamma_k)\right)\left(\left\|\boldsymbol{z}^0-\boldsymbol{z}^*\right\|^2+\sum_{k=0}^{K}\frac{2\eta_k^3 n^2 V_{\mathsf{2A}}}{L^3}\right). \tag{115}$$

For the left hand side of (115), we have

$$\mathbb{E}\left\|\boldsymbol{z}^{K+1}-\boldsymbol{z}^*\right\|^2+\sum_{k=0}^{K}(1+\gamma_k)\lambda\eta_k^2 n^2\,\mathbb{E}\left\|\boldsymbol{F}\boldsymbol{z}^k\right\|^2\geq\lambda\sum_{k=0}^{K}\eta_k^2 n^2\,\mathbb{E}\left\|\boldsymbol{F}\boldsymbol{z}^k\right\|^2$$
$$\geq\lambda\min_{k=0,1,\ldots,K}\mathbb{E}\left\|\boldsymbol{F}\boldsymbol{z}^k\right\|^2\sum_{k=0}^{K}\eta_k^2 n^2.$$

From Lemma B.11, we know that whenever $K\geq 1$,

$$\sum_{k=0}^{K}\eta_k^2 n^2=\eta_0^2 n^2(\sqrt[3]{2}\log 2)^2\sum_{k=0}^{K}\frac{1}{(k+2)^{2/3}(\log(k+2))^2}$$
$$\geq\eta_0^2 n^2(\sqrt[3]{2}\log 2)^2\cdot\frac{(K+3)^{1/3}}{(\log(K+3))^2}.$$

Meanwhile, as $x\mapsto\frac{2(\log 2)^3}{(x+2)(\log(x+2))^3}$ is a decreasing function, we have

$$\sum_{k=0}^{\infty}\frac{2(\log 2)^3}{(k+2)(\log(k+2))^3}\leq 1+\frac{2(\log 2)^3}{3(\log 3)^3}+\int_1^{\infty}\frac{2(\log 2)^3}{(x+2)(\log(x+2))^3}\,\mathrm{d}x$$
$$\leq 1+\frac{2(\log 2)^3}{3(\log 3)^3}+\frac{(\log 2)^3}{(\log 3)^2}\quad\leq\frac{3}{2}$$

and thus

$$S=\sum_{k=0}^{\infty}\eta_k^3 n^3 L^3=\eta_0^3 n^3 L^3\sum_{k=0}^{\infty}\frac{2(\log 2)^3}{(k+2)(\log(k+2))^3}\leq\frac{3}{2}\eta_0^3 n^3 L^3\leq\frac{3}{2}.$$

Thus, for the right hand side of (115), it holds that

$$\left(\prod_{k=0}^{K}(1+\gamma_k)\right)\left(\left\|\boldsymbol{z}^0-\boldsymbol{z}^*\right\|^2+\sum_{k=0}^{K}\frac{2\eta_k^3 n^2 V_{\mathsf{2A}}}{L^3}\right)\leq e^{\sum_{k=0}^{K}\gamma_k}\left(\left\|\boldsymbol{z}^0-\boldsymbol{z}^*\right\|^2+\sum_{k=0}^{K}\frac{2\eta_k^3 n^2 V_{\mathsf{2A}}}{L^3}\right)$$
$$\leq e^S\left(\left\|\boldsymbol{z}^0-\boldsymbol{z}^*\right\|^2+\frac{2SV_{\mathsf{2A}}}{nL^6}\right)$$
$$\leq e^{3/2}\left(\left\|\boldsymbol{z}^0-\boldsymbol{z}^*\right\|^2+\frac{3V_{\mathsf{2A}}}{nL^6}\right).$$

Therefore, from (115) we get

$$\lambda\eta_0^2 n^2(\sqrt[3]{2}\log 2)^2\cdot\frac{(K+3)^{1/3}}{(\log(K+3))^2}\cdot\min_{k=0,1,\ldots,K}\mathbb{E}\left\|\boldsymbol{F}\boldsymbol{z}^k\right\|^2\leq e^{3/2}\left(\left\|\boldsymbol{z}^0-\boldsymbol{z}^*\right\|^2+\frac{3V_{\mathsf{2A}}}{nL^6}\right).$$

Letting $\Lambda:=\lambda e^{-3/2}(\sqrt[3]{2}\log 2)^2$ and simply rearranging the terms gives us the desired inequality. $\square$

# G  PROOF OF LOWER BOUNDS

## G.1  PROOF OF SEG-US, SEG-RR AND SEG-FF DIVERGENCE

We prove the divergence of SEG-US, SEG-RR and SEG-FF in each proposition below, using the same worst-case problem for $n=2$. These constitute the proof of Theorem 4.2.

**Proposition G.1** (Part of Theorem 4.2). *For $n = 2$, there exists a convex-concave minimax problem $f(x, y) = \frac{1}{2} \sum_{i=1}^{2} f_i(x, y)$ having a monotone $\boldsymbol{F}$, consisting of $L$-smooth quadratic $f_i$'s satisfying Assumption 4 with $(\rho, \sigma) = (1, 0)$ such that SEG-US diverges in expectation for any choice of stepsizes $\{\alpha_t\}_{t \geq 0}$ and $\{\beta_t\}_{t \geq 0}$. That is, for all $t \geq 0$,*

$$\mathbb{E}\left[\|\boldsymbol{z}_{t+1}\|^2\right] > \mathbb{E}\left[\|\boldsymbol{z}_t\|^2\right], \quad \mathbb{E}\left[\|\boldsymbol{F}\boldsymbol{z}_{t+1}\|^2\right] > \mathbb{E}\left[\|\boldsymbol{F}\boldsymbol{z}_t\|^2\right].$$

*Proof.* We consider the case of

$$f_1(x, y) = -\frac{L}{4}x^2 + \frac{L}{2}xy - \frac{L}{4}y^2,$$

$$f_2(x, y) = \frac{L}{4}x^2 + \frac{L}{2}xy + \frac{L}{4}y^2,$$

which result in a bilinear (and hence convex-concave) objective function

$$f(x, y) = \frac{1}{2}\sum_{i=1}^{2} f_i(x, y) = \frac{L}{2}xy. \tag{116}$$

One can quickly check from the definitions of the component functions $f_1$ and $f_2$ that the corresponding saddle gradient operators are given as

$$\boldsymbol{F}_1\boldsymbol{z} = \underbrace{\begin{bmatrix} -L/2 & L/2 \\ -L/2 & L/2 \end{bmatrix}}_{:=\boldsymbol{A}_1}\boldsymbol{z}, \quad \boldsymbol{F}_2\boldsymbol{z} = \underbrace{\begin{bmatrix} L/2 & L/2 \\ -L/2 & -L/2 \end{bmatrix}}_{:=\boldsymbol{A}_2}\boldsymbol{z}, \quad \boldsymbol{F}\boldsymbol{z} = \begin{bmatrix} 0 & L/2 \\ -L/2 & 0 \end{bmatrix}\boldsymbol{z}$$

where $\boldsymbol{z} = (x, y) \in \mathbb{R}^2$. From the fact that $\|\boldsymbol{A}_i\| \leq L$ for all $i$'s, we can confirm that $f_i$'s are indeed $L$-smooth. As for Assumption 4, we can verify that

$$\frac{1}{2}\sum_{i=1}^{2}\|\boldsymbol{F}_i\boldsymbol{z} - \boldsymbol{F}\boldsymbol{z}\|^2 = \frac{L^2}{4}\|\boldsymbol{z}\|^2 = (\|\boldsymbol{F}\boldsymbol{z}\|)^2,$$

thus proving that our example $f$ indeed satisfies Assumption 4 with $(\rho, \sigma) = (1, 0)$.

We now proceed to show that for this particular worst-case example $f$, SEG-US diverges in expectation. For $t \geq 0$, the $(t+1)$-th iteration of SEG-US starts at $\boldsymbol{z}_t$, and the algorithm uniformly chooses an index $i(t)$ from $[n]$. The algorithm then makes an update

$$\boldsymbol{w}_t = \boldsymbol{z}_t - \alpha_t \boldsymbol{F}_{i(t)}\boldsymbol{z}_t,$$

$$\boldsymbol{z}_{t+1} = \boldsymbol{z}_t - \beta_t \boldsymbol{F}_{i(t)}\boldsymbol{w}_t.$$

In our worst-case example $f$, the updates can be compactly written as

$$\boldsymbol{z}_{t+1} = (\boldsymbol{I} - \beta_t \boldsymbol{A}_{i(t)} + \alpha_t \beta_t \boldsymbol{A}_{i(t)}^2)\boldsymbol{z}_t.$$

Since we have $n = 2$, the update can be summarized as

$$\boldsymbol{z}_{t+1} = \begin{cases} (\boldsymbol{I} - \beta_t \boldsymbol{A}_1 + \alpha_t \beta_t \boldsymbol{A}_1^2)\boldsymbol{z}_t & \text{with probability } 1/2, \\ (\boldsymbol{I} - \beta_t \boldsymbol{A}_2 + \alpha_t \beta_t \boldsymbol{A}_2^2)\boldsymbol{z}_t & \text{with probability } 1/2. \end{cases}$$

By the definition of $\boldsymbol{A}_1$ and $\boldsymbol{A}_2$ and using $\boldsymbol{A}_1^2 = \boldsymbol{A}_2^2 = \boldsymbol{0}$, we can verify that

$$\boldsymbol{N}_1 := \boldsymbol{I} - \beta_t \boldsymbol{A}_1 + \alpha_t \beta_t \boldsymbol{A}_1^2 = \begin{bmatrix} 1 + \frac{\beta_t L}{2} & -\frac{\beta_t L}{2} \\ \frac{\beta_t L}{2} & 1 - \frac{\beta_t L}{2} \end{bmatrix},$$

$$\boldsymbol{N}_2 := \boldsymbol{I} - \beta_t \boldsymbol{A}_2 + \alpha_t \beta_t \boldsymbol{A}_2^2 = \begin{bmatrix} 1 - \frac{\beta_t L}{2} & -\frac{\beta_t L}{2} \\ \frac{\beta_t L}{2} & 1 + \frac{\beta_t L}{2} \end{bmatrix}.$$

From this, we notice that the expectation of $\|\boldsymbol{z}_{t+1}\|^2$ conditional on $\boldsymbol{z}_t$ reads

$$\mathbb{E}\left[\|\boldsymbol{z}_{t+1}\|^2 \,\Big|\, \boldsymbol{z}_t\right] = \boldsymbol{z}_t^\top \left(\frac{\boldsymbol{N}_1^\top \boldsymbol{N}_1 + \boldsymbol{N}_2^\top \boldsymbol{N}_2}{2}\right) \boldsymbol{z}_t.$$

Working out the calculations, we can check that

$$\frac{\boldsymbol{N}_1^\top \boldsymbol{N}_1 + \boldsymbol{N}_2^\top \boldsymbol{N}_2}{2} = \begin{bmatrix} 1 + \frac{\beta_t^2 L^2}{2} & 0 \\ 0 & 1 + \frac{\beta_t^2 L^2}{2} \end{bmatrix},$$

thus resulting in

$$\mathbb{E}\left[\|\boldsymbol{z}_{t+1}\|^2 \,\Big|\, \boldsymbol{z}_t\right] = \left(1 + \frac{\beta_t^2 L^2}{2}\right)\|\boldsymbol{z}_t\|^2.$$

Since this holds for all $t \geq 0$, SEG-US diverges in expectation, for any positive stepsizes $\{\alpha_t\}_{t \geq 0}$ and $\{\beta_t\}_{t \geq 0}$. The statement on $\|\boldsymbol{F}\boldsymbol{z}_t\|$ follows by realizing that $\|\boldsymbol{F}\boldsymbol{z}\| = \frac{L}{2}\|\boldsymbol{z}\|$. $\qquad\square$

**Proposition G.2** (Part of Theorem 4.2). *For $n = 2$, there exists a convex-concave minimax problem $f(x, y) = \frac{1}{2}\sum_{i=1}^{2} f_i(x, y)$ having a monotone $\boldsymbol{F}$, consisting of $L$-smooth quadratic $f_i$'s satisfying Assumption 4 with $(\rho, \sigma) = (1, 0)$ such that SEG-RR diverges in expectation for any choice of stepsizes $\{\alpha_k\}_{k \geq 0}$ and $\{\beta_k\}_{k \geq 0}$. That is, for any $k \geq 0$,*

$$\mathbb{E}\left[\left\|\boldsymbol{z}_0^{k+1}\right\|^2\right] > \mathbb{E}\left[\left\|\boldsymbol{z}_0^k\right\|^2\right], \quad \mathbb{E}\left[\left\|\boldsymbol{F}\boldsymbol{z}_0^{k+1}\right\|^2\right] > \mathbb{E}\left[\left\|\boldsymbol{F}\boldsymbol{z}_0^k\right\|^2\right].$$

*Proof.* The proof uses the same example as Proposition G.1, outlined in (116). We show that for this particular worst-case example $f$, SEG-RR diverges in expectation. For $k \geq 0$, the $(k+1)$-th epoch of SEG-RR starts at $\boldsymbol{z}_0^k$, and the algorithm randomly chooses a permutation $\tau_k : [n] \to [n]$. The algorithm then goes through a series of updates

$$\boldsymbol{w}_i^k = \boldsymbol{z}_i^k - \alpha_k \boldsymbol{F}_{\tau_k(i+1)}\boldsymbol{z}_i^k,$$
$$\boldsymbol{z}_{i+1}^k = \boldsymbol{z}_i^k - \beta_k \boldsymbol{F}_{\tau_k(i+1)}\boldsymbol{w}_i^k,$$

for $i = 0, \ldots, n-1$. In our worst-case example $f$, the updates can be compactly written as

$$\boldsymbol{z}_{i+1}^k = (\boldsymbol{I} - \beta_k \boldsymbol{A}_{\tau_k(i+1)} + \alpha_k \beta_k \boldsymbol{A}_{\tau_k(i+1)}^2)\boldsymbol{z}_i^k.$$

Since we have $n = 2$ and there are only two possible permutations, the updates over an epoch can be summarized as

$$\boldsymbol{z}_0^{k+1} = \boldsymbol{z}_n^k = \begin{cases} (\boldsymbol{I} - \beta_k \boldsymbol{A}_1 + \alpha_k \beta_k \boldsymbol{A}_1^2)(\boldsymbol{I} - \beta_k \boldsymbol{A}_2 + \alpha_k \beta_k \boldsymbol{A}_2^2)\boldsymbol{z}_0^k & \text{with probability } 1/2, \\ (\boldsymbol{I} - \beta_k \boldsymbol{A}_2 + \alpha_k \beta_k \boldsymbol{A}_2^2)(\boldsymbol{I} - \beta_k \boldsymbol{A}_1 + \alpha_k \beta_k \boldsymbol{A}_1^2)\boldsymbol{z}_0^k & \text{with probability } 1/2. \end{cases}$$

By the definition of $\boldsymbol{A}_1$ and $\boldsymbol{A}_2$ and using $\boldsymbol{A}_1^2 = \boldsymbol{A}_2^2 = \boldsymbol{0}$, we can verify that

$$\boldsymbol{M}_1 := (\boldsymbol{I} - \beta_k \boldsymbol{A}_1 + \alpha_k \beta_k \boldsymbol{A}_1^2)(\boldsymbol{I} - \beta_k \boldsymbol{A}_2 + \alpha_k \beta_k \boldsymbol{A}_2^2) = \begin{bmatrix} 1 - \frac{\beta_k^2 L^2}{2} & -\beta_k L - \frac{\beta_k^2 L^2}{2} \\ \beta_k L - \frac{\beta_k^2 L^2}{2} & 1 - \frac{\beta_k^2 L^2}{2} \end{bmatrix}, \quad (117)$$

$$\boldsymbol{M}_2 := (\boldsymbol{I} - \beta_k \boldsymbol{A}_2 + \alpha_k \beta_k \boldsymbol{A}_2^2)(\boldsymbol{I} - \beta_k \boldsymbol{A}_1 + \alpha_k \beta_k \boldsymbol{A}_1^2) = \begin{bmatrix} 1 - \frac{\beta_k^2 L^2}{2} & -\beta_k L + \frac{\beta_k^2 L^2}{2} \\ \beta_k L + \frac{\beta_k^2 L^2}{2} & 1 - \frac{\beta_k^2 L^2}{2} \end{bmatrix}. \quad (118)$$

From this, we notice that the expectation of $\left\|\boldsymbol{z}_0^{k+1}\right\|^2$ conditional on $\boldsymbol{z}_0^k$ reads

$$\mathbb{E}\left[\left\|\boldsymbol{z}_0^{k+1}\right\|^2 \,\Big|\, \boldsymbol{z}_0^k\right] = (\boldsymbol{z}_0^k)^\top \left(\frac{\boldsymbol{M}_1^\top \boldsymbol{M}_1 + \boldsymbol{M}_2^\top \boldsymbol{M}_2}{2}\right)\boldsymbol{z}_0^k.$$

Working out the calculations, we can check that

$$\frac{\boldsymbol{M}_1^\top \boldsymbol{M}_1 + \boldsymbol{M}_2^\top \boldsymbol{M}_2}{2} = \begin{bmatrix} 1 + \frac{\beta_k^4 L^4}{2} & 0 \\ 0 & 1 + \frac{\beta_k^4 L^4}{2} \end{bmatrix},$$

thus resulting in

$$\mathbb{E}\left[\left\|\boldsymbol{z}_0^{k+1}\right\|^2 \,\Big|\, \boldsymbol{z}_0^k\right] = \left(1 + \frac{\beta_k^4 L^4}{2}\right)\left\|\boldsymbol{z}_0^k\right\|^2.$$

Since this holds for all $k \geq 0$, SEG-RR diverges in expectation, for any positive stepsizes $\{\alpha_k\}_{k \geq 0}$ and $\{\beta_k\}_{k \geq 0}$. The statement on $\left\|\boldsymbol{F}\boldsymbol{z}_0^k\right\|$ follows by realizing that $\|\boldsymbol{F}\boldsymbol{z}\| = \frac{L}{2}\|\boldsymbol{z}\|$. $\qquad\square$

**Proposition G.3** (Part of Theorem 4.2). *For $n = 2$, there exists a convex-concave minimax problem $f(x, y) = \frac{1}{2}\sum_{i=1}^{2} f_i(x, y)$ having a monotone $\boldsymbol{F}$, consisting of $L$-smooth quadratic $f_i$'s satisfying Assumption 4 with $(\rho, \sigma) = (1, 0)$ such that SEG-FF diverges in expectation for any positive stepsizes $\{\alpha_k\}_{k\geq 0}$ and $\{\beta_k\}_{k\geq 0}$. That is, for any $k \geq 0$,*

$$\mathbb{E}\left[\left\|\boldsymbol{z}_0^{k+1}\right\|^2\right] > \mathbb{E}\left[\left\|\boldsymbol{z}_0^{k}\right\|^2\right], \quad \mathbb{E}\left[\left\|\boldsymbol{F}\boldsymbol{z}_0^{k+1}\right\|^2\right] > \mathbb{E}\left[\left\|\boldsymbol{F}\boldsymbol{z}_0^{k}\right\|^2\right].$$

*Proof.* The proof uses the same example as Proposition G.1, outlined in (116). We prove that SEG-FF also diverges for this $f$. For $k \geq 0$, the $(k+1)$-th epoch of SEG-FF starts at $\boldsymbol{z}_0^k$, and the algorithm randomly chooses a permutation $\tau_k : [n] \to [n]$, as in the case of SEG-RR. The algorithm then goes through a series of updates for $i = 0, \ldots, n - 1$:

$$\boldsymbol{w}_i^k = \boldsymbol{z}_i^k - \alpha_k \boldsymbol{F}_{\tau_k(i+1)}\boldsymbol{z}_i^k,$$
$$\boldsymbol{z}_{i+1}^k = \boldsymbol{z}_i^k - \beta_k \boldsymbol{F}_{\tau_k(i+1)}\boldsymbol{w}_i^k,$$

which are the same as SEG-RR; but then, it performs another series of $n$ updates, in the reverse order. For $i = n, \ldots, 2n - 1$,

$$\boldsymbol{w}_i^k = \boldsymbol{z}_i^k - \alpha_k \boldsymbol{F}_{\tau_k(2n-i)}\boldsymbol{z}_i^k,$$
$$\boldsymbol{z}_{i+1}^k = \boldsymbol{z}_i^k - \beta_k \boldsymbol{F}_{\tau_k(2n-i)}\boldsymbol{w}_i^k.$$

Using the definition of $\boldsymbol{M}_1$ and $\boldsymbol{M}_2$ from (117) and (118), one can verify that the $2n = 4$ updates over an epoch of SEG-FF can be summarized as

$$\boldsymbol{z}_0^{k+1} = \boldsymbol{z}_{2n}^k = \begin{cases} \boldsymbol{M}_2\boldsymbol{M}_1\boldsymbol{z}_0^k & \text{with probability } 1/2, \\ \boldsymbol{M}_1\boldsymbol{M}_2\boldsymbol{z}_0^k & \text{with probability } 1/2. \end{cases}$$

From this, we notice that the expectation of $\left\|\boldsymbol{z}_0^{k+1}\right\|^2$ conditional on $\boldsymbol{z}_0^k$ reads

$$\mathbb{E}\left[\left\|\boldsymbol{z}_0^{k+1}\right\|^2 \Big| \boldsymbol{z}_0^k\right] = (\boldsymbol{z}_0^k)^\top \left(\frac{\boldsymbol{M}_1^\top \boldsymbol{M}_2^\top \boldsymbol{M}_2 \boldsymbol{M}_1 + \boldsymbol{M}_2^\top \boldsymbol{M}_1^\top \boldsymbol{M}_1 \boldsymbol{M}_2}{2}\right) \boldsymbol{z}_0^k.$$

Working out the calculations, we can check that

$$\frac{\boldsymbol{M}_1^\top \boldsymbol{M}_2^\top \boldsymbol{M}_2 \boldsymbol{M}_1 + \boldsymbol{M}_2^\top \boldsymbol{M}_1^\top \boldsymbol{M}_1 \boldsymbol{M}_2}{2} = \begin{bmatrix} 1 + 2\beta_k^6 L^6 & 0 \\ 0 & 1 + 2\beta_k^6 L^6 \end{bmatrix},$$

thus resulting in

$$\mathbb{E}\left[\left\|\boldsymbol{z}_0^{k+1}\right\|^2 \Big| \boldsymbol{z}_0^k\right] = \left(1 + 2\beta_k^6 L^6\right)\left\|\boldsymbol{z}_0^k\right\|^2.$$

Since this holds for all $k \geq 0$, SEG-FF diverges in expectation, for any positive stepsizes $\{\alpha_k\}_{k\geq 0}$ and $\{\beta_k\}_{k\geq 0}$. The statement on $\left\|\boldsymbol{F}\boldsymbol{z}_0^k\right\|$ follows by realizing that $\|\boldsymbol{F}\boldsymbol{z}\| = \frac{L}{2}\|\boldsymbol{z}\|$. $\qquad\square$

### G.2 Proof of SGDA-RR and SEG-RR Lower Bounds

**Theorem G.4.** *Suppose $n \geq 2$ and $L, \mu > 0$ satisfies $L/\mu \geq 2$. There exists a $\mu$-strongly-convex-strongly-concave minimax problem $f(\boldsymbol{z}) = \frac{1}{n}\sum_{i=1}^{n} f_i(\boldsymbol{z})$ consisting of $L$-smooth quadratic $f_i$'s satisfying Assumption 4 with $(\rho, \sigma) = (0, \sigma)$ and initialization $\boldsymbol{z}_0^0$ such that SEG-RR with any constant stepsize $\alpha_k = \alpha > 0$, $\beta_k = \beta > 0$ satisfies*

$$\mathbb{E}\left[\left\|\boldsymbol{z}_0^K - \boldsymbol{z}^*\right\|^2\right] = \begin{cases} \Omega\left(\frac{\sigma^2}{L\mu nK}\right) & \text{if } K \leq L/\mu, \\ \Omega\left(\frac{L\sigma^2}{\mu^3 nK^3}\right) & \text{if } K > L/\mu. \end{cases}$$

*where $\boldsymbol{z}^*$ is the unique equilibrium point of $f$. For a similar choice of problem $f$ (this time with $(\rho, \sigma) = (1, \sigma)$), SGDA-RR with any constant stepsize $\alpha_k = \alpha > 0$ satisfies*

$$\mathbb{E}\left[\left\|\boldsymbol{z}_0^K - \boldsymbol{z}^*\right\|^2\right] = \begin{cases} \Omega\left(\frac{\sigma^2}{L\mu nK}\right) & \text{if } K \leq L/\mu, \\ \Omega\left(\frac{\sigma^2}{\mu^2 n^2 K^2} + \frac{L\sigma^2}{\mu^3 nK^3}\right) & \text{if } K > L/\mu. \end{cases}$$

**Remark.** In Theorem G.4, we adopt techniques from the existing lower bounds for SGD-RR to prove lower bounds for the minimax algorithms SGDA-RR and SEG-RR. In the literature, there are two types of lower bounds for SGD-RR when $K \gtrsim L/\mu$: $\Omega(\frac{1}{n^2 K^2} + \frac{1}{nK^3})$ bounds for strongly convex *quadratic* functions (Safran & Shamir, 2020; 2021) and $\Omega(\frac{1}{nK^2})$ bounds for strongly convex *non-quadratic* functions (Rajput et al., 2020; Yun et al., 2022; Cha et al., 2023). Upper bounds that match the lower bounds in $n$ and $K$ are also known, which indicates that SGD-RR is one of the rare examples of minimization algorithms whose tight convergence rates for quadratic vs. non-quadratic functions differ, within the narrow scope of strongly convex and smooth functions. While it is tempting to aim for a tighter $\Omega(\frac{1}{nK^2})$ lower bound for our algorithms of interest, we note that the existing $\Omega(\frac{1}{nK^2})$ bounds for SGD-RR are proven for piecewise-quadratic functions whose Hessian is *discontinuous*. Since the discontinuous Hessian violates our Assumption 3, we instead adhere to the quadratic case to prove lower bounds $\Omega(\frac{1}{nK^3})$ for both SGDA-RR and SEG-RR (when $K \geq L/\mu$). These bounds may not be the tightest possible (since they are restricted to the quadratics), but they still suffice to demonstrate that SEG-FFA is provably superior to both SGDA-RR and SEG-RR.

### G.2.1 EXISTING LOWER BOUND FOR SGD-RR

For the proof of lower bounds for SGDA-RR and SEG-RR, we utilize results and techniques from the lower bounds proven for SGD-RR; thus, it would be profitable to summarize the existing result.

In case of SGD-RR, it is known from Theorem 2 of Safran & Shamir (2021) that there exists a minimization problem $g(\boldsymbol{x})$ such that SGD-RR satisfies a lower bound of $\Omega(\frac{1}{n^2 K^2} + \frac{1}{nK^3})$ for large enough values of $K$. We rewrite the theorem in a version in accordance with our notation and assumptions:

**Theorem G.5** (Theorem 2 of Safran & Shamir (2021))**.** *For any $n \geq 2$ and $L, \mu > 0$ satisfying $L/\mu \geq 2$, there exists a $\mu$-strongly convex minimization problem $g(\boldsymbol{x}) = \frac{1}{n}\sum_{i=1}^{n} g_i(\boldsymbol{x})$ consisting of $L$-smooth quadratic $g_i$'s satisfying Assumption 4 with $(\rho, \sigma) = (1, \sigma)$ such that SGD-RR using any constant stepsize $\alpha_k = \alpha > 0$ satisfies*

$$\mathbb{E}\left[\left\|\boldsymbol{x}_0^K - \boldsymbol{x}^*\right\|^2\right] = \Omega\left(\frac{\sigma^2}{L\mu nK} \cdot \min\left\{1, \frac{L}{\mu nK} + \frac{L^2}{\mu^2 K^2}\right\}\right).$$

The statement is equivalent to saying that for SGD-RR with constant stepsize $\alpha > 0$, the bound $\Omega(\frac{\sigma^2}{L\mu nK})$ holds for $K \lesssim L/\mu$ and $\Omega(\frac{\sigma^2}{\mu^2 n^2 K^2} + \frac{L\sigma^2}{\mu^3 nK^3})$ for $K \gtrsim L/\mu$.

The function $g = \frac{1}{n}\sum_{i=1}^{n} g_i$ used in the theorem is defined by the following component functions:

$$g_i(\boldsymbol{x}) = g_i(x_1, x_2, x_3) := \frac{\mu}{2}x_1^2 + \frac{L}{2}x_2^2 + \begin{cases} \frac{\sigma}{2}x_2 + \frac{L}{2}x_3^2 + \frac{\sigma}{2}x_3 & i \leq \frac{n}{2}, \\ -\frac{\sigma}{2}x_2 - \frac{\sigma}{2}x_3 & i > \frac{n}{2}, \end{cases} \tag{119}$$

thus making the objective function

$$g(x_1, x_2, x_3) := \frac{\mu}{2}x_1^2 + \frac{L}{2}x_2^2 + \frac{L}{4}x_3^2.$$

One can notice that the linear terms in $g_i$ (119) change signs depending on $i \leq \frac{n}{2}$ or not, and handling these sign flips is the key to the proof of lower bound.

### G.2.2 PROOF OF LOWER BOUND FOR SGDA-RR

For the SGDA-RR lower bound, we consider the following minimax optimization problem:

$$f(\boldsymbol{x}, y) = \frac{1}{n}\sum_{i=1}^{n} f_i(\boldsymbol{x}, y), \text{ where } \boldsymbol{x} \in \mathbb{R}^3, \ y \in \mathbb{R},$$
$$f_i(\boldsymbol{x}, y) = g_i(\boldsymbol{x}) - \frac{\mu}{2}y^2, \tag{120}$$

where $g_i$'s are from (119). We need to first check if the problem instance satisfies the assumptions listed in the theorem statement. Since $f(\boldsymbol{x}, y) = g(\boldsymbol{x}) - \frac{\mu}{2}y^2$ and $g$ is a $\mu$-strongly convex function,

$f$ is $\mu$-strongly-convex-strongly-concave as claimed. Also, it is easy to check from the definition of $g_i$ that each component $f_i(\boldsymbol{x}, y)$ is $L$-smooth quadratic.

Lastly, to check Assumption 4, we first define $s_1, \ldots, s_n$ as $s_i = 1$ for $i \leq \frac{n}{2}$ and $s_i = 0$ for $i > \frac{n}{2}$. Using this notation, The function $g_i$ can be compactly written as the following:

$$g_i(x_1, x_2, x_3) = \frac{\mu}{2}x_1^2 + \frac{L}{2}x_2^2 + \frac{\sigma}{2}(2s_i - 1)x_2 + \frac{L}{2}s_i x_3^2 + \frac{\sigma}{2}(2s_i - 1)x_3.$$

Therefore, the saddle gradient operators $\boldsymbol{F}_i$ of $f_i$ and $\boldsymbol{F}$ of $f$ evaluate to

$$\boldsymbol{F}_i \boldsymbol{z} := \begin{bmatrix} \nabla g_i(\boldsymbol{x}) \\ \mu y \end{bmatrix} = \begin{bmatrix} \mu x_1 \\ L x_2 + \frac{\sigma}{2}(2s_i - 1) \\ L s_i x_3 + \frac{\sigma}{2}(2s_i - 1) \\ \mu y \end{bmatrix}, \quad \boldsymbol{F} \boldsymbol{z} = \begin{bmatrix} \mu x_1 \\ L x_2 \\ \frac{L}{2}x_3 \\ \mu y \end{bmatrix},$$

which in turn yields

$$\|\boldsymbol{F}_i \boldsymbol{z} - \boldsymbol{F} \boldsymbol{z}\|^2 = \frac{\sigma^2}{4} + \left(\frac{L}{2}x_3 + \frac{\sigma}{2}\right)^2 \leq \left(\frac{L}{2}|x_3| + \sigma\right)^2 \leq (\|\boldsymbol{F}\boldsymbol{z}\| + \sigma)^2$$

for all $i = 1, \ldots, n$. This confirms that the function $f = \frac{1}{n}\sum_i f_i$ satisfies Assumption 4 with $(\rho, \sigma) = (1, \sigma)$.

If we run SGDA-RR on this problem, the updates on $\boldsymbol{x}$ done by SGDA-RR is exactly identical to what SGD-RR would perform for the minimization problem $g(\boldsymbol{x}) = \frac{1}{n}\sum_i g_i(\boldsymbol{x})$ with the same choices of random permutations. Therefore, after $K$ epochs of SGDA-RR, it follows from Theorem G.5 that

$$\mathbb{E}\left[\left\|\boldsymbol{z}_0^K - \boldsymbol{z}^*\right\|^2\right] \geq \mathbb{E}\left[\left\|\boldsymbol{x}_0^K - \boldsymbol{x}^*\right\|^2\right] = \Omega\left(\frac{\sigma^2}{L\mu n K} \cdot \min\left\{1, \frac{L}{\mu n K} + \frac{L^2}{\mu^2 K^2}\right\}\right),$$

which is in fact a tighter lower bound for SGDA-RR than what is stated in Theorem G.4. This finishes the proof.

### G.2.3 PROOF OF LOWER BOUND FOR SEG-RR

In this subsection, we prove the lower bound for SEG-RR. We will first define a new problem instance $f$ to be used here, and verify that the assumptions in the theorem statement are indeed satisfied by this new $f$. We will then spell out the update equation of SEG-RR for this example, which will serve as a basis for the case analysis that follows: we will divide the choices of stepsizes $\alpha, \beta > 0$ to four regimes and prove a lower bound for each of them. Combining the regimes will result in the desired lower bound.

For SEG-RR, we use a slightly different problem from (120). This time, we consider

$$f(\boldsymbol{x}, y) = \frac{1}{n}\sum_{i=1}^n f_i(\boldsymbol{x}, y), \text{ where } \boldsymbol{x} \in \mathbb{R}^2, \ y \in \mathbb{R},$$

$$f_i(\boldsymbol{x}, y) = \frac{L}{2}x_1^2 + \frac{L}{4}x_2^2 + \sigma(2s_i - 1)x_2 - \frac{\mu}{2}y^2,$$

(121)

where $s_i = 1$ for $i \leq \frac{n}{2}$ and $s_i = 0$ for $i > \frac{n}{2}$, as defined above.

We first check if the problem (121) satisfies the assumptions in the theorem statement. Since

$$f(\boldsymbol{x}, y) = \frac{L}{2}x_1^2 + \frac{L}{4}x_2^2 - \frac{\mu}{2}y^2$$

and $L/2 \geq \mu$ by assumption, $f$ is $\mu$-strongly-convex-strongly-concave. Also, it is straightforward to see that each $f_i$ is an $L$-smooth quadratic function. It is left to check Assumption 4. The saddle gradient operators $\boldsymbol{F}_i$ of $f_i$ and $\boldsymbol{F}$ of $f$ evaluate to

$$\boldsymbol{F}_i \boldsymbol{z} = \begin{bmatrix} L x_1 \\ \frac{L}{2}x_2 + \sigma(2s_i - 1) \\ \mu y \end{bmatrix}, \quad \boldsymbol{F} \boldsymbol{z} = \begin{bmatrix} L x_1 \\ \frac{L}{2}x_2 \\ \mu y \end{bmatrix},$$

which in turn yields

$$\|\boldsymbol{F}_i \boldsymbol{z} - \boldsymbol{F} \boldsymbol{z}\|^2 = \sigma^2,$$

for all $i = 1, \ldots, n$. This confirms that the function $f = \frac{1}{n} \sum_i f_i$ satisfies Assumption 4 with $(\rho, \sigma) = (0, \sigma)$, as required by the theorem.

For $k \geq 0$, the $(k+1)$-th epoch of SEG-RR starts at $\boldsymbol{z}_0^k = (\boldsymbol{x}_0^k, y_0^k)$ and the algorithm chooses a random permutation $\tau_k$. The algorithm then goes through a series of updates

$$\boldsymbol{w}_i^k = \boldsymbol{z}_i^k - \alpha \boldsymbol{F}_{\tau_k(i+1)} \boldsymbol{z}_i^k,$$
$$\boldsymbol{z}_{i+1}^k = \boldsymbol{z}_i^k - \beta \boldsymbol{F}_{\tau_k(i+1)} \boldsymbol{w}_i^k,$$

for $i = 0, \ldots, n-1$. For our example $f$ (121), it can be checked that a single iteration by SEG-RR reads

$$\boldsymbol{z}_{i+1}^k = \begin{bmatrix} x_{i+1,1}^k \\ x_{i+1,2}^k \\ y_{i+1}^k \end{bmatrix} = \begin{bmatrix} (1 - \beta L + \alpha \beta L^2) x_{i,1}^k \\ (1 - \frac{\beta L}{2} + \frac{\alpha \beta L^2}{4}) x_{i,2}^k - \beta \sigma (1 - \frac{\alpha L}{2})(2 s_{\tau_k(i+1)} - 1) \\ (1 - \beta \mu + \alpha \beta \mu^2) y_i^k \end{bmatrix}.$$

Aggregating the SEG-RR updates over an entire epoch ($i = 0, \ldots, n-1$) results in

$$x_{0,1}^{k+1} = (1 - \beta L + \alpha \beta L^2)^n x_{0,1}^k,$$

$$x_{0,2}^{k+1} = \left(1 - \frac{\beta L}{2} + \frac{\alpha \beta L^2}{4}\right)^n x_{0,2}^k - \beta \sigma \left(1 - \frac{\alpha L}{2}\right) \underbrace{\sum_{i=1}^n (2 s_{\tau_k(i)} - 1) \left(1 - \frac{\beta L}{2} + \frac{\alpha \beta L^2}{4}\right)^{n-i}}_{=: \Phi},$$

$$y_0^{k+1} = (1 - \beta \mu + \alpha \beta \mu^2)^n y_0^k.$$

We will now square both sides of these equations above and take expectations over $\tau_k$. In doing so, there is a useful identity:

$$\mathbb{E}[\Phi] = \sum_{i=1}^n \mathbb{E}[2 s_{\tau_k(i)} - 1] \left(1 - \frac{\beta L}{2} + \frac{\alpha \beta L^2}{4}\right)^{n-i} = 0.$$

Also, it worth mentioning that $\tau_k$ is independent of $\boldsymbol{z}_0^k = (x_{0,1}^k, x_{0,2}^k, y_0^k)$. Using these facts, we can arrange the terms to obtain

$$(x_{0,1}^{k+1})^2 = (1 - \beta L + \alpha \beta L^2)^{2n} (x_{0,1}^k)^2, \tag{122}$$

$$\mathbb{E}[(x_{0,2}^{k+1})^2] = \left(1 - \frac{\beta L}{2} + \frac{\alpha \beta L^2}{4}\right)^{2n} \mathbb{E}[(x_{0,2}^k)^2] + \beta^2 \sigma^2 \left(1 - \frac{\alpha L}{2}\right)^2 \mathbb{E}[\Phi^2], \tag{123}$$

$$(y_0^{k+1})^2 = (1 - \beta \mu + \alpha \beta \mu^2)^{2n} (y_0^k)^2. \tag{124}$$

Based on these three per-epoch update equations above, we now divide the choices of SEG-RR stepsizes $\alpha, \beta > 0$ into the following four cases and handle them separately:

1. $\alpha > \frac{1}{L}$, in which case we show that SEG-RR makes $(x_{0,1}^{k+1})^2 > (x_{0,1}^k)^2$ hold deterministically, so that if we initialize at $x_{0,1}^0 = \frac{\sigma}{\sqrt{L\mu}}$ then we have

$$\mathbb{E}\left[\|\boldsymbol{z}_0^K\|^2\right] \geq (x_{0,1}^K)^2 > (x_{0,1}^0)^2 = \frac{\sigma^2}{L\mu}.$$

2. $\alpha \leq \frac{1}{L}$ and $\beta \leq \frac{1}{\mu n K}$, in which case we show that SEG-RR initialized at $y_0^0 = \frac{\sigma}{\sqrt{L\mu}}$ suffers

$$\mathbb{E}\left[\|\boldsymbol{z}_0^K\|^2\right] = \Omega\left(\frac{\sigma^2}{L\mu}\right),$$

3. $\alpha \leq \frac{1}{L}$ and $\frac{1}{\mu n K} < \beta < \frac{1}{nL}$, in which case we show that SEG-RR initialized at $x_{0,2}^0 = 0$ suffers

$$\mathbb{E}\left[\|\boldsymbol{z}_0^K\|^2\right] = \Omega\left(\frac{L\sigma^2}{\mu^3 n K^3}\right),$$

4. $\alpha \leq \frac{1}{L}$, $\beta > \frac{1}{\mu n K}$, and $\beta \geq \frac{1}{nL}$ in which case we show that SEG-RR initialized at $x_{0,2}^0 = 0$ suffers

$$\mathbb{E}\left[\|\boldsymbol{z}_0^K\|^2\right] = \Omega\left(\frac{\sigma^2}{L\mu n K}\right).$$

Notice that the third case $\frac{1}{\mu n K} < \beta < \frac{1}{nL}$ only makes sense when $K > L/\mu$; otherwise, the third case just disappears. Hence, for the "large epoch" regime where $K > L/\mu$, the third case achieves the minimum error possible, so it holds that

$$\mathbb{E}\left[\|\boldsymbol{z}_0^K\|^2\right] = \Omega\left(\frac{L\sigma^2}{\mu^3 n K^3}\right).$$

For the "small epoch" regime ($K \leq L/\mu$), the third case does not exist and the fourth case achieves the minimum, so

$$\mathbb{E}\left[\|\boldsymbol{z}_0^K\|^2\right] = \Omega\left(\frac{\sigma^2}{L\mu n K}\right).$$

Combining the two cases yields the desired lower bound in the theorem statement. It is now left to carry out the case analysis.

**Case 1: $\alpha > \frac{1}{L}$.** For this case, we use (122) to prove divergence. Notice from $\alpha > \frac{1}{L}$ that

$$1 - \beta L + \alpha\beta L^2 = 1 + \beta L(\alpha L - 1) > 1,$$

regardless of $\beta > 0$. Hence, from (122), we get

$$\mathbb{E}\left[\|\boldsymbol{z}_0^K\|^2\right] \geq (x_{0,1}^K)^2 > (x_{0,1}^0)^2.$$

If we initialize at $x_{0,1}^0 = \frac{\sigma}{\sqrt{L\mu}}$, then this proves

$$\mathbb{E}\left[\|\boldsymbol{z}_0^K\|^2\right] \geq \frac{\sigma^2}{L\mu}.$$

**Case 2: $\alpha \leq \frac{1}{L}$ and $\beta \leq \frac{1}{\mu n K}$.** For this case, we employ (124) to show that the "contraction rate" is too small to make enough "progress." Notice from our stepsizes that

$$1 - \beta\mu + \alpha\beta\mu^2 \geq 1 - \beta\mu \geq 1 - \frac{1}{nK} \geq 0.$$

Applying this inequality to (124), we have

$$(y_0^{k+1})^2 \geq \left(1 - \frac{1}{nK}\right)^{2n}(y_0^k)^2,$$

which in turn means that the progress over $K$ epoch is bounded from below by

$$(y_0^K)^2 \geq \left(1 - \frac{1}{nK}\right)^{2nK}(y_0^0)^2 \geq \frac{(y_0^0)^2}{16},$$

where we used our assumption that $n \geq 2$ and $K \geq 1$. Hence, if our initialization was given as $y_0^0 = \frac{\sigma}{\sqrt{L\mu}}$, then this proves

$$\mathbb{E}\left[\|\boldsymbol{z}_0^K\|^2\right] \geq (y_0^K)^2 \geq \frac{(y_0^0)^2}{16} = \Omega\left(\frac{\sigma^2}{L\mu}\right).$$

**Case 3: $\alpha \leq \frac{1}{L}$ and $\frac{1}{\mu n K} < \beta < \frac{1}{nL}$.** For stepsizes in this interval, we use (123) to derive the desired bound. Here, it is important to characterize a lower bound on the quantity

$$\mathbb{E}[\Phi^2] := \mathbb{E}\left[\left(\sum_{i=1}^n (2s_{\tau_k(i)} - 1)\left(1 - \frac{\beta L}{2} + \frac{\alpha\beta L^2}{4}\right)^{n-i}\right)^2\right].$$

To this end, we can use a lemma from Safran & Shamir (2020), stated below:

**Lemma G.6** (Lemma 1 of Safran & Shamir (2020)). *Let $\pi_1, \ldots, \pi_n$ (for even $n$) be a random permutation of $(1, 1, \ldots, 1, -1, -1, \ldots, -1)$ where both $1$ and $-1$ appear exactly $n/2$ times. Then there is a numerical constant $c > 0$ such that for any $\nu > 0$,*

$$\mathbb{E}\left[\left(\sum_{i=1}^n \pi_i (1-\nu)^{n-i}\right)^2\right] \geq c \cdot \min\left\{1 + \frac{1}{\nu}, n^3 \nu^2\right\}.$$

One can notice that Lemma G.6 is directly applicable to $\mathbb{E}[\Phi^2]$, with $\nu \leftarrow \frac{\beta L}{2} - \frac{\alpha\beta L^2}{4}$. Since

$$\nu = \frac{\beta L}{2} - \frac{\alpha\beta L^2}{4} \leq \frac{\beta L}{2} \leq \frac{1}{2n},$$

we have $n^3 \nu^2 \leq \frac{1}{8\nu}$, thereby

$$\min\left\{1 + \frac{1}{\nu}, n^3 \nu^2\right\} \geq \min\left\{\frac{1}{\nu}, n^3 \nu^2\right\} = n^3 \nu^2.$$

Therefore, Lemma G.6 gives

$$\mathbb{E}[\Phi^2] \geq cn^3 \left(\frac{\beta L}{2} - \frac{\alpha\beta L^2}{4}\right)^2 = \frac{c\beta^2 n^3 L^2}{4}\left(1 - \frac{\alpha L}{2}\right)^2 \geq \frac{c\beta^2 n^3 L^2}{16}, \tag{125}$$

where the last inequality used $\alpha \leq \frac{1}{L}$. Applying (125) to (123) and also using $(1 - \frac{\alpha L}{2})^2 \geq \frac{1}{4}$,

$$\mathbb{E}[(x_{0,2}^{k+1})^2] \geq \left(1 - \frac{\beta L}{2} + \frac{\alpha\beta L^2}{4}\right)^{2n} \mathbb{E}[(x_{0,2}^k)^2] + \frac{c\beta^4 n^3 L^2 \sigma^2}{64}.$$

Unrolling the inequality for $k = 0, \ldots, K-1$ gives

$$\mathbb{E}\left[(x_{0,2}^K)^2\right] \geq \left(1 - \frac{\beta L}{2} + \frac{\alpha\beta L^2}{4}\right)^{2nK} (x_{0,2}^0)^2 + \frac{c\beta^4 n^3 L^2 \sigma^2}{64} \sum_{j=0}^{K-1} \left(1 - \frac{\beta L}{2} + \frac{\alpha\beta L^2}{4}\right)^{2nj}$$

$$= \left(1 - \frac{\beta L}{2} + \frac{\alpha\beta L^2}{4}\right)^{2nK} (x_{0,2}^0)^2 + \frac{c\beta^4 n^3 L^2 \sigma^2}{64} \cdot \frac{1 - \left(1 - \frac{\beta L}{2} + \frac{\alpha\beta L^2}{4}\right)^{2nK}}{1 - \left(1 - \frac{\beta L}{2} + \frac{\alpha\beta L^2}{4}\right)^{2n}}.$$

Now note that our initialization $x_{0,2}^0$ can be set to zero, which eliminates the need to think about the first term in the RHS. It is now left to bound the second term. First, by the stepsize range $\alpha \leq \frac{1}{L}$, $\beta > \frac{1}{\mu nK}$ and our assumption $L/\mu \geq 2$, we have

$$\left(1 - \frac{\beta L}{2} + \frac{\alpha\beta L^2}{4}\right)^{2nK} \leq \left(1 - \frac{\beta L}{4}\right)^{2nK} \leq \left(1 - \frac{L}{4\mu nK}\right)^{2nK} \leq e^{-\frac{L}{2\mu}} \leq e^{-1}.$$

Next, by Bernoulli's inequality

$$\left(1 - \frac{\beta L}{2} + \frac{\alpha\beta L^2}{4}\right)^{2n} \geq \left(1 - \frac{\beta L}{2}\right)^{2n} \geq 1 - \beta n L > 0.$$

Plugging in the two inequalities to above, we obtain

$$\mathbb{E}\left[(x_{0,2}^K)^2\right] \geq \frac{c\beta^4 n^3 L^2 \sigma^2}{64} \cdot \frac{1 - \left(1 - \frac{\beta L}{2} + \frac{\alpha\beta L^2}{4}\right)^{2nK}}{1 - \left(1 - \frac{\beta L}{2} + \frac{\alpha\beta L^2}{4}\right)^{2n}}$$

$$\geq \frac{c\beta^4 n^3 L^2 \sigma^2}{64} \cdot \frac{1 - e^{-1}}{1 - (1 - \beta n L)} = c'\beta^3 n^2 L \sigma^2$$

for a numerical constant $c' > 0$. Plugging in the lower bound $\beta > \frac{1}{\mu nK}$ yields

$$\mathbb{E}\left[\|z_0^K\|^2\right] \geq \mathbb{E}\left[(x_{0,2}^K)^2\right] = \Omega\left(\frac{L\sigma^2}{\mu^3 nK^3}\right).$$

**Case 4:** $\alpha \leq \frac{1}{L}, \beta > \frac{1}{\mu n K}$**, and** $\beta \geq \frac{1}{nL}$**.** We again use (123). By noticing that the initialization $x_{0,2}^0 = 0$, we can unroll (123) for $k = 0, \ldots, K-1$ to get

$$\mathbb{E}\left[(x_{0,2}^K)^2\right] \geq \frac{\beta^2 \sigma^2}{4} \mathbb{E}[\Phi^2] \sum_{j=0}^{K-1} \left(1 - \frac{\beta L}{2} + \frac{\alpha \beta L^2}{4}\right)^{2nj} \geq \frac{\beta^2 \sigma^2}{4} \mathbb{E}[\Phi^2], \qquad (126)$$

where the last inequality holds regardless of $\beta$ because each summand with $j \geq 1$ is nonnegative. We then invoke Lemma G.6 to lower bound $\mathbb{E}[\Phi^2]$, again with $\nu \leftarrow \frac{\beta L}{2} - \frac{\alpha \beta L^2}{4}$. Since

$$\nu = \frac{\beta L}{2} - \frac{\alpha \beta L^2}{4} \geq \frac{\beta L}{4} \geq \frac{1}{4n},$$

we have $n^3 \nu^2 \geq \frac{1}{64\nu}$, thereby

$$\min\left\{1 + \frac{1}{\nu}, n^3 \nu^2\right\} \geq \min\left\{\frac{1}{\nu}, n^3 \nu^2\right\} \geq \frac{1}{64\nu}.$$

Therefore, Lemma G.6 gives

$$\mathbb{E}[\Phi^2] \geq \frac{c}{64\nu} = \frac{c}{32\beta L} \cdot \frac{1}{1 - \frac{\alpha L}{2}} \geq \frac{c}{32\beta L}. \qquad (127)$$

Combining (127) with (126) gives

$$\mathbb{E}\left[(x_{0,2}^K)^2\right] \geq \frac{c\beta \sigma^2}{128 L} = \Omega\left(\frac{\sigma^2}{L \mu n K}\right),$$

where the last step used $\beta > \frac{1}{\mu n K}$. This finishes the case analysis, hence the proof of Theorem G.4.

# H  EXPERIMENTS

To evaluate our algorithm SEG-FFA as well as other baseline algorithms, we conduct numerical experiments on monotone and strongly monotone problems. Specifically, we consider a random quadratic problem of the form

$$\min_{\boldsymbol{x} \in \mathbb{R}^{d_x}} \max_{\boldsymbol{y} \in \mathbb{R}^{d_y}} \frac{1}{n} \sum_{i=1}^{n} \begin{bmatrix} \boldsymbol{x} \\ \boldsymbol{y} \end{bmatrix}^\top \begin{bmatrix} \boldsymbol{A}_i & \boldsymbol{B}_i \\ \boldsymbol{B}_i^\top & -\boldsymbol{C}_i \end{bmatrix} \begin{bmatrix} \boldsymbol{x} \\ \boldsymbol{y} \end{bmatrix} - \boldsymbol{t}_i^\top \begin{bmatrix} \boldsymbol{x} \\ \boldsymbol{y} \end{bmatrix}. \qquad (128)$$

We also denote $\boldsymbol{z} = \begin{bmatrix} \boldsymbol{x}^\top & \boldsymbol{y}^\top \end{bmatrix}^\top$. We choose $d_x = d_y = 20$ and $n = 40$ for experiments below.

**Monotone case & Ablation study on the anchoring step**  For an experiment for the monotone case, the random components are sampled as follows. We choose $\boldsymbol{B}_i$ so that each element is an i.i.d. sample from a uniform distribution over the interval $[0, 1]$, and $\boldsymbol{t}_i$ so that each element is an i.i.d. sample from a standard normal distribution. We chose $\boldsymbol{A}_i$ to be diagonal matrices in the following procedure: for each $j = 1, \ldots, 20$ we randomly chose a subset $\mathcal{I}_j$ of $\frac{n}{2} = 20$ indices from $[n] = \{1, \ldots, 40\}$, and set the $(j, j)$-entry of $\boldsymbol{A}_i$ to be

$$(\boldsymbol{A}_i)_{j,j} = \begin{cases} 2 & \text{if } i \in \mathcal{I}_j \\ -2 & \text{otherwise} \end{cases}.$$

We repeat the exact same procedure for $\boldsymbol{C}_i$ as well. Notice that $\sum_{i=1}^{n} \boldsymbol{A}_i = \sum_{i=1}^{n} \boldsymbol{C}_i = \boldsymbol{0}$ by design. Hence, each of the component functions will be a nonconvex-nonconcave quadratic function in general, but the objective function itself becomes a convex-concave function.

We compare the performance of the various SEGs, namely SEG-FFA, SEG-FF, SEG-RR, and SEG-US. In addition, as an ablation study on the anchoring technique, we also compare *SEG-RRA* and *SEG-USA*, which are each SEG-RR and SEG-US with an additional anchoring step, respectively. For these two methods, we take the anchoring step after every $n$ iterations. We ran the experiment on 5 random instances of (128) with stepsizes $\eta_k = \eta_0/(1+k/10)^{0.34}$ where $\eta_0 = \min\{0.01, \frac{1}{L}\}$ for SEG-FFA, and $\alpha_k = \beta_k = \eta_k$ for the remaining methods. The exponent 0.34

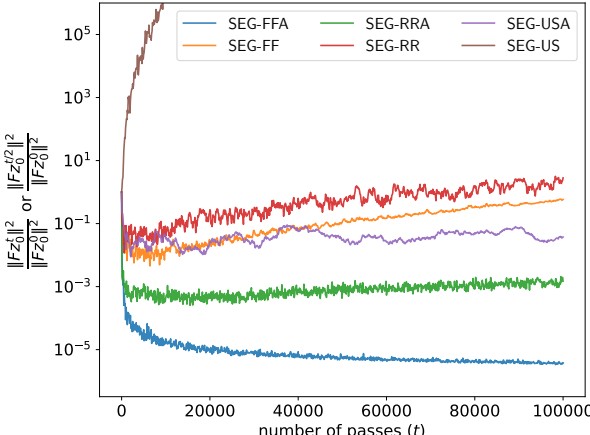

Figure 1: Experimental results in the monotone example, comparing the variants of SEG. For a fair comparison, we take the number of passes as the abscissae. As a consequence of SEG-FFA and SEG-FF using two passes per epoch, for those two methods, we get to plot $\|\boldsymbol{F}\boldsymbol{z}_0^{t/2}\|^2/\|\boldsymbol{F}\boldsymbol{z}_0^0\|^2$.

is to ensure a sufficient decay rate following Theorem 5.4, but the value of $\eta_0$ is a heuristically determined small number.

The results are plotted in Figure 1. To have a fair comparison between the methods and across the random instances, we compute the ratio $\dfrac{\|\boldsymbol{F}\boldsymbol{z}_0^t\|^2}{\|\boldsymbol{F}\boldsymbol{z}_0^0\|^2}$ where $t$ denotes the number of passes, and plot the geometric mean over the 5 runs. Notice that, for SEG-FFA and SEG-FF, we are then plotting the values of $\dfrac{\|\boldsymbol{F}\boldsymbol{z}_0^{t/2}\|^2}{\|\boldsymbol{F}\boldsymbol{z}_0^0\|^2}$ instead of $\dfrac{\|\boldsymbol{F}\boldsymbol{z}_0^t\|^2}{\|\boldsymbol{F}\boldsymbol{z}_0^0\|^2}$, because SEG-FFA and SEG-FF takes two passes per epoch (*i.e.*, the number of epochs is half the number of passes) while other methods take one pass per epoch.

As it is predictable from our theoretical results, SEG-FFA successfully shows convergence, while all of SEG-FF, SEG-RR, and SEG-US diverge in the long run. As for anchoring, it turns out that adding the anchoring step does improve the performance of the method up to a certain level, but it alone does not fully resolve the nonconvergence issue: observe that both SEG-RRA and SEG-USA fail to demonstrate convergence.

**Monotone case: comparison with (Hsieh et al., 2020)** Let us also compare the performance of SEG-FFA with the *independent-sample* double stepsize SEG (*DSEG*) by Hsieh et al. (2020). Writing in terms of the finite-sum structure, the update rule of DSEG can be written as

$$\boldsymbol{w}^k \leftarrow \boldsymbol{z}^k - \eta_{1,k}\boldsymbol{F}_{i(1,k)}\boldsymbol{z}^k$$
$$\boldsymbol{z}^{k+1} \leftarrow \boldsymbol{z}^k - \eta_{2,k}\boldsymbol{F}_{i(2,k)}\boldsymbol{w}^k$$

where $i(1,k)$ and $i(2,k)$ are random indices that are independently drawn from $[n]$ for each $k$. The stepsizes are chosen in the form of $\eta_{1,k} = \Theta(1/k^{r_1})$ and $\eta_{2,k} = \Theta(1/k^{r_2})$, where setting $r_1 \leq r_2$ is the key point of DSEG. Two choices of the exponent pair $(r_1, r_2)$ proposed in (Hsieh et al., 2020) are $(1/3, 2/3)$ for general monotone problems and $(0, 1)$ exclusively for the case when $\boldsymbol{F}$ is affine.

We again use the same component functions as in the previous experiment. The setup for running SEG-FFA are kept the same. For DSEG, we found that $\eta_{1,k} = \eta_0/(k+19)^{r_1}$ and $\eta_{2,k} = \eta_0/(k+19)^{r_2}$, where again $\eta_0 = \min\{0.01, \frac{1}{L}\}$, works the best among the candidates we have tried (and the choices in (Hsieh et al., 2020)). The number 19 in the denominators is the constant suggested in the experiments section of (Hsieh et al., 2020).

The results are displayed in Figure 2, where the details on how the plots are drawn are the same as Figure 1. Here we can see that SEG-FFA shows also a faster speed of convergence than both versions of DSEG.

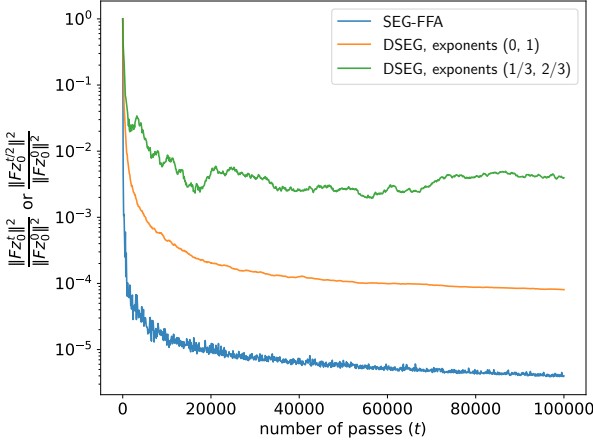

Figure 2: Experimental results in the monotone example, comparing SEG-FFA and the methods proposed by Hsieh et al. (2020). By the same reason as in Figure 1, we plot $\|\boldsymbol{F}\boldsymbol{z}_0^{t/2}\|^2/\|\boldsymbol{F}\boldsymbol{z}_0^0\|^2$ for SEG-FFA only.

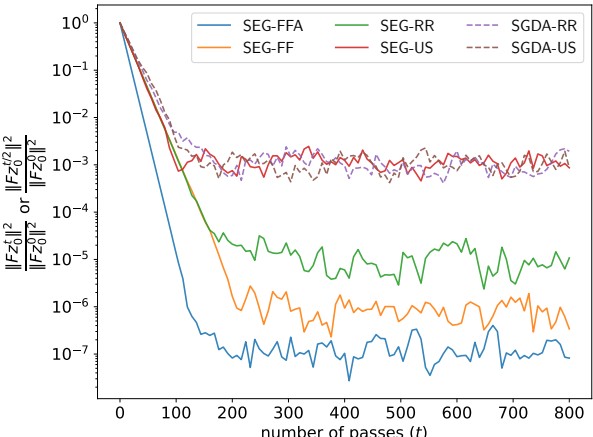

Figure 3: Experimental results in the strongly monotone example. By the same reason as in Figure 1, we plot $\|\boldsymbol{F}\boldsymbol{z}_0^{t/2}\|^2/\|\boldsymbol{F}\boldsymbol{z}_0^0\|^2$ for SEG-FFA and SEG-FF.

**Strongly monotone case** For the experiment in the strongly monotone case we again use the random quadratic problem (128), but with different choices of $\boldsymbol{A}_i$ and $\boldsymbol{C}_i$ to ensure the objective function to be strongly-convex-strongly-concave. In particular, for each $i = 1, \ldots, n$, we sample $\boldsymbol{A}_i$ by computing $\boldsymbol{A}_i = \boldsymbol{Q}_i \boldsymbol{D}_i \boldsymbol{Q}_i^\top$, where $\boldsymbol{D}_i$ is a random diagonal matrix whose diagonal entries are i.i.d. samples from a uniform distribution over the interval $[\frac{1}{2}, 1]$, and $\boldsymbol{Q}_i$ is a random orthogonal matrix obtained by computing a $QR$ decomposition of a $20 \times 20$ random matrix whose elements are i.i.d. samples from a standard normal distribution. We sample $\boldsymbol{C}_i$ by the exact same method.

As we are considering strongly-convex-strongly-concave problems, along with the variants of SEG, we also compare the performances of SGDA-RR and SGDA-US. We ran the experiment on 5 random instances of (128) with stepsizes $\eta_k \equiv 0.001$.

The results are plotted in Figure 3, where the details of the plotting are the same as the monotone cases. We again observe an agreement between the empirical result and our theory; all methods used in the experiment are expected to find a point with a reasonably small gradient, but nonetheless, the fastest decrease of the gradient norm is demonstrated by SEG-FFA. Moreover, we see that SEG-FFA eventually finds the point with the smallest gradient norm among the methods that are considered.

