# OpenReview forum: "Stochastic Extragradient with Flip-Flop Shuffling & Anchoring: Provable Improvements"
_ICLR.cc/2024/Conference — Submitted to ICLR 2024_

### Official Review · Reviewer_ojxo · 2023-10-30

**Soundness:** 2 fair
**Presentation:** 2 fair
**Contribution:** 2 fair
**Rating:** 3
**Confidence:** 4

**Summary:**

The paper proposes to combine stochastic extragradient with flip-flop shuffling and anchoring for solving convex-concave finite-sum minimax problems. Under the assumption that the objective is smooth and Hessian is Lipschitz, the derived algorithm is claimed to have better performance in both theory and some preliminary numerical experiments on a test function, compared to vanilla stochastic gradient descent ascent and extragradient, measured by the gradient norm.

**Strengths:**

The paper studies shuffling-based stochastic extragradient for convex-concave minimax problems. This setting is new and not examined before to the best of my knowledge. The paper has a unified analysis for different sample schemes (Theorem 6.1 & 6.2). The numerical experiments align with the theory.

**Weaknesses:**

1. The paper does not explain why they use the gradient norm as the convergence measure and fails to mention a line of research studying that. In convex-concave minimax problems, the standard convergence measure is the duality gap $\max_y f(\bar x,y) - \min_x f(x,\bar y)$ or minty solution $F(x)^\top (\bar x - x)$ in the sense of variational inequality. This is natural since the optimal point of interest is the Nash equilibrium or the saddle point. There does exist a line of research studying monotone inclusion or gradient norm convergence in the same setting, e.g., Halpern iteration [arXiv:2002.08872, arXiv:2203.09436] or anchoring [arXiv:1905.10899, arXiv:2102.07922, arXiv:2106.02326], but the paper does not compare with them. Specifically, the algorithm in [arXiv:2203.09436] achieves $E\Vert F z_k\Vert^2 \leq 1/k^2$ (see Theorem 4.1) with less assumptions (no Hessian Lipschitzness). The idea of combining extragradient with anchoring is also not new [arXiv:2102.07922].

2. The paper keeps mentioning that GDA and stochastic extragradient do not converge for convex-concave minimax problems and claims that one important contribution is to fix the non-convergence problem. However, for convex-concave minimax problems, it is easy to show that the average iterate of (stochastic) GDA converges with the rate $1/k^{1/2}$ measured by the duality gap, without additional smoothness or Hessian Lipschitzness assumptions (even though the last-iterate does diverge). I am also confused by the statement that previous works (Mishchenko, 2020b & Diakonikolas, 2021) on stochastic extragradient only guarantee convergence to a neighborhood. To my best understanding, these works are able to show $E\Vert F(x)\Vert\leq\epsilon$ (Theorem 4.4 of Diakonikolas, 2021) or $E[F(x)^\top (\bar x - x)]\leq\epsilon$ (Theorem 3 of Mishchenko, 2020b). This paper can be misleading and overclaiming.

3. The paper seems to claim that the bounded domain assumption is strong and thus studies the unconstrained setting. However, according to the well-known von Neumann's minimax theorem, when the objective is convex-concave, the convex and compact domain assumption is necessary to ensure the existence of the Nash equilibrium. When the domain is $R^d$, Assumption 2 is questionable, and it's not clear what solution the algorithm finds when $\Vert Fz\Vert\to 0$.

4. The assumption that Hessian is Lipschitz seems to be too strong and unusual for the considered setting. The algorithm studied in the paper is neither high-order nor has a last-iterate convergence guarantee (Theorem 5.4).

5. The convergence of SEG-RR and SEG-FF is strictly worse than SGDA-RR in the strongly monotone setting. Could the authors explain why?

6. Proposition 5.2 says the reason why anchoring is also required to match (13). Then what changes after anchoring is introduced? How does each term in (13) match? Its LHS is of order $\eta_1\eta_2/n^2$ while the RHS is of order $\eta^2$. Does this mean $\eta_1 = \eta n$ also holds? Isn't $\eta_2=2\eta_1$ still true? Moreover, $z_0-\eta n F(z_0 - \eta n F(z_0))$ in Proposition 5.3 is just one step of EG from $z_0$. Does it suggest that $N$ steps of SEG-FFA are equivalent to one step of EG, up to a small error, where $N$ can be multiple of $n$? Is this specifically for $N=2n$?

7. The paper should also include SGDA-US/RR with anchoring and SEG-US/RR with anchoring, at least in experiments, to show that both ingredients of the algorithms are necessary. What if anchoring is the key step that makes everything converge? I do not fully believe that FF is something that should be done in practice, given that its computation is twice as much as the standard method in every epoch.

8. (Minor) Could the author comment on the dependence on the condition number when the operator is strongly monotone and compare it with existing works? It could be more clear if the author explicitly listed all the necessary assumptions required for each theorem. Are Assumptions 1(1'), 2, 3, and 4 necessary for every theorem? The paper should also clearly state that the benefit of shuffling-based methods over uniform sampling only holds when $K>1$, i.e., the multi-pass case.

**Questions:**

See Weaknesses.

---

> ### Author Response · Authors · 2023-11-22
> **Official Comment by Authors (1/3)**
>
> We appreciate the reviewer for the constructive feedback and thoughtful comments
>
> 1. **Gradient norm as an optimality measure, and Nash equilibria (W1, W3):**
>
>     We would like to first point out that for (unconstrained) convex-concave problems, a point $(\boldsymbol{x}^∗, \boldsymbol{y}^∗)$ is a first-order stationary point if and only if it is a global Nash equilibrium, since $f(\boldsymbol{x}, \boldsymbol{y}^*) \geq f(\boldsymbol{x}^*, \boldsymbol{y}^*) + \langle \nabla_{\boldsymbol{x}} f(\boldsymbol{x}^*, \boldsymbol{y}^*), \boldsymbol{x} - \boldsymbol{x}^* \rangle$ and $f(\boldsymbol{x}^*, \boldsymbol{y}) \leq f(\boldsymbol{x}^*, \boldsymbol{y}^*) + \langle \nabla_{\boldsymbol{y}} f(\boldsymbol{x}^*, \boldsymbol{y}^*), \boldsymbol{y} - \boldsymbol{y}^* \rangle$.
>
>     While we agree that the squared gradient norm is not an adequate measure of convergence for *constrained* problems (as also pointed out in Section 1.1 of (Gorbunov et al., 2022b)), taking the squared gradient norm as an optimality measure is not unnatural in *unconstrained* problems. We would also like to note that Gorbunov et al. (2022b) study the convergence of EG in the unconstrained case, using the squared gradient norm as the optimality measure.
>
>     We would like to also note that the convex and compact domain assumption in von Neumann’s minimax theorem is a *sufficient* condition for the existence of a Nash equilibrium, but not a *necessary* condition. For example, consider a simple bilinear problem with $f (x, y) = xy$. This has a unique first-order stationary point $(0, 0)$, which is a global Nash equilibrium. For this reason, we believe that Assumption 2 is natural, as it is analogous to assuming the existence of a minimum in convex minimization.
>
> 2. **Existing results on anchoring (W1):**
>
>     We appreciate the reviewer for bringing up the existing results that adopt similar ideas of anchoring, which was not explained well. First of all, we have clarified in footnote 1 that our anchoring and the Halpern-based anchoring are different in their formulations. In addition, their motivations differ as the former (ours) is for the *second-order matching* that makes SEG converge in the convex-concave cases, while the latter is for acceleration in rate. Therefore, we would like to claim that our anchoring approach is new, in the sense that we adopt the idea for making SEG converge in the first place. The  (Cai et al., 2022) [arXiv:2203.09436] paper you kindly mentioned introduces the Halpern-based anchoring to SEG for acceleration, but it requires increasing the batch size after each epoch, on top of the requirement that the variance is uniformly bounded over the domain, which are quite restrictive compared to our setting. This is now stated in Section 2.
>
> 3. **Stochastic GDA (W2):**
>
>     We are not sure which paper the reviewer is referring to, and we would appreciate it if the reviewer could give us a pointer; nevertheless, we suspect that the mentioned convergence guarantee might not apply to our unconstrained convex-concave setting. This is because, in the *unconstrained* $\min_x \max_y f (x, y) = xy$ case, the duality gap $\max_y f (\bar{x}, y) − \min_x f (x, \bar{y})$ is always $\infty$ unless $(\bar{x}, \bar{y}) = (0, 0)$. Hence, duality gap results should require the assumption of bounded domain. Nevertheless, we would be happy to learn about the reference that discusses the ergodic convergence of the stochastic GDA.

---

> ### Author Response · Authors · 2023-11-22
> **Official Comment by Authors (2/3)**
>
> 4. **Statement regarding convergence to neighborhood of an optimum (W2)**
>
>     The reviewer is correct that both (Theorem 4.4, Diakonikolas et al., 2021) and (Theorem 3, Mishchenko et al., 2020b) show that they can find an $\epsilon$-stationary point. However, what we originally meant by “convergence to neighborhood” was that the analyses do not imply that the methods will reach an optimum as $k \to \infty$, in the sense that the optimality measure goes all the way to zero as $k \to \infty$. (Here $k$ denotes the number of iterations.) In particular, by taking a close look at the proof of (Theorem 4.4, Diakonikolas et al., 2021), one can see that we need to choose the number of samples per iteration that is greater than $\mathcal{O}(1/\epsilon)$ to guarantee finding an $\epsilon$-stationary point, and to guarantee reaching an optimum one needs $n \to \infty$ in the proof. In addition, (Theorem 3, Mishchenko et al., 2020b) presents a rate bound for a constant step size $\mathcal{O}(1/\sqrt{t})$ for a fixed budget of $t$ iterations, hence cannot handle the $t \to \infty$ case. Nevertheless, we learned that this is subtle and may confuse the readers. So, we decided to emphasize more on our other advantages over existing literature (especially those brought up by the reviewers) in unconstrained cases, and do not argue on the aspect of the convergence to neighborhood.
>
>     We also would like to note that, as now explicitly stated in our Section 2, Diakonikolas et al. (2021) assume the variance to be uniformly bounded over the domain, which is more restrictive than our setting. In addition, Mishchenko et al. (2020b) require the domain to be bounded (as they take the supremum over the entire domain in the final bound) and the stochastic component to be monotone almost surely.
>
> 5. **Lipschitz Hessian assumption (W4):**
>
>     For our comments for the Lipschitz Hessian assumption, we kindly ask the reviewer to refer to the general comments. We would also like to point out that our results in the strongly monotone setting are guarantees on the last iterate.
>
> 6. **Convergence rates of SEG-RR and SEG-FF (W5):**
>
>     It is indeed true that our convergence rates $\mathcal{O} ( \frac{1}{n K^{2-3\epsilon}} )$ for SEG-RR and SEG-FF are slightly slower than the known $\tilde{\mathcal{O}} ( \frac{1}{nK^2} )$ of SGDA-RR. We believe that it is an artifact of our attempt on establishing a unified analysis (Theorem 6.2), and it is not a serious limitation since $\epsilon$ is a number that we can choose arbitrarily small. Also, our main focus is to show the effectiveness of SEG-FFA, which attains a strictly improved rate over SGDA and other SEG variants on both strongly-monotone and monotone problems, with provable gaps (recall Theorems 4.2 and 5.6 in particular).
>
> 7. **On the questions regarding Section 5 (W6):**
>
>     Proposition 5.3 states that after introducing the anchoring step with $\theta =1$, choosing $\beta = \eta$ and $\alpha = \eta/2$ makes the LHS and RHS terms in (13) (which is now (15) in the revised manuscript) to match with "$\eta_1 = \eta_2 = n\eta$". This is not easy to verify because, in case of filp-flop sampling ($N = 2n$), the sum $\sum_{0 \leq i < j \leq N-1} D\boldsymbol{T}_j (\boldsymbol{z}_0) \boldsymbol{T}_i (\boldsymbol{z}_0)$ on the RHS contains both of the sums in the LHS of (13). So, we would like to refer the reviewer to Lemma C.4 for more exact calculations.
>
>     Here, the reviewer is correct on observing that the overall update made in the FFA epoch (consisting of $2n$ steps and the anchoring) is equivalent to a single step of EG plus some small noise. This is in fact the key idea in many existing analysis of shuffling-based methods: aggregate the stochastic updates over an epoch and write them as a single deterministic update plus small noise. For SEG-FFA we have $N = 2n$ because one epoch contains $2n$ updates, but we chose this general notation $N$ because other methods may have different values of $N$; for example, we have $N = n$ for SEG-RR. We have further clarified this in our revised Section 5.
>
> 8. **Anchoring but without FF (W7):**
>
>     According to our analyses demonstrated in Section 5, adding anchoring on US/RR will not show a decisive improvement, because anchoring by itself cannot achieve second-order matching. Nevertheless, we performed an ablation study testing the effect of anchoring applied to SEG-US and SEG-RR; please see Appendix H. We have not considered applying anchoring to SGDA-US and SGDA-RR, since it is obvious that they diverge for a simple bilinear finite-sum problem when $f_1(x, y) = f_2(x, y) = \dots = f_n(x, y) = xy$.
>
>     As for the computational burden caused by flip-flop shuffling, one can simply reduce the number of epochs by half. So, in our experiments, we make comparisons between RR and FF schemes with respect to the number of passes through the components.

---

> ### Author Response · Authors · 2023-11-22
> **Official Comment by Authors (3/3)**
>
> 9. **Three minor comments (W8):**
>
>     We agree that it is best to include the dependency on the condition number, but for the rebuttal period we had to focus on addressing other major issues. We will make this clear in our next revision.
>
>     In the initial submission, some theorems did not explicitly include the assumptions used. As correctly guessed by the reviewer, Assumptions 1–4 are used in most of the theorems. We have modified the statements accordingly.
>
>     We added "when $K$ is large enough" below Theorem 4.1.

---

### Official Review · Reviewer_ZXRs · 2023-11-01

**Soundness:** 3 good
**Presentation:** 3 good
**Contribution:** 3 good
**Rating:** 8
**Confidence:** 3

**Summary:**

It is known that the random reshuffling can often improve stochastic gradient descent methods. The paper considers the case of stochastic extragradient method (SEG) for minimax optimization and proposes SEG with flip-flop anchoring as the random reshuffling counterpart. The methods are designed based on a second-order matching arguments. The paper proves that the method converges under general assumptions and has superior convergence rate than SEG.

**Strengths:**

I think the paper is good and has some novelties. The flip-flop shuffling is only analyzed in the basic quadratic case in the previous literature. So the analysis in this paper will be very useful for future extentions. The idea of using anchoring to improve the convergence is also quite impressive. I did not check all the prove details, but the results seem fairly reasonable for me.

**Weaknesses:**

See questions.

**Questions:**

The experiments section should be expanded or put into the appendix. Details of the experiments should be provided. The last sentence "suggest that the convergence rate derived in this paper may have room for improvement" is confusing. Can the authors elaborate on this?

My second question is that can the analysis of flip-flop shuffling be applied to other settings? As far as I know, this is the first time it is analyzed beyond quadratic setting, so I am interested if the analysis can be applied to usual convex optimization cases.

---

> ### Author Response · Authors · 2023-11-22
>
> We appreciate the reviewer for the positive feedback and thoughtful comments.
>
> 1. **On the experiments (Q1):**
>
>     As suggested, we expanded the experiment section with details, and moved it to Appendix H.
>
> 2. **Clarification on the last sentence in Section 7 (Q1):**
>
>     We entirely rewrote the experiment section to accommodate the reviewer’s suggestions, and the last sentence in Section 7 is now removed. Anyway, by the last sentence in Section 7, we were pointing to the observation that even though the step size chosen according to our analysis successfully converges, the algorithm also converges with much larger choices of step size. This behavior suggests that maybe the convergence of SEG-FFA can be shown for a larger step size and hence with a faster rate; this is what we meant by "room for improvement".
>
> 3. **Flip-flop shuffling to other problems (Q2):**
>
>     Since a minimization problem can be viewed a special case of a minimax problem whose objective function is constant with respect to the maximization variable, at the very least, our results should be directly applicable to convex minimization problems. Still, our arguments and proofs are tailored to minimax problems, so investigating whether one can improve upon our results for minimization problems would be an interesting future work.

---

> > ### Comment · Reviewer_ZXRs · 2023-12-05
> > **Updated Review**
> >
> > I agree with other reviewers that there are a lot of minor details that can be improved. But I still stand by my opinion that the analysis for flip-flop shuffling is a novel contribution and this paper should be considered to be accepted.

---

### Official Review · Reviewer_emGH · 2023-11-03

**Soundness:** 2 fair
**Presentation:** 3 good
**Contribution:** 2 fair
**Rating:** 3
**Confidence:** 4

**Summary:**

The paper focuses on shuffling-based SEG for solving finite-sum min-max optimization problems. In particular, it studies the convergence of the method in a strongly monotone and monotone regime. The proposed analysis reveals that both random reshuffling and the flip-flop shuffling alone cannot fix the non-convergence issue of classical SEG in Convex-Cocave problems. By using anchoring on top of the algorithms, the authors develop the SEG with the flip-flop anchoring (SEG-FFA) method, which successfully converges in Convex-Concave problems.

**Strengths:**

The paper is well-written, and the main contributions are clear. To the best of my knowledge, this is one of the first papers that provides an analysis of random reshuffling variants for solving min-max problems. Existing variants of the random reshuffling method, as correctly pointed out by the authors, focus on the Gradient descent ascent method. Having an analysis of a shuffling-based SEG is a great next step in this literature.

**Weaknesses:**

However, I believe this work has some misleading statements, and the presentation of specific parts is unclear.

Let me provide some details below:

1) There are a few inconsistencies in the paper.  In my opinion, some parts and claims are not totally clear based on prior works.
For example, the authors claim that there is an example that SEG-US and SEG-RR diverge. However, the example that they select is a simple bilinear problem for which we know now that the SEG-US converges to the solution with a proper selection of step-sizes (see Hsieh et al 2020).  In particular, the authors claim that SEG-US diverges in expectation for any positive step-size selection. This comes in contradiction with existing known theoretical results. Why is that the case? Can the authors provide a detailed comparison with prior work?

2) In several parts of the paper, the authors claim that EG is a good approximation of PP. This was already disproved in Gorbunov et al. 2022b, where the authors emphasize a significant difference between EG and the Proximal Point method.

3) The section 5 had the potential to be very informative. In the first paragraph of this section, the authors mentioned that the section is where the underlying cause for non-convergence of SEG-RR is investigated, and the motivation of sing anchoring is explained . However, the section is not clear, and several important parts are missing. The presentation of the "SECOND-ORDER MATCHING" part is not robust, which makes the understanding and importance of the results in section 5 unclear. For example, there is no proper definition of what second-order means in EG update rules, and the results are not positioned in the literature.

4)  #### On theoretical statements:
The authors wrote the following in their work: "$O(\eta^3)$ for the approximation error turns out to be the key to the exact convergence to an optimum under the monotone setting". This sentence, unfortunately, was not adequately explained in detail in the main paper. Why this statement is necessary?

5)  #### On theory:
Statement of the theorems do not include all necessary assumption. I imagine that Assumptions 3 and 4 need to hold for every theorem, but this is not precisely stated.

6)  #### Important comment on Assumptions:
The authors claim that Assumptions 3 and 4 are reasonable. They pointed out that the work of Golowich et al., 2020 uses Assumption 3(ii) to argue that this is a reasonable assumption. However, assumption 3(ii) is never used in any other paper on analysis for SEG (a first-order method). For example (Gorbunov et al., 2022b) remove assumption 3(ii) and still have nice convergence guarantees for SEG-US. In addition, Assumption 4 is also not reasonable to have. Of course, assumption 4 is more relaxed than bounded variance, but as explained in Gorbunov et al., 2022a this is not a necessary condition to show convergence. SEG-US does not require any bound on the stochastic gradient to guarantee convergence  (see Gorbunov et al., 2022a ),

7) #### On Experiments:
I believe algorithms in the experiments are compared with wrong step-size selection, which led to unfair comparison. That is, even if in Appendix A, all pseudocodes use $\alpha_t$ and $\beta_t$ to denote the extrapolation and update steps, respectively.
 in the experiments, $\eta_k=\alpha_k=\beta_k$ is used for all methods. However, for the bilinear example, it is known that using the same step-size leads to no convergence of SEG-US. See (Hsieh, et al., 2020)

8) Finally, the paper needs further experiments for completeness. For example, what is the practical benefit of the proposed method compared to other RR algorithms for solving strongly monotone problems?

----
Having shared the above concerns in statements and presentations, analyzing SEG with random reshuffling is exciting and relevant for the ML community. I will happily increase my score if answers are given to my above criticism. That is, can the authors relax the existing assumptions and heavily improve the presentation of this work?

**Questions:**

See the Weaknesses section.

---

> ### Author Response · Authors · 2023-11-22
> **Official Comment by Authors (1/2)**
>
> We appreciate the reviewer for acknowledging the importance of studying the shuffling-based SEG and for providing the comprehensive feedback.
>
> 1. **Comparing with (Hsieh et al., 2020) (W1):**
>
>     You are correct that the initial version was confusing, and we have now revised the paper (in footnote 2) so that it is clear that we focus on a *same-sample* version of the SEG (i.e., use the same component function for both extrapolation and update steps), which we believe is a more efficient and natural implementation of EG in without-replacement sampling scenarios. Hsieh et al. (2020) consider different independent samples for the two steps within one update of the SEG, so there is no contradiction between (Hsieh et al., 2020) and our results.
>
>     A noteworthy remark regarding the comparison of our algorithm against (Hsieh et al., 2020) is that the existing paper shows convergence to an optimum for a convex-concave problem but does not have an explicit rate; in contrast, we prove convergence with a rate.
>
> 2. **On whether EG is a good approximation of PP (W2):**
>
>     The claim by Gorbunov et al. (2022b) on the “significant” difference between EG and PP is based on the observation that PP update, as an operator, is always nonexpansive, while EG update may be expansive. In contrast, we look at the two algorithms from a different angle; our motivation was that a method that differs from EG by at most $\mathcal{O}(\eta^3)$ would lead to convergence in convex-concave settings, where $\eta$ is the step size. This was based on the observation, in (Mokhtari et al., 2020), that the update equations of the convergent methods—namely EG and PP (and even OGDA)—differ from each other by at most $\mathcal{O}(\eta^3)$ per epoch. Importantly, the updates of divergent methods such as GDA and various SEGs without FFA differ from the convergent methods by at least $\mathcal{O}(\eta^2)$, leading to our conjecture that the $\mathcal{O}(\eta^2)$ error is the main cause of nonconvergence. We believe this concern arose because of our unclear writing of Section 5, and we put efforts on improving the presentation of that section.
>
> 3. **On the presentation of Section 5 and second-order matching (W3-4):**
>
>     By *second-order matching* we mean choosing proper step sizes, sampling scheme, and anchoring scheme so that our without-replacement SEG can *deterministically* match the update equation of a convergent algorithm (EG or PP) up to the $\mathcal{O}(\eta^2)$ terms (*i.e.*, *second-order* terms in the Taylor expansion). We added the term “Taylor expansion” and improved corresponding sentences in the revision to make the definition of second-order matching clear.
>
>     We have also improved Section 5 to make our argument “$\mathcal{O}(\eta^3)$  for the approximation error turns out to be the key to the exact convergence to an optimum under the monotone setting” clearer. We provide a brief summary of Section 5 below for your understanding.
>
>     We show in Section 5.1 that the second-order matching is not achievable for both SEG-RR and SEG-FF. SEG-RR suffers a difference of at least $\mathcal{O}(\eta^2)$ from the convergent methods. For SEG-FF, as we show in Proposition 5.2, the best we can do is to choose specific step sizes and get an $\mathcal{O}(\eta_1^3)$ approximation of an instance of EG+ with $\eta_2 = 2\eta_1$. However, unfortunately, EG+ with $\eta_2 = 2\eta_1$ fails to converge already on bilinear problems. On the contrary, as we show in Proposition 5.3, SEG-FFA can approximate the standard EG with an error of $\mathcal{O}(\eta^3)$. In Section 5.2, we show that this second-order matching property of SEG-FFA indeed enables us to establish improved convergence rates.
>
> 4. **On the Assumptions for Theorems (W5):**
>
>     In the initial submission, statements of some theorems did not explicitly include the assumptions used. As correctly guessed by the reviewer, Assumptions 3 and 4 are used in most of the convergence proofs. We have modified the statements accordingly.

---

> ### Author Response · Authors · 2023-11-22
> **Official Comment by Authors (2/2)**
>
> 5. **Reasonableness of the assumptions 3 and 4 (W6):**
>
>     For our comments for Assumption 3, we invite the reviewer to check the general comment.
>
>     In regard of Assumption 4, the analyses of Gorbunov et al. (2022a) does not require any bound on the variance, but they do require the stochastic gradients to be sufficiently regulated in the following sense: assuming for each $i$ that the operator *$\boldsymbol{F}_i$* is $\mu_i$-strongly monotone (allowing $\mu_i < 0)$, it should hold that $\sum_{i=1}^{n} \mu_i ({\boldsymbol{1}_{{\mu_i \geq 0}}} + 4 \cdot \boldsymbol{1}_{ \mu_i  \textless 0 }) > 0$ where $\boldsymbol{1}$ denotes the indicator function (see eq. (9) in their paper). Thus, this is close to assuming strong monotonicity of the majority of component functions $\boldsymbol{F}_i$, and we believe that this already should provide sufficient level of “contraction” to show convergence without bounded variance assumptions. In stark contrast, we do *not* make any (strong) monotonicity assumptions on $\boldsymbol{F}_i$; our monotonicity assumptions only apply to their mean $\boldsymbol{F}$. Thus, we believe that our paper and (Gorbunov et al., 2022a) are not directly comparable because of different regularity conditions on the stochastic gradients. We now briefly mention this in footnote 3.
>
> 6. **On the experiments (W7-8):**
>
>     We hope now that your concern on our choice of same step size $\alpha_k = \beta_k$ in the experiment has been somewhat alleviated, considering that we focus on the same-sample case. Nevertheless, we learned that a numerical comparison with a convergent independent-sample SEG in (Hsieh et al., 2020) is necessary, so we added an experiment in which we discovered that SEG-FFA outperforms the method in (Hsieh et al., 2020). We also conducted an experiment on strongly monotone problems in which SEG-FFA again outpaced all other baseline methods.

---

> > ### Comment · Reviewer_emGH · 2023-12-04
> > **I keep my score.**
> >
> > I have read the rebuttal and the rest of the reviews.
> > I decided to keep my score.
> >
> > The authors posted a rebuttal just a few hours before the end of the discussion period, and they did not allow back-and-forth discussions with the reviewers.
> >
> > More importantly, the rebuttal and paper updates show that the authors are not aware of the main theoretical results in the area. The rebuttal includes misleading claims. Let me clarify.
> >
> > Like the original submission, I still believe this work has misleading statements, and the presentation of specific parts is unclear. I suggest the authors carefully check previous theoretical analyses and main assumptions used in prior works to update their paper. As I mentioned in my original review, this direction is interesting and worth exploring.
> >
> > 1. the current paper changes substantially compared to the original submission regarding presentation, theorems, claims, and statements. The current version does not even have experiments in the main paper, which, in my opinion, should have been one of the main contributions of the work.
> >
> > 2. Misleading statement: the authors claim that SEG converges in bilinear problems because it uses independent samples and not the same samples. This is not true. See the references in Hsieh et al. (2020) and Gorbunov et al. (2022a) for previous work in the area.
> >
> > 3. Misleading statement: The authors claim that Gorbunov et al.'s (2022a) results are close to assuming strong monotonicity of most component functions. This is also not true. The results can hold with only one F_i being strongly convex, and the rest have even negative \mu_i (not monotone).
> >
> > 4. Assumption 4 is very strong to have. If one aims to have meaningful theorems, one should not have such an assumption. Based on the current state-of-the-art analysis of SEG, this is not a reasonable assumption, as the authors claim.
> >
> > 5. The experiments in the first submission and current version are not sufficient. More experiments to justify the main theoretical claims of the paper are needed, and they should be part of the main paper.

---

### Author Response · Authors · 2023-11-22
**General comments**

We thank all the reviewers for their constructive feedback. We have made a major revision of the paper, addressing the concerns and comments raised by reviewers. Some noteworthy updates include the following:

* We now emphasize our main contribution by providing a comprehensive list of limitations of the existing analyses of SEG for the unconstrained finite-sum minimax problems, which we resolve all together in this paper. By further improving the related work section (Section 2), we believe that the current version better positions our work with respect to the prior literature (including those pointed out by
the reviewers).
* To address concerns on assumptions used in our paper, we also provide a better justification of the Hessian Lipschitzness assumption (Assumption 3(ii)). Please check our response below on Hessian Lipschitzness.
* We put our best efforts on improving the presentation of Section 5 and the discussion around “second-order matching” (in Taylor expansion), which was the key idea behind the development of the SEG-FFA.
* We moved our experiment section to Appendix H, and conducted larger-scale experiments spanning convex-concave and strongly-convex-strongly-concave cases. The experiments also include a comparison against existing convergent method by Hsieh et al. (2020) and an ablation study testing the effect of anchoring applied to uniform sampling (SEG-US) and random reshuffling (SEG-RR) versions of SEG.

Next, we would like to discuss some important issues commonly raised by the reviewers.

1. **On the misleading claims**

    Reviewers expressed concerns about statements that could be misleading, and we thank them for their careful reading and for drawing our attention to areas that could be enhanced in our paper. We have responded to each concern individually in details, and improved the paper accordingly.

2. **On the Hessian Lipschitzness assumption**

    The Hessian Lipschitzness assumption stems from the analysis of flip-flop sampling. In Proposition 5.3, we aggregate $2n$ stochastic updates (i.e., an epoch) of SEG-FFA and write it as a single update of deterministic EG from the start-of-epoch iterate $\boldsymbol{z}_0^k$, modulo a small additive noise (Proposition 5.3). In order to do that, the Hessian of component functions evaluated at the mid-epoch iterates should be close to those evaluated at $\boldsymbol{z}_0^k$; this is where Hessian Lipschitzness comes into play. Indeed, as correctly pointed out by Reviewer ZXRs, existing analyses of flip-flop sampling (Rajput et al., 2022) are limited to quadratic functions, which trivially have $0$-Lipschitz Hessians. Hence, our analysis is a step forward.

    We agree that this may look like a rather strong assumption. However, we would like to also point out that the Lipschitz Hessian assumption does not lead to unfair comparisons against other algorithms, since we show in our lower bounds that the baseline algorithms do not particularly benefit from this assumption. In particular, we show in Theorem 4.2 that all SEG-US, SEG-RR, and SEG-FF can diverge in convex-concave problems, and in Theorem 5.6 that SGDA-RR and SEG-RR cannot converge faster than SEG-FFA in strongly-convex-strongly-concave settings. Importantly, these lower bounds are proven with quadratic component functions; _i.e._, under $0$-Lipschitz Hessian condition. We leave studying whether one can remove this assumption as an interesting future work.

---

### Meta-Review · Area_Chair_cYDJ · 2023-12-07

**Metareview:**

The paper considers the problem of making the gradient small in finite-sum smooth (strongly )convex-(strongly )concave settings. It proposes a variant of a stochastic extragradient (SEG) method with anchoring that accesses the component functions from the finite sum using random permutations of a specific kind (standard random reshuffling and flip-flop shuffling proposed in recent work for quadratic problems). The paper establishes convergence bounds in smooth strongly convex-strongly concave settings (with matching, algorithm-specific lower bounds) and in smooth convex-concave settings, under additional assumptions involving Lipschitzness of the Jacobian and bounded variance of sampled component gradients.

While the paper is interesting and contributions are of value to the community, I (and also the reviewers) find it to not be ready for publication yet, despite the revisions provided in the rebuttal phase. The main issues have to do with the presentation and positioning of the results. The paper conflates finite sum and stochastic approximation settings, providing inaccurate and possibly misleading statements of the related work and the paper's position within it. In particular, one of the main criticisms of related work that the paper provides is that existing work (which unlike the present paper is on more general stochastic approximation settings, where the objective function is expressed as an expectation w.r.t. an *unknown* distribution, with possibly infinite support) makes an assumption about bounded variance. This is criticism is problematic for at least two reasons: (i) in all the related work discussed in the paper, handling the more general variance that scales with the squared norm of the operator is an easy exercise: the added "error" terms can easily be cancelled out by the negative operator squared terms used to argue about convergence by properly adjusting the step size; and (ii) in stochastic approximation settings, the dependence on variance is *unavoidable,* while the same is not true for the less challenging finite sum settings where we normally see no explicit dependence on the variance (of course, if the finite sum is solved as an empirical version of a stochastic approximation problem, then the number of component functions $n$ would necessarily depend on the variance for generalization bounds to hold, so there is possible *implicit* dependence).

The variance assumption that appears in the present paper is not needed for finite sum min-max problems, as can be seen from related work (e.g., Alacaoglu & Malitsky, though this is generally the case for other work on finite sum optimization). In the present paper, this assumption is made *because of* the sampling without replacement in the algorithm and the paper should have discussed that transparently. It is also worth noting that while a similar assumption is made in related work on shuffled SGD (for convex minimization problems) and it appears necessary, in related work it is assumed to hold only w.r.t. an optimum $x^*$ -- not w.r.t. all points $x$ as in the present paper, and this should be discussed appropriately.

Additionally, any work on (stochastic, variance-reduced) finite sum minimization usually includes, as a sanity check, comparison (in terms of the total number of arithmetic operations or runtime) to full vector update methods, to demonstrate that one is better-off using such methods in place of standard methods like EG. I do not see such a comparison in the present paper and I cannot even easily make it by looking at the results, as the table in the introduction and the theorem statements omit dependence on all problem parameters (i.e., they are stated primarily in terms of the iteration count).

I also disagree with authors that their anchoring step is completely different from what we usually see in Halpern-style methods. The anchoring step is *exactly* the Halpern iteration. I would go even further and say that we are anyway viewing the entire epoch as one iteration of the algorithm -- so the algorithm can be seen as being Halpern-type -- considering that the convergence guarantee holds only for the iterates at the beginning/end of an epoch. This is *not* a weakness of the work (I would even consider it a strength), but it needs to be properly discussed.

Finally, I would like to make my position clear here: I believe that the paper has valuable contributions and could be published at one of the top ML venues. However, for that to happen, the paper requires a very careful revision with clear statements and a clear comparison to the existing literature.

**Justification For Why Not Higher Score:**

The paper contains misleading statements and is not properly placed in the context of the existing literature. Even though these issues were raised by the reviewers, they were not properly addressed in the rebuttal, thus I do not feel confident that they would be addressed at all should be paper be accepted. I object to it being published in its current form.

**Justification For Why Not Lower Score:**

N/A

---

### Decision · Program_Chairs · 2024-01-16

Reject